# Targeting NRAS via miR-1304-5p or farnesyltransferase inhibition confers sensitivity to ALK inhibitors in ALK-mutant neuroblastoma

Perla Pucci [1], Liam C. Lee[1,17], Miaojun Han[1,18], Jamie D. Matthews [1], Leila Jahangiri[1,2,3], Michaela Schlederer[4], Eleanor Manners[1,19], Annabel Sorby-Adams[5,6], Joshua Kaggie [7], Ricky M. Trigg[1,20], Christopher Steel[1], Lucy Hare[1,8], Emily R. James [1], Nina Prokoph [1], Stephen P. Ducray [1], Olaf Merkel [9,10], Firkret Rifatbegovic [11], Ji Luo [12], Sabine Taschner-Mandl [11], Lukas Kenner [9,10,13,14,15], G. A. Amos Burke [8] & Suzanne D. Turner [1,10,16] ✉

Targeting Anaplastic lymphoma kinase (ALK) is a promising therapeutic strategy for aberrant *ALK*-expressing malignancies including neuroblastoma, but resistance to ALK tyrosine kinase inhibitors (ALK TKI) is a distinct possibility necessitating drug combination therapeutic approaches. Using high-throughput, genome-wide CRISPR-Cas9 knockout screens, we identify *miR-1304-5p* loss as a desensitizer to ALK TKIs in aberrant *ALK*-expressing neuroblastoma; inhibition of *miR-1304-5p* decreases, while mimics of this miRNA increase the sensitivity of neuroblastoma cells to ALK TKIs. We show that *miR-1304-5p* targets NRAS, decreasing cell viability via induction of apoptosis. It follows that the farnesyltransferase inhibitor (FTI) lonafarnib in addition to ALK TKIs act synergistically in neuroblastoma, inducing apoptosis in vitro. In particular, on combined treatment of neuroblastoma patient derived xenografts with an FTI and an ALK TKI complete regression of tumour growth is observed although tumours rapidly regrow on cessation of therapy. Overall, our data suggests that combined use of ALK TKIs and FTIs, constitutes a therapeutic approach to treat high risk neuroblastoma although prolonged therapy is likely required to prevent relapse.

Resistance to targeted agents such as tyrosine kinase inhibitors (TKIs) is a major cause of relapse and cancer related deaths. Indeed, clinical use of inhibitors of the tyrosine kinase, anaplastic lymphoma kinase (ALK) can lead to rapid relapse[1–3]. The mechanisms underlying this have yet to be fully elucidated. For example, in the case of aberrant *ALK*-expressing non-small cell lung cancer (NSCLC), resistance has been reported in the context of mutations of the target protein that render it insensitive to ALK TKIs and/or amplification leading to an excess of target[2–4]. However, resistance mechanisms can also be accountable to so-called bypass tracks which have yet to be fully elucidated[1,5,6]. It is therefore clear that the use of ALK TKIs as single agents is not ideal and combinations of inhibitors targeting multiple pathways might better prevent relapsed/refractory (r/r) disease and lead to a cure[7–9].

One of the cancers that expresses aberrant ALK and is driven by this protein is Neuroblastoma (NB), a solid tumour of the sympathetic nervous system, originating from neural crest cells that largely develops in the adrenal medulla and in the sympathetic ganglion chain running from the neck to the pelvis. NB is a complex and heterogeneous malignancy due to its varying molecular and clinical features which make it the most common and deadly extracranial solid malignancy in children, accounting for 7–8% of all childhood cancers and 15% of all paediatric malignancy-related deaths[10]. NB is classified as being of a low-, intermediate-, or high-risk with low and intermediate risk disease effectively responding to surgery or mild chemotherapy, with some rare patients experiencing spontaneous regression in less aggressive cases[11,12]. Nevertheless, most NB patients are diagnosed with high-risk NB (60%), for which there are a lack of effective treatments, resulting in a poor prognosis with a 5-year survival rate of less than 50%[11–16]. These patients sometimes present with aberrant molecular profiles often associated with aggressive phenotypes, such as *MYCN* amplification (20% overall and 50% of high-risk cases), *ALK* amplification and/or *ALK* kinase-domain point mutations (8–10% overall and 15% of high-risk cases)[11,12,17–19]. Recent evidence has also highlighted that telomere maintenance mechanisms as well as mutations in *RAS* and *TP53*, and alterations in their respective pathways may be a useful source for risk determination at diagnosis and subsequent treatment stratification[20].

For these children, novel therapeutic approaches are urgently needed and ALK TKIs are slowly making their way into the paediatric oncology setting[21,22]. However, r/r disease is a distinct possibility as discussed above, and has been reported for *ALK*-expressing NSCLC and anaplastic large cell lymphoma[4,23–25]. Hence, additional drug targets for combination therapy of NB and other ALK-aberrant cancers to prevent r/r disease are urgently needed[1–3].

In this work, we describe a genome-wide CRISPR-Cas9 knockout (GecKO) screen[26] to identify additional pathways modulating ALK TKI response in aberrant *ALK*-expressing NB cells. In particular, *miR-1304-5*p is identified, which we show acts as a tumour suppressor in NB cells via induction of apoptosis in vitro, by targeting the MAP Kinase pathway. These data suggest that combined inhibition of ALK and RAS, via upregulation of *miR-1304-5p*, might constitute a therapeutic approach to treat ALK TKI-naïve disease. Indeed, a combination of *miR-1304-5p* mimics and ALK TKIs increases the response to treatment of NB cells. Furthermore, the farnesyl transferase inhibitor (FTI) lonafarnib shows synergistic activity when administered to aberrant *ALK*-expressing NB cells in combination with ALK TKIs both in vitro and in vivo. Overall, this study shows that miR-1304-5p is an NB tumor suppressor and regulator of ALK TKI response, targeting oncogenic NRAS and paving the way for the use of miRNA- or FTI-based ALK TKI combination treatments for aberrant ALK-expressing NB.

## Results
### Genome wide CRISPR-Cas9 Knockout (GeCKO) screens identify miRNA targets as potential ALK TKI synergistic therapeutic vulnerabilities in NB

A GeCKO CRISPR screen was conducted, and quality control performed as detailed in Fig. 1A and Supplementary Fig. 1. The NB cell line SH-SY5Y that expresses the ALK^F1174L mutant (expressing Cas9) was exposed to the ALK TKIs ceritinib or brigatinib, at their respective effective dose (ED)$_{50}$ and ED$_{75}$ concentrations for 14 days, following lentiviral transduction of a sgRNA library containing 3 sgRNA/gene, targeting 19,050 genes and 1864 microRNAs (miRNAs)[27]. Surviving clones were subject to next-generation sequencing (NGS) to identify which genes, following their ablation, allowed the aberrant *ALK*-expressing NB cells to survive in the presence of an ALK inhibitor, therefore representing potential therapeutic targets. Analysis of the sequencing data by k-means clustering showed common but also diverging genes/microRNAs that on KO conferred a survival advantage

across different concentrations (300 and 750 nM) of ALK TKIs (brigatinib and ceritinib) (Supplementary Fig. 1C). For each treatment condition, an arbitrary threshold (>1.8) of fold difference (FD) in gene enrichment between DMSO and ALK TKI treatment was set, and the genes with multiple sgRNAs exceeding the FD threshold were considered to be significantly enriched (Fig. 1B, C and Supplementary Fig. 1D, Supplementary Data 1). The stringency of these data was further increased, as only those genes which on knockout enabled cell survival following exposure to both brigatinib and ceritinib at either concentration (i.e., 300 nM or 750 nM) were identified as 'hits' (Fig. 1B, C and Supplementary Fig. 1D). Within these data, 9 miRNAs met these criteria, of which only one (*hsa-mir-136*) was common to the collective higher and lower doses of both ALK TKIs, suggesting that different sensitivities may arise at variable TKI concentrations (Fig. 1B). The top 4 miRNAs (*hsa-mir-136, hsa-mir-1304, hsa-mir-7975*, and *hsa-mir-4746*) were identified in at least 3 of the 4 screens (Fig. 1C–G). As miRNAs are negative regulators of several signalling pathways, these were further investigated.

### Inhibition of miRNA *miR-1304-5p* decreases sensitivity to ALK TKIs in an ALK mutant NB

To validate the GeCKO screen hits individually, inhibitors of *hsa-miR-136-5p*, *hsa-miR-1304-5p*, *hsa-miR-4746-5p*, or *hsa-miR-7975-5p*, along with a scrambled negative control, were individually transfected into the SH-SY5Y cell line and their expression was validated via RT-qPCR (Supplementary Fig. 2A–D). At 48 h following transfection, the cells were treated with increasing concentrations of brigatinib (Fig. 2A, B) or ceritinib (Fig. 2C, D) for 72 h. We also report the genomic location of these miRNAs and whether they cluster with other miRNAs or belong to the same family (Suppl. Tab. 1). Of note, only *miR-1304* and *miR-136* belong to a known miRNA family, and only *miR-136* clusters with other miRNAs (<10 kb from *hsa-mir-136*). Of all the miRNAs analysed, only inhibition of *miR-1304-5p* significantly decreased sensitivity to both of the ALK TKIs, brigatinib (Fig. 2A, B) and ceritinib (Fig. 2C, D).

### *miR-1304-5p* inhibits NB cell viability and induces apoptosis

In order to determine whether *miR-1304-5p* acts as a tumour suppressor for NB, an inhibitor of *miR-1304-5p* was transfected into a panel of 17 NB cell lines of differing ALK, p53, and MYCN status (Suppl. Table 2) (Fig. 2E). Among the cell lines tested, 16 demonstrated a significant increase in viability when *miR-1304-5p* was inhibited. In keeping with these observations, mimics of mature *miR-1304-5p* transfected into the same cell line panel led to a significant reduction in cell viability for all of the NB cell lines (Fig. 2F).

To identify the cellular mechanism(s) underlying the observed decrease in cell viability of NB cells transfected with mature *miR-1304-5p* mimics, cell cycle and apoptosis were assessed. Transfection of a *miR-1304-5p* mimic into cells led to a 17000-fold increase in *miR-1304-5p* expression in SH-SY5Y (ALK F1174L, MYCN non-amplified; Supplementary Fig. 3A) and a 3000-fold increase in KELLY cells (ALK F1174L, MYCN amplified; Supplementary Fig. 3B). Correspondingly, a significant ($p < 0.05$) increase in caspase 3/7 activity representative of apoptosis was observed 72 h after transfection of the miRNA in both SH-SY5Y and KELLY cell lines (Supplementary Fig. 3E). Induction of apoptosis following transfection of cells with a *miR-1304-5p* mimic was confirmed via WB showing increased levels of PARP cleavage (Supplementary Fig. 3F), and increased numbers of Annexin V-positive cells detected by flow cytometry (Supplementary Fig. 3G, H). However, no significant effects on the cell cycle were observed (Supplementary Fig. 3C, D and Supplementary Fig. 4A–L).

### *miR-1304-5p* targets NRAS

To identify the molecular mechanisms driving the cellular phenotype induced by *miR-1304-5p*, genome-wide expression microarray (HT-12 v4) analysis was performed on SH-SY5Y and KELLY cells transfected

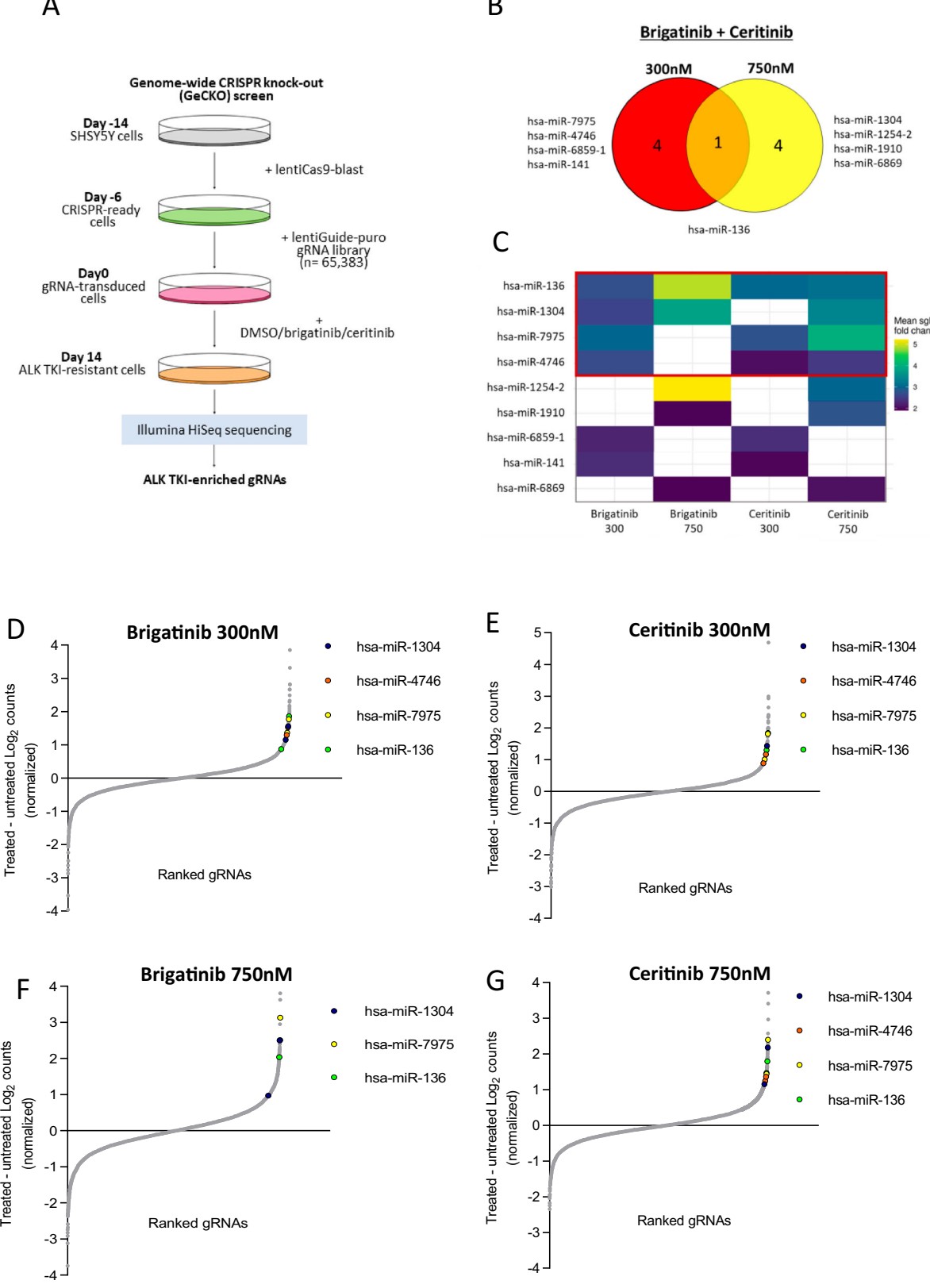

with a *miR-1304-5p* mimic or scrambled negative control, detecting 1267 and 1103 altered genes in SH-SY5Y and KELLY cell lines, respectively (Fig. 3A, Supplementary Data 2 and 3). Gene set enrichment analysis (GSEA)[28] was then performed with MSigDB's 'hallmark gene sets' (Supplementary Data 4)[29]. The top-10 enriched gene sets were ranked with a positive normalized enrichment score (NES) index by the size of genes

in each gene set (i.e., the number of genes identified in each gene set that are included in the input data); for both KELLY and SH-SY5Y cell lines, 6 gene sets were identified that were significantly ($p < 0.05$ in at least one cell line) altered following *miR-1304-5p* transfection with a negative correlation: 'xenobiotic_metabolism', 'epithelial_mesenchymal_transition', 'estrogen_response_late', 'interferon_gamma_response',

**Fig. 1 | A GeCKO screen conducted in SH-SY5Y NB cells identifies genes that following CRISPR cas9-induced excision, alter sensitivity to ALK TKIs.**
**A** Experimental schema of the CRISPR GeCKO screen conducted on SH-SY5Y cells stably expressing Cas9 and transduced with the GeCKO Version 2 sgRNA library before incubation with the indicated ALK inhibitors, followed by high throughput sequencing. **B** Venn diagram of miRNA genes identified by the CRISPR GeCKO screens, whose loss of expression desensitizes SH-SY5Y cells to brigatinib and ceritinib (at 300 nM and/or 750 nM concentrations of each drug). **C** Heatmap of miRNA hits identified in both brigatinib and ceritinib screens (at 300 nM and/or 750 nM of each drug) showing the top 4 miRNAs that were identified in ≥3 screens (in the red square). Colored squares represent the mean values of fold changes (ALKI TKI versus DMSO treated) of the selected sgRNAs, from lower (blue) to higher (yellow) fold changes. **D**–**G** gRNAs ranked by fold-change vs difference in $\log_2$ normalized read counts between DMSO and ALK TKI-treated SH-SY5Y cells. Only miRNA sgRNAs common to brigatinib and ceritinib treatments in at least three GeCKO screens are shown (*has-miR-1304*, *has-miR-136*, *hsa-miR-7975*, *has-miR-4746*). Source data are provided in a Source Data file.

'glycolysis, kras_signaling' and 'mtorc1_signaling' (Fig. 3B and Supplementary Data 4).

To investigate *miR-1304-5p* target genes, we next compared the genome-wide expression microarray data with miRNA-target in silico predictions. The 3′-UTRs of 3,336 coding genes were predicted to bind *miR-1304-5p* by Target Scan (cumulative weighted context ++ score ≤ −0.01; Fig. 3A and Supplementary Table 4). Among these predicted target genes, the mRNA levels of 393 and 391 genes were downregulated in SH-SY5Y and KELLY cell lines respectively according to our microarray results, and 276 in both cell lines (Fig. 6A). KEGG pathway analysis of these 276 genes, showed that 28 were involved in pathways associated with cancer regulation (Fig. 3A, Supplementary Fig. 6A). Expression of these 28 genes was validated via RT-qPCR after transfection with *miR-1304-5p* mimics into SH-SY5Y cells (Supplementary Fig. 5A). Among them, 22 genes were downregulated by more than 20% of their original expression levels due to the *miR-1304-5p* mimic (Supplementary Fig. 5A). Based on the GSEA analysis and evidence gleaned from the literature[30–32], 4 genes associated with the RAS/MAPK pathway were singled out as potential *miR-1304-5p* targets and therefore selected for further investigation: *NRAS* (NB RAS viral oncogene homolog), *RRAS* (related RAS viral oncogene homolog), *PTPN11* (Protein Tyrosine Phosphatase, Non-Receptor Type 11; also referred to as *SHP2*) and *IQGAP1* (IQ motif containing GTPase activating protein 1).

The expression levels of NRAS, RRAS, PTPN11, and IQGAP1 were assessed by RT-qPCR and a Western blot upon transfection of a *miR-1304-5p* mimic into SH-SY5Y and KELLY cells (Supplementary Fig. 5A, Fig. 3C, Supplementary Fig. 7). NRAS, RRAS PTPN11 and IQGAP1 levels were significantly reduced following transfection of the *miR-1304-5p* mimic (Fig. 3C), with phospho-ERK protein levels also lowered (Fig. 3C). In order to confirm *miR-1304-5p* regulation of NRAS, RRAS, PTPN11 and IQGAP1 expression, the 3′-UTRs of these 4 genes were cloned into the psiCHECK2 dual-luciferase plasmid vector and then individually transfected into SH-SY5Y cells. Luciferase activity was significantly decreased for all the target genes upon *miR-1304-5p* mimic transfection, suggesting that NRAS, RRAS, IQGAP1 and PTPN11 are targets of *miR-1304-5p* (Fig. 3D). Notably, rescue experiments conducted by transduction of the cDNAs of the 4 genes (including ORF and 3′ UTR regions) into SH-SY5Y cells showed an increase in viability (Fig. 3E) and furthermore, overexpression of NRAS, RRAS, IQGAP1 or PTPN11 (Supplementary Fig. 5B–D) counteracted the effects of *miR-1304-5p* on RAS/MAPK pathway target gene expression and activity as indicated by NRAS, RRAS, IQGAP1, PTPN11 and pERK expression (Fig. 3E and Supplementary Fig. 5B–D). Furthermore, analyzing previously published expression data derived from NB primary patient tumours of the TARGET dataset[33], we show that *NRAS* transcript levels positively correlate with *PTPN11* but not with *RRAS* or *IQGAP1* (Supplementary Fig. 6B–D). Additionally, *NRAS* is of prognostic significance in the same cohort of NB patients whereby those with higher levels of *NRAS* have a worse prognosis ($p < 0.05$; Fig. 3F). No significant prognostic value could be associated with *RRAS*, *IQGAP1* nor *PTPN11* expression.

Notably, analysis of a second patient dataset (Kocak) showed that NRAS expression levels predict for prognosis whereby higher expression levels correlate with an inferior event-free (EFS) and overall survival (OS) in keeping with data shown in Fig. 3F for the TARGET dataset

($p = 0.034$, 79 vs 31 months median survival; log-rank test). Unfortunately, analysis of other clinical variables in the Kocak dataset including age, disease stage and sex is largely not feasible due to the low numbers of patients that fall into these categories. For example, there is a trend towards a correlation between NRAS expression with survival in patients >18 months of age and this analysis is significant for EFS ($p = 0.032$) but not for OS (Supplementary Fig. 5H, I). In addition, stratification for survival according to disease stage showed a significant correlation between high NRAS expression and stages 1, 2 and 4, with a stronger trend and significance reached for EFS at stage 4 ($p = 0.018$) (Supplementary Fig. 5I). Notably, there is also a significant correlation between NRAS expression and survival for male patients ($p = 0.0008$ for OS and $p = 0.0006$ for EFS; Supplementary Fig. 5H, I) whereby male patients with higher expression levels of NRAS have a significantly inferior outcome. According to these data, male patients and/or those with a higher stage disease may stand to benefit more from this therapeutic approach. The correlation between NRAS expression with lower OS and EFS observed in the Kocak dataset is largely confirmed in the SEQC dataset (especially for the entire patient cohort, and according to age and sex) but due to the reduced number of patients, these correlations are not significant (Supplementary Fig. 5J, K). In the SEQC dataset we also observed a stronger correlation between higher NRAS expression with lower survival in high-risk patients and this is significant for EFS ($p = 0.002$; Supplementary Fig. 5K). Based on these analyses, patients with high-risk NB may benefit more from NRAS-targeting therapeutic approaches.

Additionally, *NRAS* is the only gene amongst our target dataset that is significantly upregulated in ceritinib resistant ($n = 2$) versus sensitive ($n = 2$) orthotopic NB xenograft models[34] (Supplementary Fig. 5E). Therefore, we focused on the potential of NRAS as a *miR-1304-5p* target for further investigation as a therapeutic target.

### *miR-1304-5p* and *NRAS* are therapeutic targets in combination with ALK TKIs in aberrant *ALK*-expressing NB cell lines

In order to investigate the therapeutic utility of ALK TKIs and *miR-1304-5p* in NB, cells were transfected with *miR-1304-5p* mimics prior to ALK TKI treatment, which significantly decreased the $ED_{50}$s for both brigatinib and ceritinib in *MYCN* non-amplified SH-SY5Y ($p = 0.0047$ and $p = 0.0042$, respectively; Fig. 4A, B), *MYCN* amplified KELLY ($p = 0.0038$ and $p = 0.0002$, respectively; Fig. 4C, D) and *MYCN* non-amplified ALK wild type LA-N-6 cells (Supplementary Fig. 10A, B). These data suggest that miRNA-based therapies could be promising strategies to sensitize cells to ALK inhibition. Indeed, transfection of NB cells with *miR-1304-5p* mimics prior to treatment with ALK TKIs significantly increased apoptosis as determined by caspase 3/7 activity compared to ceritinib treatment alone for SH-SY5Y, KELLY ($p < 0.0001$ and $p < 0.0001$, respectively; Fig. 4E, F) and LA-N-6 cells ($p = 0.0007$; Supplementary Fig. 10C) as well as for brigatinib treatment alone for SH-5Y5Y ($p = 0.045$; Fig. 4E, F) and LA-N-6 cells ($p = 0.023$; Supplementary Fig. 10C).

However, the latter did not reach significance for the KELLY cell line perhaps due to a dosage effect. Notably, we validated this effect on apoptosis via both flow cytometry to assess Annexin V/PI staining (Supplementary Fig. 8A–D) and WB for PARP cleavage as well as showing inhibition of downstream NRAS signaling (specifically by

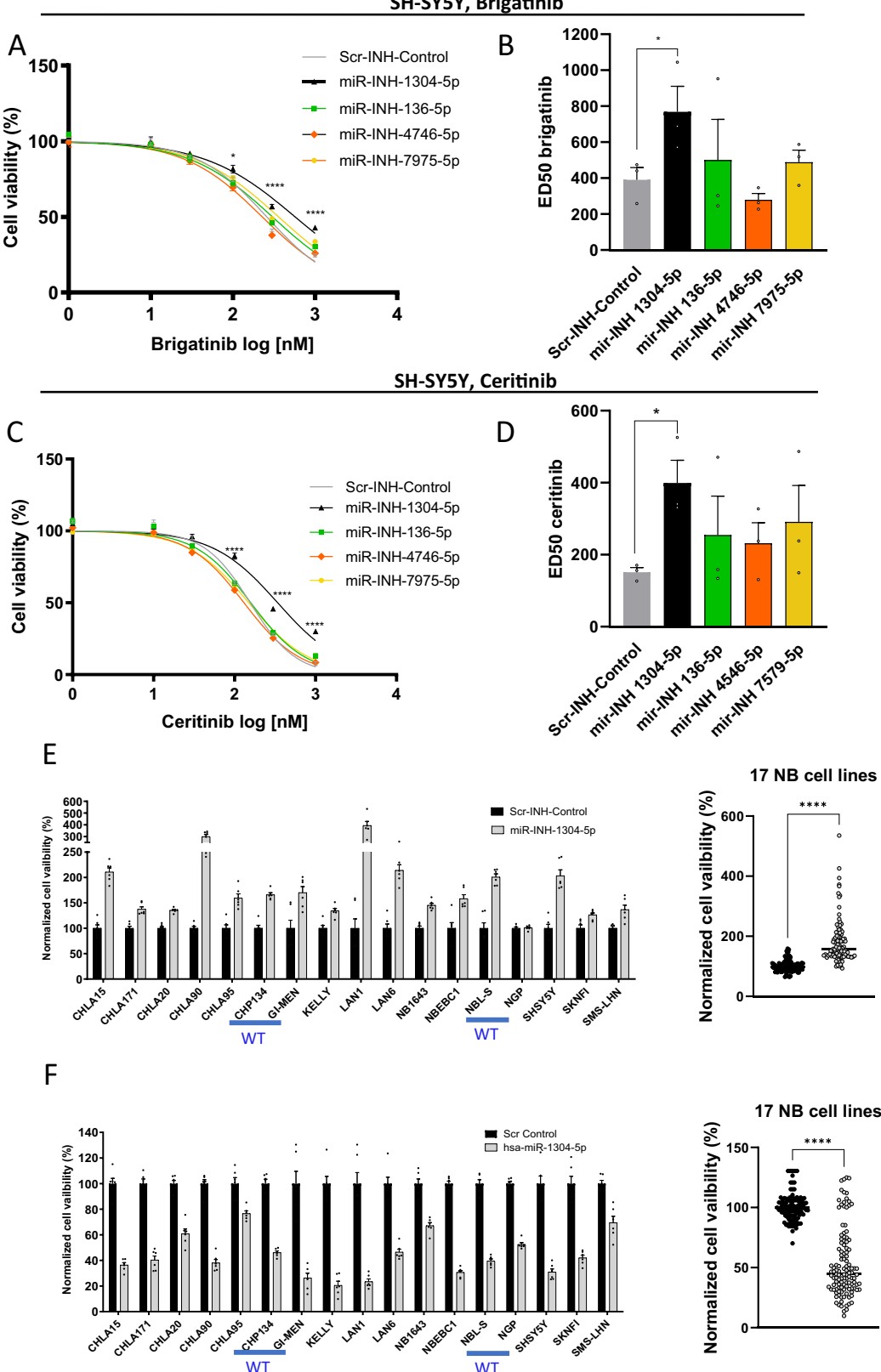

examining phospho-ERK levels) (Supplementary Fig. 8E, F), using a combination of a *miR-1304-5p* mimic and an ALK TKI (ceritinib) in KELLY and SHSY5Y cells lines.

As miRNAs are yet to be efficaciously implemented as therapeutics in the clinic, we chose to study the effects of the validated *miR-1304-5p* target, *NRAS* whose expression also has prognostic significance in NB (Fig. 3F and Supplementary Fig. 5F–K). NB cells were transfected with NRAS Dicer-Substrate Short Interfering RNAs (DsiRNA) prior to ALK TKI treatment. NRAS knockdown was confirmed via RT-qPCR and WB (Supplementary Fig. 9A, B) and significantly decreased the $ED_{50}$ values for both brigatinib and ceritinib in ALK-mutant NB cells (SH-SY5Y) (Supplementary Fig. 9C, D). As

**Fig. 2 | *miR-1304* inhibition affects viability of a range of NB cell lines and decreases sensitivity of ALK mutant NB cells to brigatinib and ceritinib.** **A**–**D** SH-SY5Y cell viability and ALK TKI ED$_{50}$s, 5 days post transfection of the indicated miRNA inhibitors followed by 72 h exposure to either brigatinib (**A**, **B**) or ceritinib (**C**, **D**). All results are normalized to the viability of the untreated (DMSO) control for each miRNA inhibitor treatment conducted. ED$_{50}$s were calculated using a non-linear regression curve. Results are shown as means ± SEM from three biological replicates each conducted with technical triplicates. Statistical comparison was conducted using a two-tailed Student's *t* test of the means of three biological replicates in **B**, **D**; **B** *p* = 0.037, **D** *p* = 0.019. **E**, **F** Cell viability 72 h following transfection with **E** the *miR-1304-5p* inhibitor or **F** a *miR-1305-5p* mimic in a panel of 17 NB cell lines. ALK WT cells are highlighted in blue = CHP-134, GIMEN, NBL-S, NGP. The other cell lines have activating ALK mutations, as specified in Supplementary Table 2. Data points (*n* = 6) shown in the graphs on the left are from six technical replicates (independent transfections). Statistics were conducted using the means ± SEM from 17 different cell lines (biological replicates) with a two-tailed Student's *t* test; ****$p < 10^{-15}$. Source data are provided in a Source Data file.

expected, transfection of NB cells with NRAS siRNA prior to treatment with ALK TKIs significantly increased apoptosis as determined by caspase 3/7 activity compared to either ceritinib or brigatinib treatment alone (Supplementary Fig. 9E). Overall, these data suggest that the *miR-1304-5p* target, NRAS, is a valid therapeutic target.

NRAS is of prognostic significance for NB (Fig. 3F and Supplementary Fig. 5F–K) and has previously been shown to be a potential druggable target[35,36]. However, there are no specific NRAS inhibitors and those drugs that target downstream pathway proteins such as MEK have previously been shown to lack efficacy in NB[37]. Therefore, we treated cells with lonafarnib (Zokinvy), an oral FTI that was recently (November 2020) approved for the treatment of Hutchinson-Gilford Progeria Syndrome (HGPS or Progeria) and processing-deficient Progeroid Laminopathies (PL)[36,38]. Whilst FTIs have multiple targets, we reasoned that their activity upstream in the MAPK and other RAS-regulated pathways would provide broader inhibition of downstream pathways and in doing so, prevent pro-survival feedback signaling from occurring as has previously been reported in NB with the use of MEK inhibitors[37]. Effective concentrations of single agents were determined for use in drug combination experiments by treating cells with increasing doses and measuring cell viability (Fig. 5A–C). Treatment of SH-SY5Y cells with lonafarnib in combination with the ALK TKIs decreased cell viability compared to administration of ALK TKIs alone, with additive effects at lower concentrations (synergy score: > −10 and <10) and synergistic effects at higher concentrations (synergy score: >10) for both brigatinib (Fig. 5D, Supplementary Fig. 11A) and ceritinib (Fig. 5E, Supplementary Fig. 11B). All concentrations are clinically achievable as they are below the lower C$_{max}$ per dose used in the clinic[39]. Notably, this combination treatment was additive, rather than synergistic at higher concentrations in *MYCN* amplified KELLY cells, suggesting that MYCN status could affect the cellular response to lonafarnib when given in combination with brigatinib (Fig. 5F, Supplementary Fig. 11C) or ceritinib (Fig. 5G; Supplementary Fig. 11D). Similar to the effects observed on treatment of cell lines with *miR-1304-5p* mimics in combination with ALK TKIs, the combination of brigatinib or ceritinib with lonafarnib increased apoptosis as determined by caspase 3/7 activity in SH-SY5Y (Fig. 5H) and KELLY cells (Fig. 5I), although this increase did not reach significance for brigatinib and lonafarnib treated KELLY cells. Notably, these synergistic effects on apoptosis were confirmed via Annexin V staining assessed by flow cytometry (Supplementary Fig. 12A–D) and PARP-cleavage via WB (Supplementary Fig. 12E, F) as well as showing inhibition of downstream NRAS signaling (specifically phospho-ERK levels) via WB (Supplementary Fig. 12E, F). Notably, particularly high synergism was noted between the FTI and the third generation ALK inhibitor lorlatinib, for both the ALK mutant PDX, COG-N-415 (Supplementary Fig. 13A–D) and the ALK WT NGP cell line (Supplementary Fig. 13E–H).

## ALK TKIs and lonafarnib show synergistic activity with tumour regression in the treatment of *ALK*-mutated NB patient derived xenografts

Due to the noted effect of ALK TKI combination treatments with both *miR-1304-5p* mimics or lonafarnib in NB cells, we further investigated this regulatory axis in 3 ALK mutant, high-risk NB PDXs: PDX FELIX−COG-N-426x (ALK$_{F1245C}$; MYCN WT), COG-N-415 (ALK$_{F1174L}$; MYCN

amplified) and COG-N-557 (ALK$_{F1245L}$; MYCN amplified) (Fig. 6A–F, Supplementary Fig. 14). In all cases, this combination resulted in decreased cell viability compared to administration of either agent alone, with highly synergistic effects observed at higher concentrations (synergy score: >10) (Fig. 6A, C, E, Supplementary Fig. 14A–C) accountable to increased apoptosis (Fig. 6B, D, F). Of note, COG-N-415 PDX cells grow more aggressively than the other PDX cells, and despite the sub-optimal effect observed on monotherapy (exposure to either lonafarnib or ceritinib alone), significant synergy was observed between ALK TKIs and lonafarnib (Fig. 6E, F).

Subsequently, NSG mice were injected sub-cutaneously with early passage NB PDX cells (FELIX-PDX COG-N-426x or COG-N-415x) suspended in Matrigel, and when the tumours reached ~75 mm³ the mice were treated daily with either vehicle (20% hydroxypropyl-beta-cyclodextrin), single-agent ceritinib (30 mg/kg), single-agent lonafarnib (40 mg/kg) or both agents in combination (Supplementary Fig. 15). Single-agent treatment with ceritinib or lonafarnib significantly delayed tumour growth compared to vehicle alone (median EFS = 21.5 days, *p* < 0.001, and median EFS = 18.5 days, *p* < 0.001 versus median EFS = 10.5 days respectively; an endpoint event was defined as a tumour reaching 15 mm in any direction; Fig. 7A, B). Combination treatment with ceritinib and lonafarnib led to an exceptional response with all the animals in this treatment group showing no obvious signs of tumour expansion in the following 30 days of continuous treatment (Fig. 7A, B). All compounds were well-tolerated, with no significant decrease in body weight nor lethal toxicity observed (Fig. 7C and Supplementary Fig. 17A). MRI analysis was performed after 30 days of daily treatment and showed that one of the mice had no detectable tumour although the others had residual tumours of 1–6 mm diameter (Fig. 7D). However, 2 mice with small palpable tumours from the combination treatment group analysed at day 9 and 10 following cessation of treatment showed tumour progression (Fig. 7D, Supplementary Fig. 17B). Indeed, by 24 days following cessation of treatment, all of the mice in the combination treatment group (including the mouse with no tumour observable by MRI) had relapsed, suggesting that continual and/or further optimisation of the dosing regimen is required to sustain control of tumour growth (Supplementary Fig. 17C).

Tumours were harvested from the vehicle, lonafarnib and ceritinib-only treated mice at the experimental endpoint showing typical NB morphology of small, round, monomorphic cells with nuclear hyperchromasia and scant cytoplasm (Supplementary Fig. 17D). As expected, treatment with lornafarnib or ceritinib led to a decrease in pERK expression levels although this was more pronounced in the lornafanib-treated mice (Fig. 7E and Supplementary Fig. 17E, D).

A second in vivo study was conducted with subcutaneous injection of COG-N-415x PDX cells (Supplementary Fig. 16A–D), which confirmed data produced with the COG-N-426x PDX. Notably, the COG-N-415x PDX line is MYCN amplified compared to the MYCN non-amplified COG-N-426x line, confirming that our findings could be relevant to a broader cohort of patients. Given that MAPK inhibitors have previously been shown to lack efficacy in ALK-aberrant NB due to activation of a negative feedback loop through PI 3-Kinase, we also examined expression of pAkt in the tumour cells on inhibition of NRAS

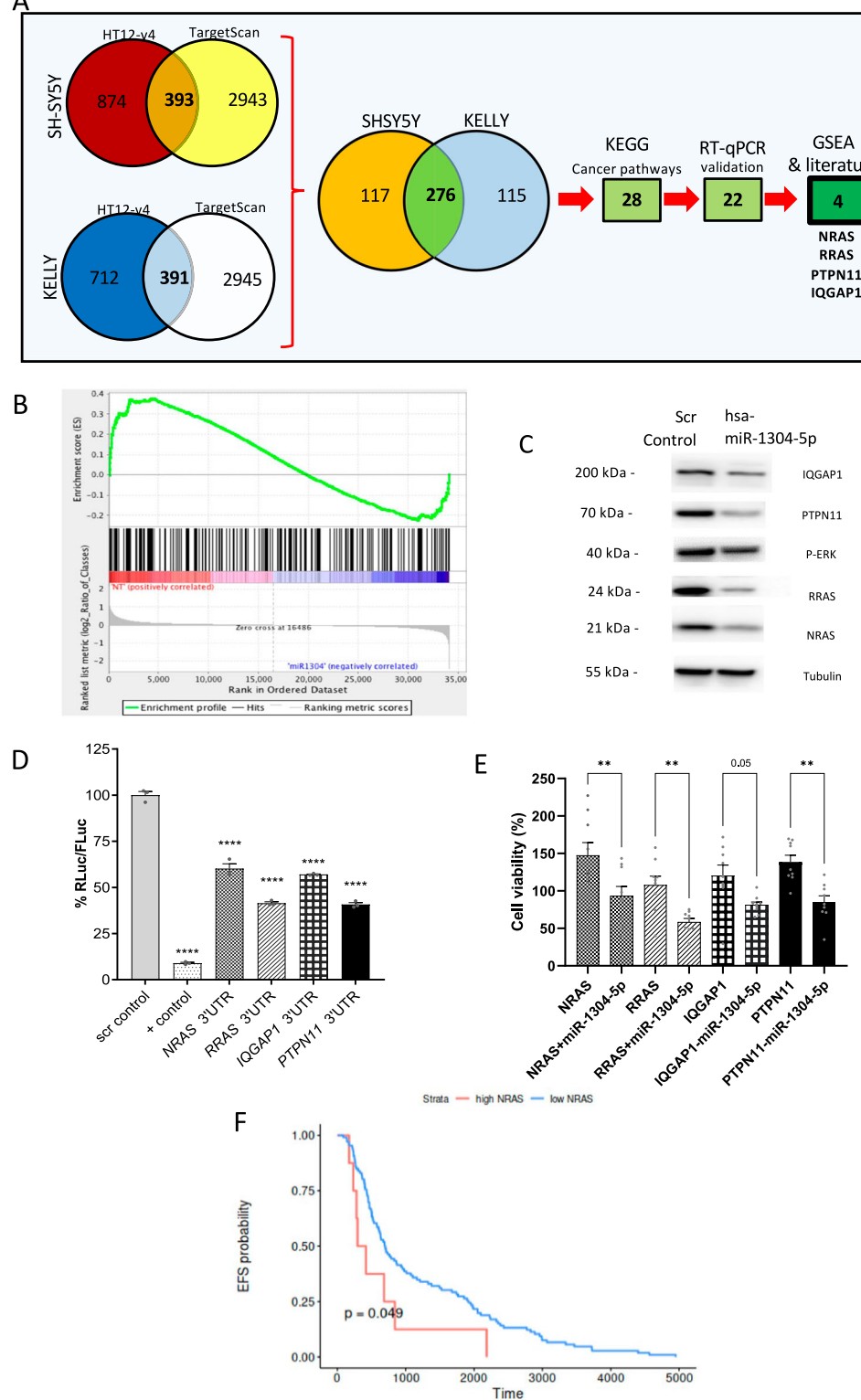

via a specific siRNA (Supplementary Fig. 18A), and in the PDX tumours shown in Fig. 7E (Supplementary Fig. 18B). No change in pAkt levels were observed, suggesting that targeting proteins upstream of MAPK (and other RAS-regulated pathways) prevents this feedback loop from occurring.

Overall, these studies confirm the high efficacy and tolerability of combined ALK TKI and FTI treatment in xenograft models of high-risk, ALK-aberrant, NB.

## Discussion

Resistance to chemotherapy is a major contributor to poor EFS and OS outcomes in cancer[1,40]. The introduction of targeted therapies into clinical practice paved the way for personalized medicine and improvements in the efficacy of cancer treatment but resistance remains a problem and has been reported in many studies, including those conducted with ALK TKIs[2–4,41]. Clinical trials for ALK TKIs in the treatment of paediatric malignancies including NB are in their relative

**Fig. 3 | *miR-1304-5p* inhibits NRAS expression. A** Experimental schema to identify target genes of *miR-1304-5p* with selection of the top-target genes (*NRAS, PTPN11, IQGAP1, RRAS*). **B** GSEA of 'Hallmark_KRAS_signaling_up'. GSEA was performed to rank the genes based on their differential expression levels between cells transduced with a non-targeting control (NT) compared to a *miR-1304-5p* mimic (miR1304). A deviation from an enrichment score (ES) of 0 in the profile (green line) reflects the degree of overrepresentation of genes at the top (positive ES score) or bottom (negative ES score) of the ranked gene list. Expression profiles derived from biological triplicate samples were analyzed by GSEA. **C** Western blot of the indicated proteins 72 h post transfection of SH-SY5Y with a *miR-1304-5p* mimic representative of two independent experiments with similar results. **D** Dual luciferase assay signal driven from the 3'UTR of the 4 target genes transfected into SH-SY5Y, upon *miR-1304-5p* mimic transfection (ratio of target gene-firefly luciferase (Fluc)

to the Renilla luciferase (Rluc) control vector and normalized to the negative control (Scr control)). Data points of $n = 3$ biological replicates are shown with bars representing the means ± SEM of the three biological replicates. **E** Cell viability (CTG) of SH-SY5Y cells transduced to express each of the 4 target genes on the left panel and the 4 target genes ± a *miR-1304-5p* mimic normalized to empty vector transduced cells in the right panel. Data points of $n = 3$ biological replicates (each with three technical replicates) are shown with bars representing the means ± SEM of three biological replicates. Statistical comparison was conducted using a one-way ANOVA with Tukey's post-test (**D**, **E**); **D** ****$p = 3.3 \times 10^{-13}$, ****$p = 2.52 \times 10^{-9}$, ****$p = 1.03 \times 10^{-11}$, ****$p = 9.99 \times 10^{-10}$, ****$p = 8.99 \times 10^{-12}$; **E** **$p = 0.0034$, **$p = 0.0083$, **$p = 0.0039$. **F** EFS of 143 NB patients with high or low expression of *NRAS* (z-score=1), measured by bulk RNA Sequencing[33] (Log-rank test, *$p = 0.049$). Source data are provided in a Source Data file.

infancy[22,42–46]. Nevertheless, de novo and acquired resistance can occur for all first, second, and third generation ALK inhibitors, thereby making ALK TKI efficacy challenging[2,3,41,47–49]. Hence, additional targets for combination treatments of NB and ALK-aberrant cancers are urgently needed[1–3,8,9].

In this study, we performed a CRISPR-Cas9 GeCKO screen of NB cells in order to investigate the genes that when downregulated render cells less sensitive to ALK TKIs. These genes and the pathways in which they reside could play a role as tumour suppressors in signalling pathways that sensitise NB to ALK inhibitors, thereby representing potential therapeutic targets for drug combinations. Via our GeCKO screen, NB cells for which gene-specific knockout permitted cell survival in the presence of ALK TKIs (i.e., decreased sensitivity to ALK TKIs) were identified. Most of the genes identified have either previously not been associated with cancer, or act as known tumour suppressors in a variety of cancers, including NB[50–52]. In the latter regard, *sMEK1* is a gene that encodes a Serine/threonine-protein phosphatase that stimulates apoptosis in response to paclitaxel treatment in ovarian cancer via inhibition of mTOR signalling[53], and *NPRL2* promotes sensitivity to irinotecan in colon cancer by activating the DNA damage checkpoint pathway[51]. Notably, one lncRNA was also identified in our "top" hit list, *MRPL39*, which acts as a tumour suppressor in gastric cancer[52]. Most of the miRNAs identified by the screens have previously been reported to act as tumour suppressors in NB and other cancers, including *miR-1304* and *miR-141*, the latter via stimulation of drug sensitivity in NB[50,54,55].

We highlighted *miR-1304-5p* inhibition as a potential desensitizer of NB cells to ALK TKIs. Increasing evidence implicates non-coding RNAs (ncRNAs) in response to cancer treatment[56–59] with miRNAs often downregulated in resistant tumour cells with subsequent upregulation of the oncogenes they regulate[57,58]. As such, novel therapeutic approaches have been studied including ncRNA-based treatments such as antisense-oligonucleotides (ASOs) and miRNA mimic-based formulations[57,60]. Hence, the potential of targeting *miR-1304-5p* in *ALK*-aberrant NB was explored further. Indeed, *miR-1304-5p* inhibition enhanced cell viability, in multiple genetically distinct NB cell lines regardless of ALK or other genetic status; *miR-1304-5p* upregulation had the opposite effect, suggesting this as a potential therapeutic vulnerability across multiple genetic sub-types of NB. The decrease in cell viability on *miR-1304-5p* upregulation was attributed to apoptosis rather than cell cycle arrest, suggesting that this might represent an efficacious therapeutic approach in keeping with data previously reported by Li et al., which identified *miR-1304* as a tumour suppressor in NSCLC[61].

Targeted delivery of miRNAs can simultaneously affect a range of proteins, therefore having great potential as therapeutic agents, as shown by pre-clinical studies of pancreatic and liver cancer[62,63]. Furthermore, miRNA mimics have also been investigated in clinical trials, suggesting that miRNA induction is a promising cancer treatment although not yet clinically actionable[64,65]. Despite this, clinical miRNA-based therapeutics are still in their infancy and further studies are

needed to establish well-tolerated doses and delivery methods to increase the overall treatment benefits and to avoid high toxicity[64,65]. Additionally, efficient miRNA delivery requires chemical modifications due to their instability and rapid degradation in biological fluids. This might affect the specificity of miRNAs and lead to off-target and saturation effects, thus leading to unexpected gene regulation and toxicities[66–68].

As over-expression of miRNAs is yet to be successfully applied in the clinic, we identified targets of miR-*1304-5p* as being potential alternative therapeutic options. Genome-wide expression arrays identified *miR-1304-5p* to significantly impact four gene sets, including those associated with the RAS/MAPK pathway. In particular, a cross-comparison analysis, between in silico predictions of *miR-1304-5p* target genes and genes significantly downregulated (>20%) by a *miR-1304-5p* mimic was conducted to narrow down the list of potential targets of *miR-1304-5p*. Among these targets, 4 are effectors of the RAS/MAPK pathway (*NRAS, RRAS, PTPN11, IQGAP1*) and hence were validated as potential targets of *miR-1304-5p* in NB. PTPN11 preferentially dephosphorylates RAS proteins increasing their association with RAF ultimately activating the downstream MAPK cascade[69]; NRAS and RRAS interact with RAF proteins to activate the MAPK pathway with a preference over other kinase pathways such as PI3K[70,71]; IQGAP1 is a scaffold protein that enhances the interaction between MEKs (MAPK/ ERK kinase) and ERKs (extracellular signal-regulated kinases) and is indispensable for RAS-driven neoplastic cell survival[72,73]. Downregulation of these genes by *miR-1304-5p* was confirmed by Western blot and dual-luciferase assays, the latter of which identified a functional interaction between the 3'-UTRs of these genes and *miR-1304-5p*.

Due to the presence of 16 potential seed regions for direct targeting by *miR-1304-5p* in these 4 genes, site-directed mutagenesis to confirm the direct action of this miRNA on these targets was not possible in this study and remains to be determined, i.e., whether these genes are direct targets of *miR-1304-5p* or whether changes in their expression and activity are downstream effects. However, on over-expression of the 4 target genes individually and in combination with *miR-1304-5p* (Fig. 3, Supplementary Fig. 5), effects on the RAS/MAPK pathway are counteracted, which is supportive of these genes being direct targets of *miR-1304-5p*, although these data require further validation as detailed above.

NRAS was selected for further study since it was the only *miR-1304-5p* target identified in our study to negatively associate with NB patient survival (albeit across all NB genotypes rather than specific to *ALK*-aberrant NB) and ALK TKI resistance in in vivo models[33,34]. Additionally, *PTPN11* expression positively correlates with that of *NRAS* in NB patients. Notably, recent evidence regarding the use of a PTPN11 inhibitor in NRAS-mutated NB, showed increased survival and a decreased tumour volume in preclinical models; this evidence suggests that the RAS/MAPK pathway may be an effective therapeutic target for this malignancy[32]. In addition, NRAS activating mutations, amplification or upregulation have been implicated in resistance to

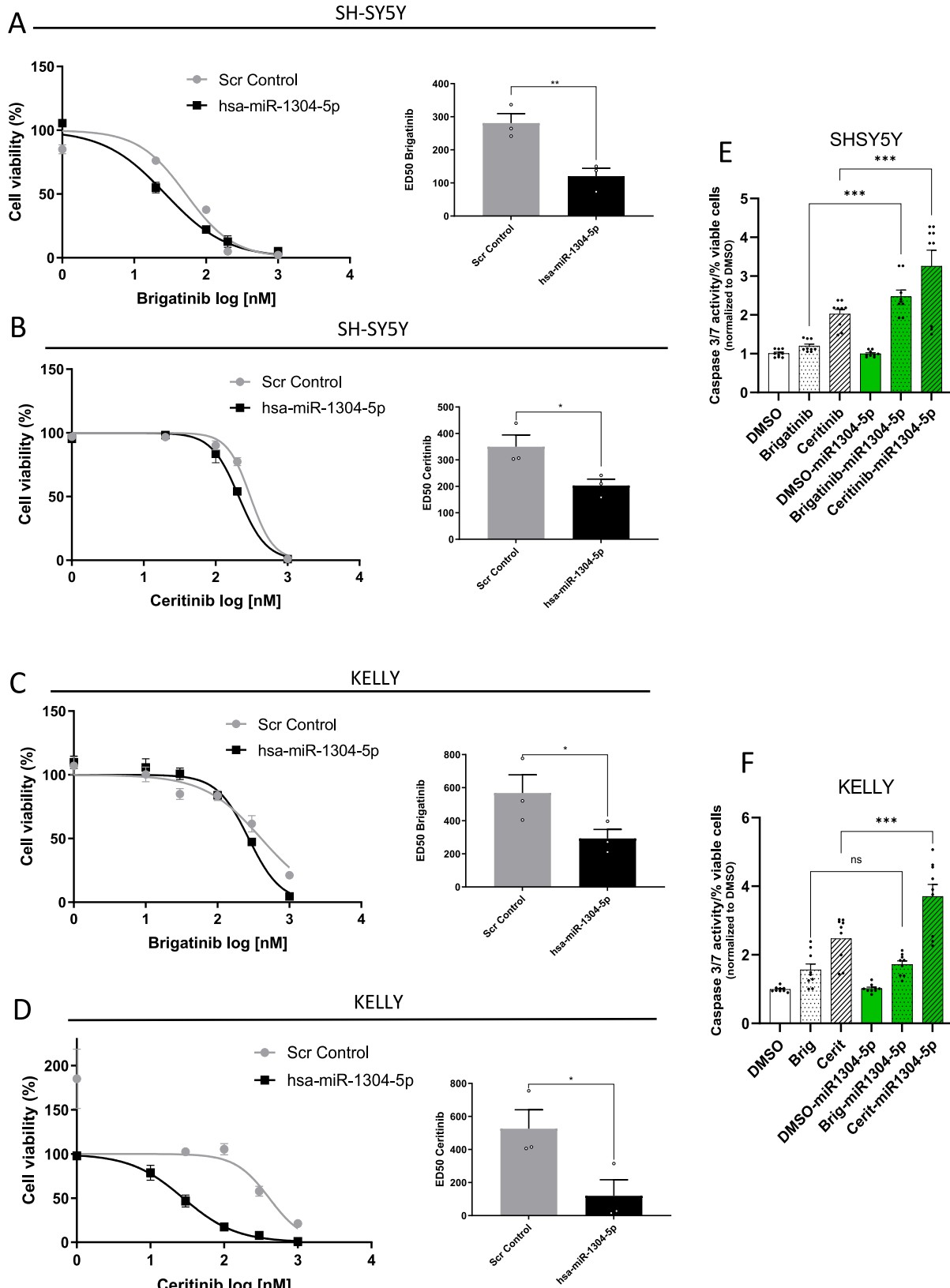

targeted therapies such as TKIs used in NB and other cancers, such as hepatocellular carcinoma (HCC), metastatic colorectal cancer and NSCLC[74–77].

As MAPK pathway inhibitors, in particular the MEK inhibitor Trametinib has previously been shown to have limited efficacy in *ALK*-aberrant NB due to activation of pro-survival feedback signaling through Akt/mTORc[37], we decided to investigate an inhibitor of NRAS activity, a protein upstream of both PI 3-Kinase and MEK[78]. In particular, we investigated the therapeutic efficacy of the FTI lonafarnib. Lonafarnib already has an orphan drug designation and has been approved for use in children with Hutchinson-Gilford progeria syndrome and for the treatment of certain processing-deficient progeroid

**Fig. 4 | The *miR-1304-5p* mimic is a therapeutic target when used in combination with ALK TKIs. A–D** SH-SY5Y (**A**, **B**) and KELLY (**C**, **D**) cell viability (measured via CTG) and $ED_{50}$ upon a combination of *miR-1304-5p* mimic transfection and brigatinib (SH-SY5Y (**A**) and KELLY (**C**)) or ceritinib (SH-SY5Y (**B**) and KELLY (**D**)) treatment for 72 h at the indicated doses. Graphs on the left show technical triplicates and are representative of three biological replicates. $ED_{50}$ values shown in the graphs on the right are calculated from the non-linear fit curves represented on the left and show the means ± SEM of biological triplicates. Statistical comparisons were conducted with a Student's *t* test of the three biological replicates.

**E, F** Apoptosis determined by caspase 3/7 activity in SH-SY5Y (**E**) and KELLY (**F**) cell lines treated with a combination of a *miR-1304-5p* mimic and brigatinib (1–1000 nM) or ceritinib (1–1000 nM). Data points ($n = 3$ technical replicates of each of three biological replicates) are shown with columns representing the means ± SEM of three biological replicates. Significance was determined using a one-way ANOVA with Tukey's post-test. **A** **$p = 0.0063$; **B** *$p = 0.045$; **C** *$p = 0.045$; **D** *$p = 0.027$; **E** ***$p = 0.00026$, ***$p = 0.00045$; **F** ***$0.00044$. Source data are provided in a Source Data file.

laminopathies in patients one year of age and older[35,36]. Indeed, we showed that targeting NRAS via either molecular knock-down or pharmacological inhibition with lonafarnib did not affect pAKT signalling neither in vitro nor in vivo (Supplementary Fig. 10A, B), suggesting that lonafarnib could overcome this feedback activation and result in a stronger anti-tumour effect.

In particular, our results show that lonafarnib acts synergistically with ALK TKIs in the treatment of NB both in vitro and in vivo particularly when *MYCN* is non-amplified. Since RAS inhibition could have a downstream effect on MYCN inhibition[79,80], it is reasonable to suggest that when *MYCN* is amplified, this effect could be partially reduced. *MYCN*-amplified NB could synergistically benefit from MYCN inhibition in addition to combination therapy targeting the RAS/MAPK pathway, as suggested in the literature[81,82]. Another study has shown an inverse correlation between MAPK signalling and MYCN status in response to targeted therapy (MEK and CDK4/6 inhibitors) in preclinical models of NB, suggesting that therapy targeting the MAPK pathway could have improved efficacy when *MYCN* is not amplified[83].

Overall, our study shows synergistic activity of an ALK TKI and lonafarnib. Notably this evidence was observed in 3 different PDX models, of differencing ALK and MYCN status. Additionally, none of the animals treated with a combination of ALK TKI and lonafarnib showed any signs of toxicity and all the animals in the combination treatment group were alive at the end of the study without any evidence of a growing tumour mass. Notably, our data suggest that the combination of ALK TKIs and FTIs is worthy of further investigation for the treatment of ALK wild type NB as well as to ensure tolerability of a lengthier treatment in ALK mutant NB models. Our in vivo *data* represent a significant finding and whilst later relapse was observed on cessation of therapy, the remission observed on treatment is of clinical significance.

## Methods

Our research complies with all relevant ethical regulations: Animal work was carried out under UK Home Office project licence number P4DBEFF63 according to the Animals (Scientific Procedures) Act 1986, was approved by the University of Cambridge Animal Welfare and Ethical Review Body (AWERB) and complied with all relevant ethical regulations for animal testing and research in the UK.

In all results as described in the main text and in the figure legends, biological replicates are intended as cell flasks cultured separately (a different passage of cells) for all in vitro experiments. Biological replicates are intended as separate animals in the in vivo studies.

### Cell lines and cell culture

The NB cell lines CHLA-15 (RRID:CVCL_6594), CHLA-20 (RRID:CVCL_6602), CHLA-90 (RRID:CVCL_6610), CHLA-95 (RRID:CVCL_6611), CHLA-171 (RRID:CVCL_6597), COG-N-426 (Felix) (RRID:CVCL_LF58), COG-N-415 (RRID:CVCL_AQ23), COG-N-557, LA-N-5 (RRID:CVCL_0389), LA-N-6 (RRID:CVCL_1363), NB-1643 (RRID:CVCL_5627), NB-EBC1 (RRID:CVCL_E218), SK-N-FI (RRID:CVCL_1702), and SMS-LHN (RRID:CVCL_9539) were obtained from the Children's Oncology Group Childhood Cancer Repository. CHP-134 (Cat#06122002), KELLY (Cat#92110411), LA-N-1 (Cat#

06041201) and SH-SY5Y (Cat# 94030304) were obtained from the European Collection of Authenticated Cell Cultures. GI-ME-N (ACC 654), NBL-S (ACC 656) and NGP (ACC 676) were obtained from German Collection of Microorganisms and Cell Cultures and 293FT (Cat#R70007) was obtained from Thermo Fisher Scientific.

CHLA-15, CHLA-20, CHLA-90, CHLA-95, CHLA-171, NB-1643, NB-EBC1, COG-N-426 (Felix), COG-N-415, COG-N-557, and NBL-S cells were cultured in IMDM (Gibco, #21980032) supplemented with 20% FBS, 1% insulin-transferrin-selenium (ITS; Gibco, Cat#41400045) and 1% penicillin and streptomycin (PS). CHP-134, GI-ME-N, KELLY, LA-N-1, LA-N-6, NGP, SK-N-FI and SMS-LHN cells were cultured in RPMI 1640 medium (Gibco, Cat#21875091) supplemented with 10% FBS, 1% ITS and 1% PS. SH-SY5Y and 293FT cells were cultured in DMEM (Gibco, Cat#41966029) supplemented with 10% FBS and 1% PS. Cells were grown at 37 °C in a humidified incubator with 5% $CO_2$. All cells were mycoplasma-free and subjected to quarterly in-house testing.

### Amplification of the GeCKO CRISPR screen library

The CRISPR screen library (Human CRISPR Knockout Pooled Library (GeCKO v2) (#1000000049; Addgene) was a gift from Dr. Feng Zhang (MIT)[27]. Each library (100 ng at a concentration of 50 ng/μl) was transformed into ElectroMAX Stbl4 Competent Cells (ThermoFisher) via electroporation with the Gene Pulser II Electroporation System (BioRad) (electroporator conditions: 1.2 kV, 25 μF, 200); this process was carried out with 4 replicate reactions. Cells were recovered in SOC media (NEB) and cultured for 1.5 h at 32 °C before plating onto ampicillin (100 μg/mL) Lysogeny broth (LB) agarose bioassay dishes (Nunc), which were incubated overnight at 32 °C. The bacterial colonies were then harvested and the CRISPR plasmid libraries were isolated using an EndoFree Plasmid Maxi Kit (Qiagen).

### CRISPR GeCKO screens; transduction of cells with the GeCKO library and exposure of cells to drugs

LentiCas9-Blast (Addgene; plasmid #52962) constructs were individually packaged into lentiviral particles in 293FT cells (Invitrogen) with second generation lentiviral packaging plasmids, psPax2 (Addgene; Plasmid #12260) and pMD2.G (Addgene; Plasmid #12259). The lentiviral construct to be packaged, psPax2, and pMD2.G were co-transfected (1:1:1) with TransIT-293 (MirusBio) in OptiMEM (Thermo-Fisher) into 293FT cells. Viral particles were harvested 60 hours after transfection. LentiCas9-Blast particles were applied to NB cells and stable Cas9 NB cells were selected in blasticidin (8–10 μg/ml; ThermoFisher). Stably expressing cells (SH-SY5Y Cas9-expressing NB cells; $10^6$ per well) were plated in a six-well plate (Corning) in 10% FBS/DMEM. After 24 h, virus libraries (in volumes ranging from 0 to 500 μl) were added to the cells for 24 h and cells were selected in 1 μg/mL of puromycin (ThermoFisher). The effective MOI was calculated after 3 days of selection, as the average cell counts from the duplicate wells following puromycin selection divided by the average cell counts from the duplicates with no selection reagent. The virus volume yielding a MOI nearest to the 0.3–0.4 range was selected leading to the use of $8 \times 10^6$ cells per 10 cm plate with a total of $40 \times 10$ cm plates ($320 \times 10^6$ cells) being transduced. Cells were selected in puromycin (1 μg/mL), and the effective MOI of the cells transduced for the screens was verified once again. SH-SY5Y cells ($320 \times 10^6$) were transduced as

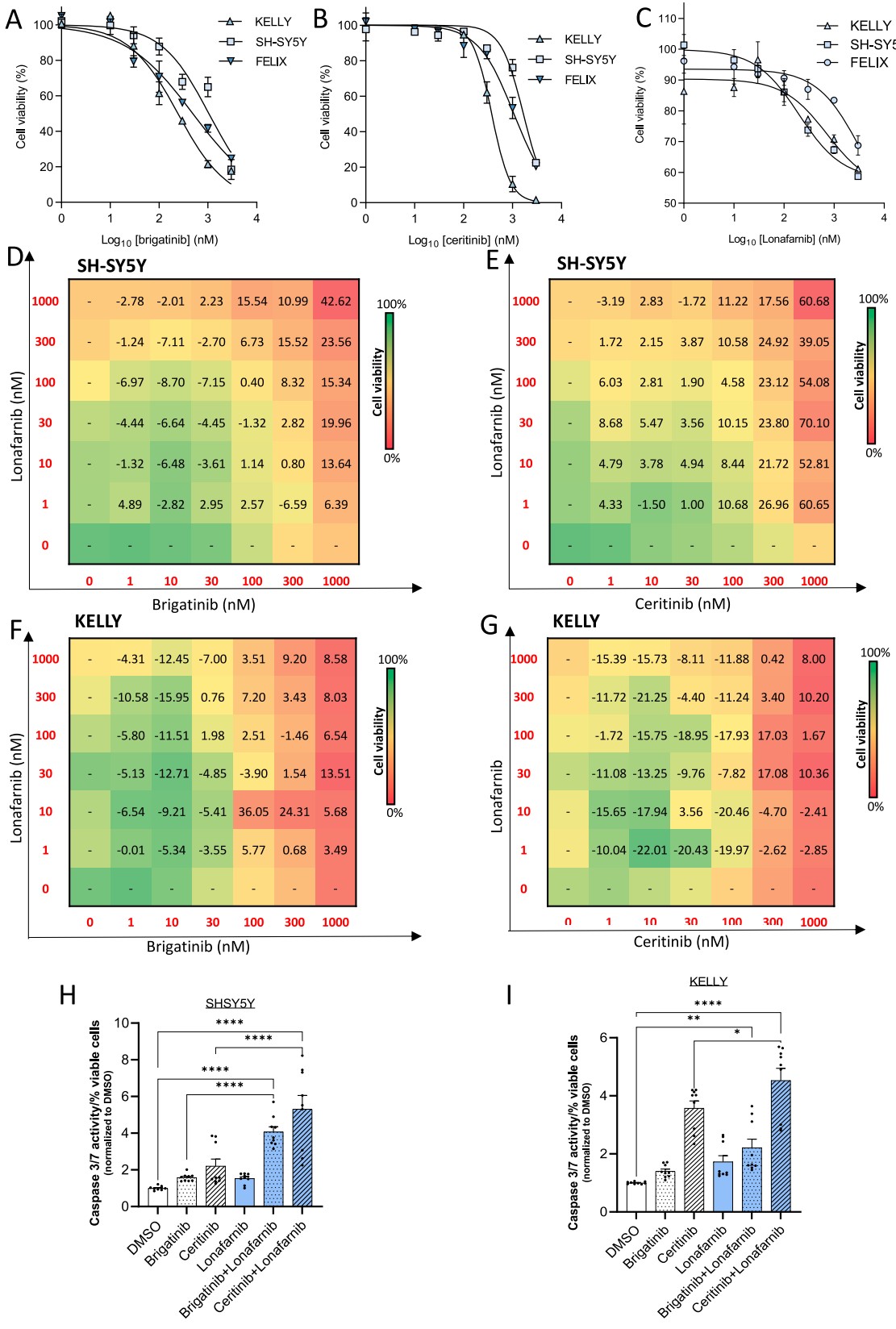

described above and puromycin (1 µg/mL) was added to the cells 24 h after transduction and incubated for a further 7 days. On day 7, cells were split into 5 different plates before drugs were added (4 TKI conditions plus DMSO vehicle control) in duplicates with a minimum of $35 \times 10^6$ cells per replicate (plated in $12 \times 15$ cm plates) with a separate $35 \times 10^6$ cells frozen down to be used as the Day 0 sample for

genomic DNA analysis. The 4 ALK TKI conditions were: 300 nM AP26113 (brigatinib), 750 nM AP26113 (brigatinib), 300 nM LDK-378 (ceritinib) and 750 nM LDK-378 (ceritinib). Cell pellets with a minimum of $35 \times 10^6$ cells were harvested 13 days after the addition of ALK TKIs, at which point the screens were terminated and the cell pellets were frozen at −80 °C until library preparation.

**Fig. 5 | ALK inhibitors and FTI act synergistically in ALK-aberrant MYCN non-amplified NB cell lines. A–C** SH-SY5Y, KELLY, and FELIX cell viability (CTG) upon treatment with brigatinib, ceritinib, or lonafarnib for 72 h. **D–G** Dose-response matrices of lonafarnib plus brigatinib or ceritinib in SH-SY5Y (**D, E**) and KELLY (**F, G**), treated for 72 h with the combination of agents or agents alone. The synergy scores and viability levels shown result from the average of two independent experiments. Loewe synergy scores were calculated with Synergy Finder and are shown in each square for each dose pair. Color gradients represent cell viability as a % compared to the DMSO vehicle control treated cells (from green 100% viability to red 0%). Synergy scores >10 represent synergism in activity of the inhibitors.

**H, I** Apoptosis determined by caspase 3/7 activity per cell population, normalized for the untreated (DMSO) control, in SH-SY5Y (**H**) and KELLY (**I**) cells treated with a combination of lonafarnib (1 μM) and brigatinib (1 μM) or ceritinib (1 μM), or each agent alone for 48 h. Data points (n = 3 technical replicates of each of three biological replicates) are shown with columns representing the means ± SEM of three biological replicates. Significance was determined using a one-way ANOVA with Tukey's post-test of the means of the three biological replicates.
**H** ****$p = 2.32 \times 10^{-6}$, ****$p = 6.58 \times 10^{-11}$, ****$p = 1.3 \times 10^{-5}$, ****$p = 2.6 \times 10^{-7}$;
**I** *$p = 0.0104$, ***$p = 0.0002$, ****$p = 4.5 \times 10^{-11}$. Source data are provided in a Source Data file.

## Preparation of HiSeq libraries

Frozen cell pellets were lysed and their gDNA was extracted using a QIAamp DNA Blood Maxi Kit as per the manufacturer's instructions (Qiagen, Cat#51194). PCR was conducted on the gDNA samples to amplify the sgRNA guide sequence region and to append the Illumina (HiSeq) compatible adapters and barcodes. A two-step PCR was performed: First, the amount of input gDNA for each sample was calculated in order to achieve 500X library coverage, which was calculated to be ~220 μg DNA per replicate. For each replicate sample, 22 separate 100 μl PCR reactions with 10 μg of input genomic DNA in each reaction with Herculase II Fusion DNA Polymerase (Agilent, Cat#600677) 1:50, distilled water, 5× Herculase II reaction buffer, dNTP mix, Primer #1 (10 μM) and Primer #2 (10 μM) were completed in an ABI Veriti Thermal Cycler (Applied Biosystems). The resulting amplicons from the first PCRs were then pooled together and mixed thoroughly to be used as the template for the second PCR reaction. Oligonucleotides (Sigma-Aldrich) used for the first PCR were:

Genomic PCR Forward 5'-AATGGACTATCATATGCTTACCGTAAC TTGAAAGTATTTCG-3'

Genomic PCR Reverse−5'-CTTTAGTTTGTATGTCTGTTGCTATTA TGTCTACTATTCTTTCC-3'.

The second PCR was designed to attach standard Illumina adaptors and unique barcodes to the amplicons from the first PCR to allow for HiSeq reactions and demultiplexing of the reads, respectively. The second PCR was conducted with the same reaction conditions but using 5 μl of the first PCR amplicon as the template, in 100 μl reactions with a total of 22 reactions. Primers for the second PCR were synthesized as IDT DNA Ultramer Oligonucleotides (Integrated DNA Technologies) and included a staggered region with variable lengths of sequence to enhance the library complexity.

## NGS analysis and GSEA of screen data

Analyses were performed with MaGECK-VISPR to ensure that the screens were adequately executed to allow for the acquisition of interpretable data. Quality control (QC) analysis of all raw fastq files derived from the screen samples was conducted at the sequence, read count, and sample levels. The sequence level QC was designed to identify potential technical issues associated with the sequencing process itself and was conducted with the FastQC script (Babraham Bioinformatics) that was embedded in MaGECK-VISPR. Sequence level analysis of GeCKO screen samples satisfied the general QC guidelines (Fig. 1B). Sample level QC was conducted to analyze the level of consistency between samples. Sample level QC determined the distributions of normalized sgRNA read counts, represented by a box plot and a cumulative distribution function (CDF) plot. It also involved plotting the samples on the first 3 components of a principle component analysis (PCA) (Fig. 1C). The PCA plots matched brigatinib and ceritinib-treated samples, potentially suggesting that these ALK TKI may induce similar resistance mechanisms. In addition, pairwise Pearson correlation analysis was performed using log-transformed sgRNA read counts of every sample in the GeCKO screen. The pairwise correlation analysis also matched samples from the brigatinib and ceritinib treated samples with values very close to 1

(Fig. 1D). MaGECK, a turn-key bioinformatics script designed specifically for CRISPR screens, was used as a maximum likelihood estimation approach[84]. More specifically, MaGECK-MLE does not rely on a rank-based analysis, allowing for more accurate and detailed comparisons of gene selections. This allows MaGECK-MLE to establish more refined clusters of genes that could not be revealed by previous methods. Hallmark and gene ontology gene expression datasets were downloaded from the molecular Signatures database (MSigDB) v6.2 and analyzed with GSEA v3.0 (www.broadinstitute. org/gsea).

## Transfection of miRNA mimics, hairpin inhibitors, and DsiRNAs

Cells were seeded in 96-well ($10^4$) or 6-well plates ($2 \times 10^5$) with a lipid:miRNA mixture prepared using the Lipofectamine RNAiMAX reagent (Invitrogen, Loughborough, UK), according to the manufacturer's protocol. The final miRNA mimic, inhibitor and DsiRNA concentration used was 40 nM. All mimics and hairpin inhibitors were purchased from Dharmacon miRIDIAN™ /Horizon (Suppl. Tab. 3). DsiRNAs were purchased from Integrated DNA Technologies, Inc. (IDT).

## Cell viability assays

Cells ($10^4$) were seeded into 96-well White Polystyrene Microplates (Corning, Cat# CLS3610) and after 24 h, the media was aspirated before one of the following procedures was carried out:

(a) Transfection with miRNA mimics, inhibitors or DsiRNAs (see corresponding methods section) for 48/72 h. (b) Transfection with miRNA mimics, inhibitors or DsiRNAs for 48 h and subsequent addition of 100 μL of fresh media containing drugs brigatinib (MedChem Express HY-12857) or ceritinib (MedChem Express HY-15656) in log-scale concentrations (0, 0.1, 1, 10, 30, 100, 300, 1000, 3000 nM) for 72 h. (c) Addition of 100 μL of fresh media containing drugs brigatinib (MedChem Express HY-12857), ceritinib (MedChem Express HY-15656), lonafarnib (MedChem Express HY-15136) or lorlatinib (MedChem Express HY-12215) in log-scale concentrations (0, 0.1, 1, 10, 30, 100, 300, 1000, 3000 nM) for 72 h.in log-scale concentrations (0, 1, 10, 30, 100, 300, 1000 nM) for 72 h.

The DMSO concentration was maintained at <0.06% for all conditions. For analysis of cell viability, 100 μL CellTiter-Glo reagent (Promega, Cat#C7571) was added to the cells, and cells were incubated at room temperature for 10 minutes. Luminescence was read on a SpectraMax i3 microplate reader (Molecular Devices).

## Synergy experiments

For dose–response matrices, cells were treated with log-scale concentrations of ALK TKIs as described above in addition to log-scale concentrations of lonafarnib, as indicated in 7 × 7 grids and the DMSO concentration was maintained at 0.02%. Potential synergy between ALK TKIs and lonafarnib was evaluated by calculating the synergy score based on the Loewe model[85] using synergy finder[86]. The synergy score is calculated for each combination of drug concentrations and also as an overall value and is defined as: >10 = synergistic; Between −10 and 10 = additive; <−10 = antagonistic.

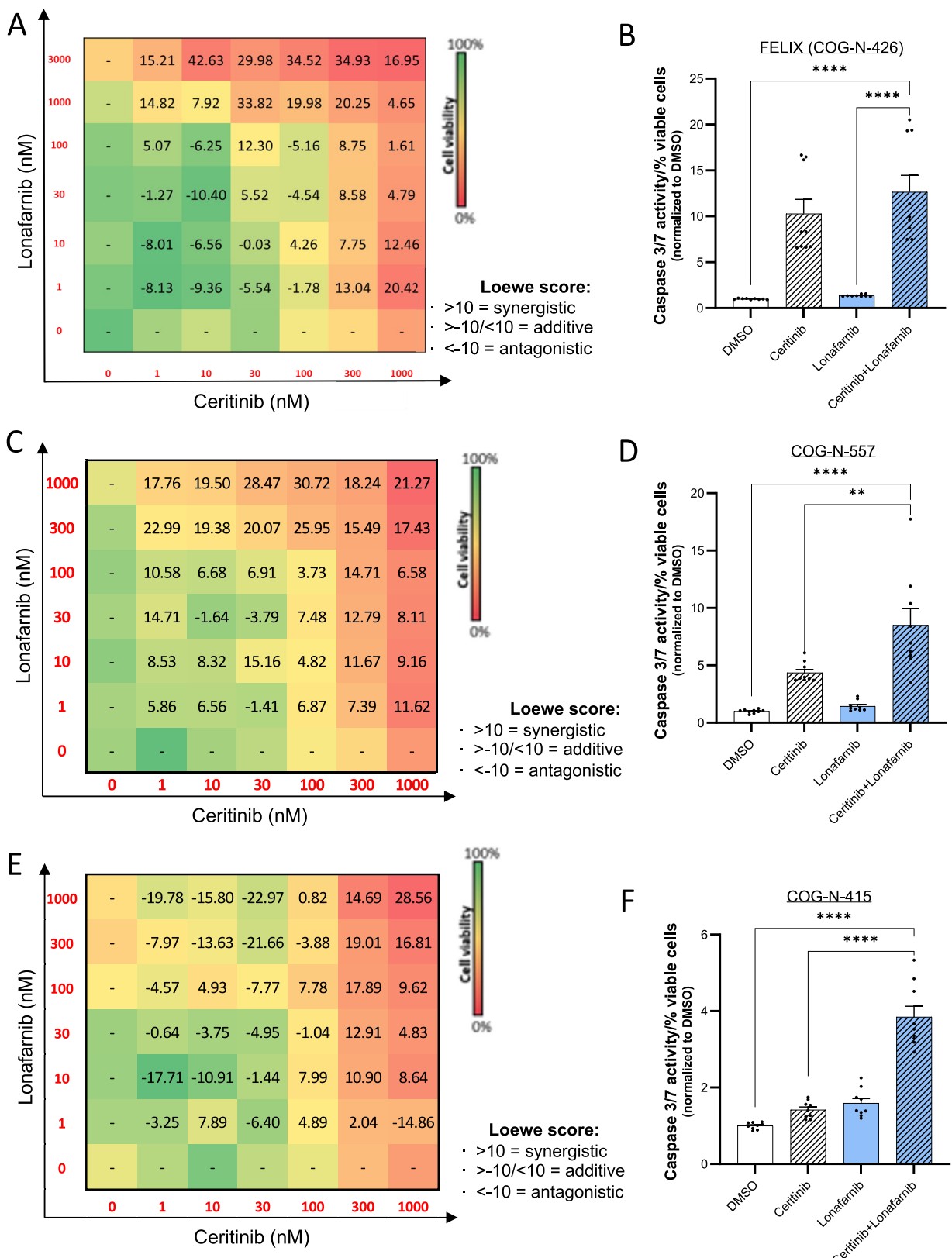

## Cell cycle analysis

Approximately 500,000 SH-SY5Y or KELLY cells were collected by centrifugation following trypsinization, washed with cold PBS, resuspended in 300 µl of cold PBS and fixed by the dropwise addition of 700 µL of 70% ice-cold ethanol (Sigma-Aldrich). Cells were fixed for 30 minutes on ice, then washed twice with ice-cold PBS before resuspending in 100 µg/mL RNAse (Sigma-Aldrich) for 30 minutes at 37 °C. Propidium iodide (PI; 50 µg/mL, Sigma-Aldrich) was added to the cells which were analyzed using a FACS Accuri™ C6 Plus Flow Cytometer (BD Biosciences). Single cells were gated using the FL2-area and -width parameters and a minimum of 10,000 events were collected per sample. Data analysis was conducted with FlowJo software (Treestar).

**Fig. 6 | A combination of an ALK inhibitor and an FTI act synergistically in PDX cell models via induction of apoptosis. A** Dose-response matrix of lonafarnib (1–3000 nM) and ceritinib (1–1000 nM) alone or in combination, following 72 h incubation with COG-N-426 (FELIX) PDX cells. Loewe synergy scores (Synergy Finder) and cell viability (CTG) result from two biological replicates. Color gradients: % cell viability normalised to DMSO (from green: 100%, to red: 0%). Scores >10 represent synergism. **B** Apoptosis (caspase 3/7 activity per cell population normalized to DMSO) of COG-N-426 cells treated with a combination of lonafarnib (1 μM) and ceritinib (1 μM), or single agents (same doses), for 48 h. Data points of biological replicates (each with three technical replicates) are shown with columns representing means ± SEM of biological triplicates. **C, E** Dose-response matrix of lonafarnib (1–1000 nM) and ceritinib (1–1000 nM) alone or in combination, following 72 h incubation in COG-N-557 (**C**) and COG-N-415 (**E**) PDX cells. Loewe

synergy scores (Synergy Finder) and cell viability (CTG) results are the average from two biological replicates, each conducted with two technical replicates. Color gradients: % cell viability normalised to DMSO (from green: 100%, to red: 0%). Scores >10 represent synergism. **D, F** Apoptosis (caspase 3/7 activity per cell population normalized to DMSO) of COG-N-557 (**D**) and COG-N-415 (**F**) cells treated with a combination of lonafarnib (1 μM) and ceritinib (1 μM), or single agents (same doses), for 72 h. Data points of biological replicates (each with three technical replicates) are shown with columns representing means ± SEM of biological triplicates. One-way ANOVA with Tukey's post-test of the means of biological triplicates have been used for statistical analysis. **B** ****$p = 3.5 \times 10^{-8}$, ****$p = 6.15 \times 10^{-5}$; **D** ****$p = 1.17 \times 10^{-6}$, **$p = 0.002$; **F** ****$p = 4.14 \times 10^{-7}$, ****$p = 5.48 \times 10^{-6}$. Source data are provided in a Source Data file.

## Apoptosis assays

SH-SY5Y or KELLY cells ($10^4$) were seeded in 96-Well White Polystyrene Microplates (Corning, Cat# CLS3610) and after 24 h, the media was aspirated and cells were transfected with 40 nM of a miRNA mimic, inhibitors or DsiRNAs (see corresponding methods section) for 72 h before 100 μL of Caspase 3/7-Glo apoptosis reagent (Promega, Cat#C7571) was added to each well and cells were incubated at room temperature for 1.5 h. Luminescence was read on a SpectraMax i3 microplate reader (Molecular Devices). Data were normalized to the scrambled non-targeting control (NC) and cell viability was assessed using GraphPad Prism 8 software (GraphPad Software Inc). Approximately 500,000 SH-SY5Y or KELLY cells were collected by centrifugation following trypsinisation and processed with the Annexin V APC kit (eBioscience™, Thermofisher, cat# 88-8007-74) according to the manufacturer's protocol. Briefly, cells were washed once in PBS, then resuspended in 100 μL Annexin V Binding Buffer containing 5 μL/reaction APC-Annexin V and incubated for 15 min at room temperature. Excess APC-Annexin V antibody was then removed from the cells following centrifugation, washed once in Annexin V Binding Buffer (Thermo Fisher Scientific, Cat#V13246), and cells resuspended in 200 μL Annexin V Binding Buffer containing 1 mg/mL propidium iodide (Sigma-Aldrich, Cat#P4170-10MG). Data were acquired using a Fortessa flow cytometer and analyzed using FlowJo V10 software. Cells were gated according to physical parameters in order to discard cell debris (FCS/SSC) and cell clumps (width/area).

## RNA and miRNA extraction and RT-qPCR analysis

Total RNA was isolated from cell lines using the RNeasy Plus Mini Kit (Qiagen, Cat#74134). In total, 1 μg RNA was reverse transcribed using iScript Reverse Transcription Supermix (Bio-Rad, Cat#1708840) and 20 ng cDNA template was used for qPCR. cDNA was amplified with PowerUp SYBR Green Master Mix (Thermo Fisher Scientific, Cat#A25918) and 600 nM primers on a QuantStudio 6 Flex Real-Time PCR System (Thermo Fisher Scientific) under standard cycling conditions. Relative quantification (ΔΔCT) analysis was conducted with normalization to GAPDH. All reactions were performed in technical triplicates. In the case of analysis of NRAS expression the High-Capacity cDNA Reverse Transcription Kit (Thermo Fisher Scientific cat# 4368814) was used followed by the TaqMan RT-qPCR assay (NRAS, Hs00180035m1, cat3 4331182, ThermoFisher). For isolation of miRNA, miRNeasy mini (Qiagen, cat4 217004) and PureLink miRNA Isolation kits were according to the manufacturer's instructions. *miR-1304-5p* expression was analyzed using TaqMan RT-qPCR with the TaqMan™ MicroRNA Reverse Transcription Kit (Applied Biosystems, cat2 4366596) and the *hsa-miR-1304-5p*–TaqMan® MicroRNA Assay (cat2 4427975, ThermoFisher). Relative quantification (ΔΔ$C_T$ method) analysis was conducted with normalization to *GAPDH* (Thermo Fisher Scientific, Hs02786624g1, 4331182), *HPRT1* (Thermo Fisher Scientific, Hs02800695_m1, 4331182), *U6* snRNA (TaqMan® MicroRNA Assays, cat# 4427975) and *RNU24* (Thermo Fisher Scientific, cat# 001001) as indicated.

## HT-12 v4 genome-wide expression microarray

A *Hsa-miR-1304-5p* mimic (Qiagen) or the AllStars Negative Control siRNA was transfected at 40 nM with Lipofectamine RNAiMAX (1.5 μL each 96 well and incubated for 5 minutes at room temperature, before dropwise addition onto the cells) (ThermoFisher) into SH-SY5Y and KELLY cells. Total RNA was extracted with the RNeasy Plus Mini Kit (Qiagen, Cat#74134) and applied to Illumina Human HT-12v4 expression arrays (Eurofins Genomics, Germany). The microarray readouts were re-formatted to Gene Cluster Text (GCT) file format and a Categorical class (CLS) file was generated (http://software.broadinstitute.org/cancer/software/gsea/wiki/index.php/Data_formats). Both files were used as input for GSEA[28] on quantile normalized mRNA read counts from each cell line with the MSigDB's 'hallmark gene sets'[29] on the javaGSEA Desktop Application to compare *hsa-miR-1304-5p* mimic (Qiagen) versus Negative Control gene expression profiles of SH-SY5Y and KELLY cell lines.

## Immunoblot analysis

Adherent cells were washed, and lysates prepared with pre-chilled RIPA Lysis and Extraction Buffer (Thermo Fisher Scientific, Cat#89900) supplemented with 1–2% Halt Protease and Phosphatase Inhibitor Cocktail (Thermo Fisher Scientific, Cat#78440). Around 50 μg protein lysate per sample was resolved by SDS-PAGE and transferred to a 0.45 μm PVDF membrane. Primary antibodies used were as follows: anti-IQGAP1 (1:200; Santa Cruz Biotechnology (SCBT), cat# sc-376021 or 1:2000; Proteintech, cat# 22167-1-AP), PTPN11 (1:200; SH-PTP2 B-1, SCBT, cat# sc-7384 or 1:1000; Proteintech, cat#.20145-1-AP), anti-p42/44 MAP Kinase (1:1000; Cell Signalling Technology (CST), cat# 9102), anti-pan-Ras (1:200; SCBT, cat# sc-166691), anti-NRAS (1:500; Proteintech, cat# 10724-1-AP), anti-RRAS (1:500; Proteintech, cat# 27457-1-AP), anti-NF1 (1:500; Proteintech, cat# 27249-1-AP), anti-phospho-p42/44 MAP Kinase (Thr202/Tyr204) (1:1000; CST, cat# 9101 S), anti-phospho-AKT (1:1000; CST, cat# 9271), vinculin (1:200; SCBT, Cat# sc-73614), anti-PARP (1:1000; CST, cat# 9542), anti-cleaved PARP (1:1000; CST, cat#9541) and anti-β-Tubulin (1:1000; CST, cat#2146).

## 3′-UTR dual-luciferase assays

The 3′-UTR of the genes of interest were amplified by PCR from SH-SY5Y cDNA, using the ProtoScript II First Strand cDNA Synthesis Kit (NEB). The segments of the 3′-UTRs with the predicted miRNA target sites were PCR amplified with Q5 High-Fidelity PCR Kit (NEB) using the specific oligonucleotides. psiCHECK-2 and the amplicons were digested with XhoI (NEB) and NotI-HF (NEB) restriction enzymes at 37 °C overnight and ligated with T4 DNA ligase (NEB). The ligated plasmids were then transformed in NEB STABLE cells (NEB) and plasmids isolated with QIAprep Spin Miniprep Kit (Qiagen). To generate the positive control plasmid, an oligonucleotide that is complementary to the *hsa-miR-1304-5p* sequence (5′- UUUGAGGCUACAGUGAGAUGUG -3′) was annealed and cloned into the psiCHECK-2 vector (Promega) vector:

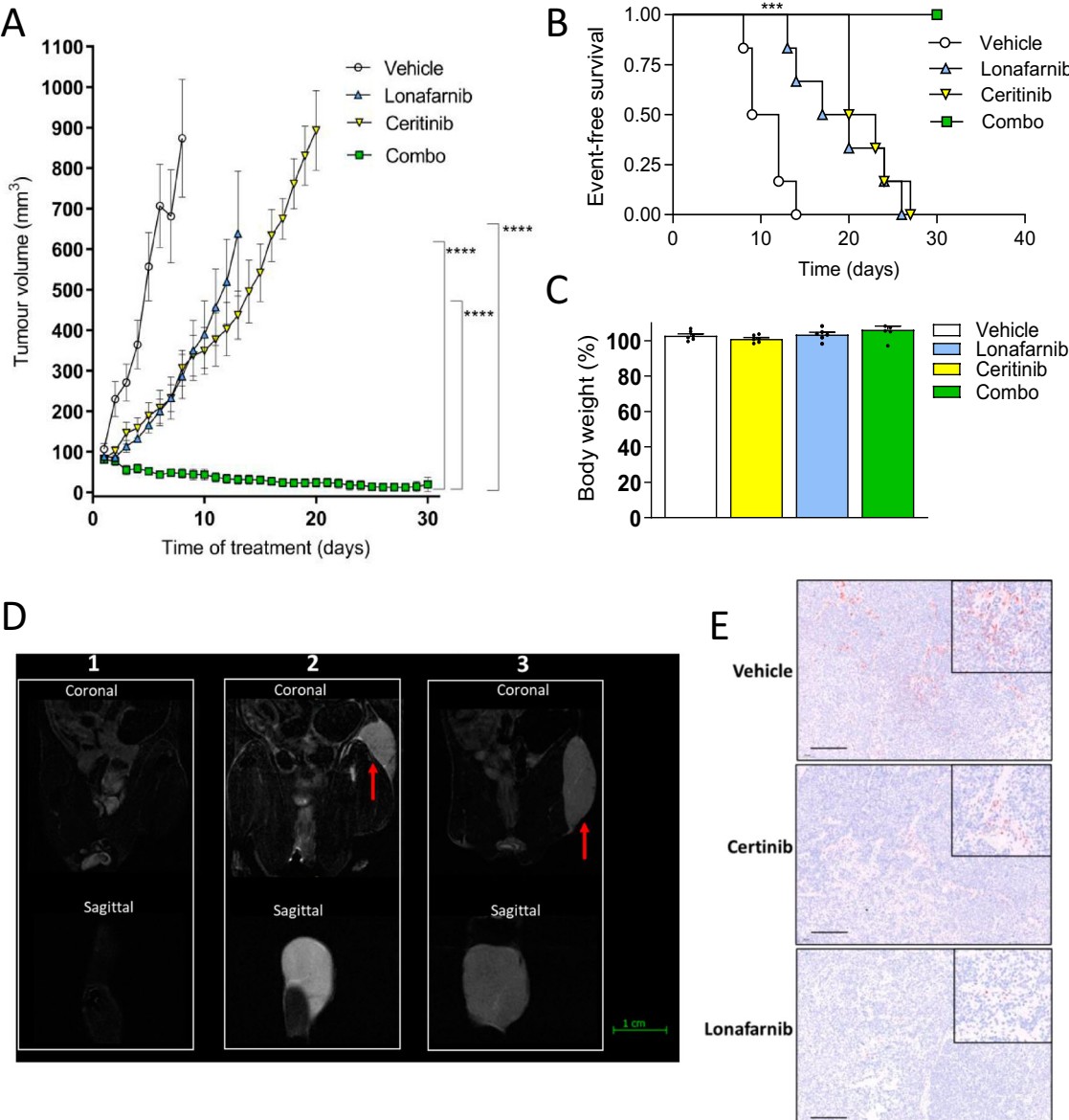

**Fig. 7 | A combination of an ALK inhibitor (ceritinib) with a FTI (lonafarnib) significantly reduces PDX tumour growth in vivo. A** Tumour volume over time of NSG mice injected sub-cutaneously with COG-N-426x primary NB cells which reached 75 mm³ before daily administration of either vehicle (20% hydroxypropyl beta cyclodextrin), ceritinib (30 mg/kg), lonafarnib (40 mg/kg), or ceritinib and lonafarnib (combo, same doses). The study endpoint is tumours reaching 15 mm diameter or following 30 days of treatment, whichever came first. Data shown represent means ± SEM from 6 mice (*n* = 6) at each time point. The one-way ANOVA with Tukey's post-test was used to determine significance at each experimental endpoint in **A**. ****$p$ = 1.22 × 10⁻¹³, ****$p$ = 3.77 × 10⁻⁸, ****$p$ < 10⁻¹⁵. **B** Kaplan–Meier EFS analysis. Data points (*n* = 6) represent means ± SEM, shown until the experimental endpoint (as defined above) of the first animal within each treatment group, ***$p$ = 0.0007 (Log-rank test). **C** Mouse body weight at the experimental endpoint relative to baseline weights for each treatment group. Data points (*n* = 6) are shown with means ± SEM. The one-way ANOVA with Tukey's post-test was used to determine significance at each experimental endpoint in **C** and significance was not reached in any case. **D** MRI scan of coronal and sagittal sections of animals treated with a combination of lonafarnib and ceritinib at the end of treatment (left panel) compared to relapses after 9 (central panel) and 10 (right panel) days following the end of treatment. Red arrows = tumour mass; green bar = 1 cm. **E** Immunohistochemistry (pERK) in tumours from mice treated with vehicle (20% hydroxypropyl-beta-cyclodextrin), ceritinib (30 mg/kg) or lonafarnib (40 mg/kg) at the study endpoint. Images are representative of three independent replicates with similar results. Magnification bar = 100 μm. Inserts = ×200 magnification. Source data are provided in a Source Data file.

psiUTR-pos_F 5′-TCGACACATCTCACTGTAGCCTCAAAGT-3′
psiUTR-pos_R 5′-GGCCACTTTGAGGCTACAGTGAGATGTGT-3′.

Similarly, to generate the negative (scrambled; scr) control plasmid, the positive control oligonucleotide sequence was scrambled (http://www.invivogen.com/sirnawizard/scrambled.php) to achieve least base pairing to the miRNA while maintaining the overall base composition:

psiUTR-scr_F 5′-TCGAACAACTATCCGTCCATAATCCGGT-3′
psiUTR-scr_R 5′-GGCCACCGGATTATGGACGGATAGTTGT-3′.

The isolated plasmids were transfected into 293FT cells with TransIT-293 (MirusBio) reagent in 24-well plates at 50% confluence. The next day, cells were transfected with *hsa-miR-1304-5p* mimics (Qiagen) or AllStar Negative Control siRNA (Qiagen). Following transfection of the miRNA (at 48 h), firefly and renilla luciferase activities were assayed with the Dual-Glo Luciferase Assay System (Promega) by adding 100 μL/well of a 96-well plate well, incubating at room temperature for 10 minutes and reading using a SpectraMax i3 (Molecular Devices).

## Patient derived xenograft studies

NSG mice were obtained from Charles River and housed in groups of 2–6 mice per cage in individually ventilated cages with a 12 light/dark cycle. All procedures were carried out under UK Home Office licence P4DBEFF63 according to the Animals (Scientific Procedures) Act 1986 and were approved by the University of Cambridge Animal Welfare and Ethical Review Board (AWERB). COG-N-426x and COG-N-415x patient-derived xenograft (PDX) cells were obtained from the Childhood Cancer Repository maintained by the Children's Oncology Group (COG). Cells were suspended in Matrigel (Corning) diluted 1:2 with PBS and $5 \times 10^5$ cells (300 μL) were injected into the left flank of NSG mice at 8 weeks of age. Tumours were measured daily with manual callipers and tumour volumes estimated using the modified ellipsoid formula: $V = ab^2/2$, where a and b ($a > b$) are length and width measurements respectively. Once tumours reached ~75 mm³, mice were randomly allocated into four treatment groups ($n = 6$–8 per group, with the same number of females and males in each study) and treated daily with the following agents by oral gavage at 10 μL/g body weight: vehicle (20% hydroxypropyl-beta cyclodextrin), ceritinib (30 mg/kg in vehicle), lonafarnib (40 mg/kg in vehicle) or a combination of ceritinib and lonafarnib at the same doses. Mice were euthanized once tumours reached 15 mm in any direction (defined as an event for EFS analysis). The maximal tumour size permitted by our Project Licence (20 mm) was not exceeded in any of the studies.

## Immunohistochemistry (IHC)

IHC was performed as reported previously[5]. Briefly, tumours were fixed in 10% neutral-buffered formalin for 48 h before paraffin-embedding. Tissue sections were stained with hematoxylin and eosin or with antibodies against ALK (CST, Cat#3633) with a dilution of 1:500 and incubation overnight at 4 °C, pERK (CST, Cat#4370) with a dilution of 1:300 and incubation overnight at 4 °C, pAKT (CST, Cat#3787) with a dilution: 1:50 and incubation overnight at 4 °C. Heat antigen retrieval was conducted with a citrate buffer ph6. The signal was developed under visual control with 3-Amino-9-ethylcarbazole.

## Patient sample analysis

CBioPortal (www.cbioportal.org) was queried using a publicly available TARGET study (TARGET, 2018, phs000218 (https://ocg.cancer.gov/programs/target) available at https://portal.gdc.cancer.gov/projects." Study ID phs000467). This dataset consists of 1089 samples from 1076 patients. Among them, 143 samples with bulk RNA seq RPKM data, 249 with Agilent microarray and 59 with copy number alterations are available. We have also analysed the SEQ and Kocak datasets from the platform R2. By default, the R2 platform "scans" a range of cutoff values, applying a log-rank test to each, and selects the cut-off point which yields the minimum $P$ value. Using this methodology, type I error is inflated due to hundreds of comparisons, necessitating multiple-testing correction. In our analyses, we instead manually chose an NRAS expression value cutoff, based on the expression distribution of the cohort, as represented in Extended Data Fig. 5. This was done because the population distribution appears bimodal.

## Magnetic resonance imaging

Magnetic resonance imaging (MRI) data was acquired using a 3 T BioSpec Bruker system (Ettlingen, Germany) with a 40 mm quadrature volume coil. Animals were anaesthetised with isoflurane (induction 3%, maintenance 2%) in 100% oxygen, adjusted thereafter to normalise respiration rate, which was maintained at 40–60 breaths per minute using a pneumatic pillow (ERT Control Gating Module, SA Instruments, New York, United States). Animals were positioned in an MRI compatible cradle fitted with a heated air supply (Thermo Fisher Scientific, Massachusetts, United States), allowing the temperature to be maintained at 35-36 °C using a rectal probe. Anatomical 3D FISP sagittal images were acquired with a scan time of 4 m 28 s, with the following parameters: field-of-view (FOV) = $30 \times 30 \times 30$ mm³, matrix = $128 \times 128 \times 64$, TR = 6.5 ms, TE = 3 ms, bandwidth = 37 kHz, averages = 3, with a flip angle of 15 degrees and eight segments. For tumoral contrast, sagittal 3D T2-weighted images were acquired with turboRARE and the following parameters: TE = 84 ms, TR = 1200 ms, averages = 1, scan time = 7 m 40 s, echo spacing 12 ms, RARE factor = 16, FOV = $30 \times 30 \times 30$ mm³, matrix size = $100 \times 100 \times 64$, bandwidth = 20.8 kHz, with fat suppression. Following image acquisition animals were placed in a recovery box on a heated pad set to 37 °C. Images were viewed in ParaVision 360 V2.0 with the T2-weighted image (green) overlaid onto the anatomical 3D FISP image (grey), which then allowed for 3D visualisation and high contrast.

## Statistical analysis

All Student's $t$ tests, one- and two-way ANOVA models, correlation analyses and Kaplan–Meier survival analyses were conducted with GraphPad Prism 8/9 software.

## Reporting summary

Further information on research design is available in the Nature Portfolio Reporting Summary linked to this article.

## Data availability

All data are available in the main text or the supplementary materials. PDX models were provided under an MTA from the Childhood Cancer Repository maintained by the Children's Oncology Group (COG). The raw sequencing data generated in this study have been deposited in the NCBI, SRA database under accession code PRJNA903183 [BioSample (24) SRA (24)]. The normalized read counts from the CRISPR GeCKO, the microarray Gene Analysis in SHSY5Y, the microarray Gene Analysis in KELLY, and the GSEA[28] data generated in this study are provided in the Supplementary Data files 1–4, respectively. The remaining data are available within the Article, Supplementary Information, or Source Data file. Source data are provided with this paper.

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

## Acknowledgements

This research was supported by the Cambridge NIHR BRC Cell Phenotyping Hub. J.K. acknowledges support from the NIHR Cambridge Biomedical Research Centre. We would like to acknowledge the Childhood Cancer Repository, which provided the neuroblastoma PDX (Texas Tech University Health Sciences Center School of Medicine, Lubbock, TX 79430, USA) and is funded by Alex's Lemonade Stand Foundation for Childhood Cancer. Funding for this research was awarded to S.D.T. by Neuroblastoma UK (grant number: NBUKTurner19), and Children with Cancer UK (grant number: 16-209), which supported P.P. and R.M.T. L.C.L. and L.H. were supported with funding from the Cancer Research UK Cambridge center postgraduate training program (grant number: C9685/A25117). N.P., S.P.D, S.D.T, O.M., M.H., L.J., and L.K. received funding from the European Union's Horizon 2020 Marie Skłodowska-Curie Innovative Training Networks (ITN-ETN) ALKATRAS under grant agreement no.: 675712. J.M. was supported with funding from the Alex Hulme Foundation (grant number AHF02). S.D.T. is supported by the project National Institute for Cancer Research (Programme EXCELES, ID Project No. LX22NPO5102)—Funded by the European Union—Next Generation EU. This work was supported by the Cancer Research UK

Cambridge Centre [CTRQQR-2021\100012] and the NIHR Cambridge Biomedical Research Centre (NIHR203312). The views expressed are those of the authors and not necessarily those of the NIHR or the Department of Health and Social Care.

## Author contributions

Conceptualization: P.P., L.C.L., R.M.T., G.A.A.B., S.D.T. Methodology: P.P., L.C.L., R.M.T., J.L., S.D.T. Investigation: P.P., L.C.L., E.M., R.M.T., S.P.D., L.J., M.H., J.D.M., J.K., A.S.-A., N.P., C.S., L.H., E.R.J. Funding acquisition: L.K., O.M., S.D.T. Resources: F.R., S.T.M. Supervision: S.D.T. Writing: P.P., S.D.T. All authors approved the final version of the manuscript.

## Competing interests

G.A.A.B. has received institutional consultancy fees from Roche, Takeda, Novartis, and Janssen. All other authors have no conflicts of interest to declare.

## Additional information

[1]Department of Pathology, Division of Cellular and Molecular Pathology, University of Cambridge, Cambridge CB20QQ, UK. [2]Department of Life Sciences, Birmingham City University, Birmingham, UK. [3]Nottingham Trent University, School of Science & Technology, Clifton Lane, Nottingham NG11 8NS, UK. [4]Department of Pathology, Division of Experimental and Translational Pathology, Medical University of Vienna, 1090 Vienna, Austria. [5]MRC Mitochondrial Biology Unit, University of Cambridge, The Keith Peters Building, Cambridge Biomedical Campus, Hills Road, Cambridge CB2 0XY, UK. [6]Department of Medicine, University of Cambridge, Addenbrookes Hospital, Hills Road, Cambridge CB2 0QQ, UK. [7]Department of Radiology, University of Cambridge, Cambridge Biomedical Campus, Cambridge CB2 0QQ, UK. [8]Department of Paediatric Haematology, Oncology and Palliative Care, Addenbrooke's Hospital, Cambridge CB2 0QQ, UK. [9]Department of Pathology, Medical University of Vienna, Vienna 1090, Austria. [10]European Research Initiative for ALK related malignancies (ERIA), Cambridge CB2 0QQ, UK. [11]St. Anna Children's Cancer Research Institute, CCRI, Zimmermannplatz 10, 1090 Vienna, Austria. [12]Laboratory of Cancer Biology and Genetics, Center for Cancer Research, National Cancer Institute, National Institutes of Health, Bethesda, MD 20814, USA. [13]Unit of Laboratory Animal Pathology, University of Veterinary Medicine Vienna, Vienna, Austria. [14]Center for Biomarker Research in Medicine (CBmed), Graz, Austria. [15]Christian Doppler Laboratory for Applied Metabolomics (CDL-AM), Medical University of Vienna, Vienna, Austria. [16]Faculty of Medicine, Masaryk University, Brno, Czech Republic. [17]Present address: Merck & Co, 2000 Galloping Hill Rd, Kenilworth, NJ 07033, USA. [18]Present address: OncoSec, San Diego, CA 92121, USA. [19]Present address: Chelsea and Westminster Hospital, NHS Foundation Trust, London SW10 9NH, UK. [20]Present address: Functional Genomics, GlaxoSmithKline, Stevenage SG1 2NY, UK. ✉e-mail: sdt36@cam.ac.uk

