## [Peer Review File · Nature Communications]

Reviewers' Comments:

Reviewer #1:

Remarks to the Author:

This is a very interesting manuscript addressing an important clinical need which is to decipher mechanism of drug resistant in highly-aggressive tumors such as high-risk NB. The authors use a large panel of NB cell lines and clinically-representative in vivo models.

While the study is logically designed and experimentally-well executed, some of the proposed mechanisms are explored very superficially and need additional controls to strengthen their conclusions.

Below are indicated some concerns that should be addressed before acceptance for publication:

- 1.- In figure 2A-D, the levels of mature miRNA after miRNA inhibition must be shown. The fact that only one of the miRNAs has an effect on cell proliferation questions whether the other miRNAs have been successfully inhibited. A table showing the genomic location of these miRNAs and whether they cluster with other miRNAs or belong to the same family would be very informative.
- 2.-Figure 3A-B are insufficiently detailed. A table, that may be included in extended data, showing all hallmarks analyzed with their NES score and associated p-value of the performed GSEA must be included.
- 3.-Figure 3C seems to be saturated and only performed in one cell line. Same analyses in a second cell line must be included. Why a pan-RAS is used instead of an NRAS antibody?. Please, explain. Apparently, after overexpressing miR-1304-5p the total levels of ERK1/2 are also downregulated. Are these proteins also a potential target of miR-1304-5p? Since one of the main messages of the paper is that the downregulation of NRAS is responsible for miR-1304-5p effects, it's important to dissect at which levels the pathway is affected.
- 4.-The 3'UTR analysis of Figure 3D is incomplete. To truly claim that these are direct targets, 3'UTR sites must be mutated and see if the reduction on luciferase activity is abolished.
- 5.-Figure 3E is also incomplete. The levels of the targets (i.e NRAS, RRAS, etc..) should be shown after overexpression and with +/- miR-1304-5p overexpression to correlate the effects on the cell viability. Furthermore, monitoring MAPK activity through phospho-ERK would help to see if NRAS and others is sufficient to counteract the effects of miR-1304-5p on the RAS/MAPK pathway.
- 6.-Figure 3F is also incomplete. The correlation of NRAS with neuroblastoma outcome is potentially interesting, but the authors did only a very preliminary analyses. Is this correlation maintained in other data sets with more samples (e.g. Kocak, SEQN)?. A more thorough analyses indicating correlation with other clinical variables (age, sex, stage, MYCN-amplification, genomic aberrations) will be more informative and may help to identify patient populations that would be more susceptible to the proposed therapy.
- 7.-Data characterizing miR-1304-5p is insufficient. Caspase-3/7 DVD-cleavage assay is only partially informative and more analyses should be confirmed to understand the effect. In addition of using a method that directly quantifies number/percentage of apoptotic cells (e.g. DAPI staining, annexin-V staining), a western-blot showing the processing of caspase substrate such as PARP or fodrin will also confirm the apoptotic execution program. Furthermore, whether induction of apoptosis is the reason of decreased cell number will also have to be confirmed in presence of apoptosis inhibitors. Cell Cycle analysis is also partial. FACS plots must be shown and at different time points to confirm if any change in cell cycle precludes the induction of apoptosis.
- 8.-Figure 4 is incomplete. Caspase 3/7 analysis have to be complemented with other apoptotic assays, particularly offering information about the fraction of cells affected. It is also important to see if additive/synergic effect are also observed at the molecular level (e.g. levels of cleaved caspase and caspase substrates).
- 9.-In figure 5 and 6, caspase 3/7 assays should also be complemented as indicated above.
10. In figure 7C, mouse weight variation should be represented during the whole course of treatment. It has no sense to do it at a single time point. On-target effect should be better demonstrated. Analyzing Phospho-ERK in a single tumor section from a single mice is not sufficient. Better include western-blot analyses of samples and/or quantify % of P-ERK positive cells in multiple sections from 3-4 mice per group. Same applies to other immunohistochemistry analyses of the paper such as those shown in Extended Figure 9 and 10.
11. Figure 7B is misleading. The chosen endpoint may confuse the results of the study. Whether 15mm diameter of tumor may be adequate and ethically recommendable, 30 days of treatment it's insufficient to estimate the efficacy of the treatment. A 20-30% time extension from the last

mouse sacrifice of the single-treatments arms would help to determine if the combination is more effective than the single treatments alone. So, in this case, a minimum of 35-40 days of follow-up would be the right choice.

Reviewer #2:

Remarks to the Author:

Authors use a high-throughput, genome-wide CRISPR-Cas9 knockout screens to identify miR-1304-5p loss as a desensitizer to ALK TKIs in both ALK wt and ALK mutant NB. They go onto that the FTI lonafarnib in addition to ALK TKIs cooperate in NB, inducing apoptosis in vitro. Combined treatment of an ALK mt NB patient derived xenograft with an FTI and an ALK TKI complete regression of tumour growth was observed although tumours rapidly regrew on cessation of therapy.

1. Introduction and discussion reiterates abstract, perhaps redundant

2. Can Figs 2E and 2F use some sort of marker or color coding to distinguish ALK wt from ALK mt lines? In 2E, why do ALK wt lines show robust responses to the miRNA inhibitor? Only NGP is flat, and many ALK mutant lines show responses less robust than those seen in ALK wt lines? Can the authors include an assay for inhibitors that show these inhibitors inhibit their appropriate targets as anticipated?

3. In Fig 3F, was the group split in half (the most accurate way to do this), or was the group split to maximize the p-value? If the latter, please show data for the 143 patients split in half.

4. Can experiments in Fig 4 be extended to include ALK wt lines? In Figs 4-6, does combination therapy cooperate to block signaling downstream of NRAS?

5. Generalizability of results requires extending in vivo experiments beyond a single pdx line. Authors claim that the combination therapy works in vivo to induce apoptosis, did I miss these data? If I did not, can these data be added?

Reviewer #3:

Remarks to the Author:

The authors have conducted a CRISPR screen to identify genetic factors that influence sensitivity to ALK inhibition in neuroblastoma (NB). To this end, the NB SY5Y cell line was selected and surviving clones under exposure to two ALK inhibitors were identified to allow selection of genes which render resistance to ALK inhibition. This revealed both protein and non-coding genes. In a further selection step, one single miRNA across all four experiments and four miRNAs for three out of four screens were identified. The authors test these four miRNAs further and identify miR-1304-5p for subsequent functional studies. For this miRNA a more global TSG function is suggested in NB and NRAS is identified as regulated target. Next, NRAS is identified as target regulated by miR-1304-5p. Finally, in a series of in vitro and in vivo experiments, the combination of ALK inhibition with NRAS inhibition using the farnesyl transferase inhibitor Lonafarnib is shown to yield additive or synergistic effects, although upon interruption of treatment in mouse xenografts tumors do recur.

Major comments:

The major concern for this paper is that previous work has been reported also using a CRISPR screen under control and ALK inhibition conditions already pointing in a more direct and more convincing way the critical importance of NRAS activation (Berlak et al., 2022) and thus also strongly reducing the novelty of this paper. This work, although referred together with other papers, is largely neglected in putting the presented data into context. While CRISPR screen like any screen will pick up and miss targets, it is troublesome that in the present screen NF1 was not

identified as target, in particular given that the same cell line SY5Y was selected.

Further comments:

- the rationale for the choice of the ALK inhibitors in this study should be better explained in particular for brigatinib which inhibits both ALK and EGFR.
- references to risk staging in NB are not appropriate (ref 9, 10 line 40), excellent recent reviews for NB also covering the more recent staging system and risk factors are available, also the paper by Akkerman et al. is adding the newest views on risk factors in NB.
- line 41-45: this section is somewhat misleading to readers, the authors refer to high risk NBs with genomic aberrations and refer to ALK mutations occurring in 8-10% of NB cases, it seems as this figure refers to high risk only but this is not correct, ALK mutations occur across all NB risk entities
- overall, the paper hinges on different findings and then moves further to NRAS as target, first protein coding genes are identified but hardly explored although the discussion starts on several protein coding genes while this is not expanded in the results section. The screen of a large panel of cell lines (disregarding ALK status or drug sensitivity) attributes a more global tumor suppressor activity for the miR-1304 but no further experimental data support the validity or mechanistic basis of these data, e.g. the NRAS or NF1 status in these cell lines could also be taken into account at least.
- the use of lonafarnib is interesting but it should be noted that the exact mechanism of action of this class of agents is currently unknown and cytotoxic actions may also be due to the modulation of other targets, including RhoB, the centromere-binding proteins and other proteins that have not yet been identified. In this stage this is underdeveloped in the current paper leaving doubt on the exact mechanism of the observed combinatorial effects between ALK and RAS inhibition.

Revision

Reviewers comments are repeated here in black type

Our responses are in red.

Legends of new added Figures and Tables are in blue in this file, for clarity.

Reviewer #1 - miRNA, neuroblastoma, in vivo models (Remarks to the Author):

“This is a very interesting manuscript addressing an important clinical need which is to decipher mechanism of drug resistant in highly aggressive tumors such as high-risk NB. The authors use a large panel of NB cell lines and clinically-representative in vivo models.

While the study is logically designed and experimentally-well executed, some of the proposed mechanisms are explored very superficially and need additional controls to strengthen their conclusions.

Below are indicated some concerns that should be addressed before acceptance for publication:

1. In figure 2A-D, the levels of mature miRNA after miRNA inhibition must be shown. The fact that only one of the miRNAs has an effect on cell proliferation questions whether the other miRNAs have been successfully inhibited.”

We thank the reviewer for the useful comments which we believe have helped us to substantially improve our manuscript. We have now added the expression levels of the 4 miRNAs following transfection of miRNA inhibitors (Extended Data Figure 2). *Hsa-miR-1304-5p* and *Hsa-miR-4746-5p* were successfully inhibited, *Hsa-miR-136-5p* was not detected via RT-qPCR, and *Hsa-miR-7975* was upregulated when analysed with a TaqMan assay specific for both 3p and 5p variants; no assay specific for *Hsa-miR-7975-5p* is available, and we hypothesize that *Hsa-miR-7975-3p* could compensate for 5p inhibition. This upregulation still does not result in any changes in the ALK TKI response, reinforcing our choice of excluding these miRNAs from further analyses in this context. We have referred to these data in the manuscript on page 4 lines 15-16:

“(…) and their expression was validated via RT-qPCR (Extended Data Fig. 2 A-D) (…)”.

Extended Data Figure 2. *miR-1304-5p* silencing upon transfection with specific miRNA inhibitors

(A-D) miRNA expression upon transfection with specific miRNA inhibitors tested by RT-qPCR, relative to the mimic-scrambled control (Scr-INH-Control): *miR-1304-5p* in A, *miR-136-5p* in B, *miR-4746-5p* in C and *miR-7975* in D, the latter using a gene expression assay that cannot discriminate between -3p and -5p variants. Numbers shown represent the average Ct values. N/D = not detectable. Statistical comparisons were conducted with the Student's t-test of means \pm SEM. * $p < 0.05$, ** $p < 0.01$.

A table showing the genomic location of these miRNAs and whether they cluster with other miRNAs or belong to the same family would be very informative.

We have added a table (Extended Data Table 1) showing the genomic location of these miRNAs and whether they cluster with other miRNAs or belong to the same family. Of note, only *miR-1304* and *miR-136* belong to a known miRNA family and only *miR-136* clusters with other miRNAs (< 10kb from *hsa-mir-136*). These data are referred to in the manuscript on page 4, lines 17 to 20:

"We have also reported the genomic location of these miRNAs and whether they cluster with other miRNAs or belong to the same family (Extended Data Table 1). Of note, only *miR-1304* and *miR-136* belong to a known miRNA family and only *miR-136* clusters with other miRNAs (< 10kb from *hsa-mir-136*)."

miRNA identified by the screen	Genomic location	Family	Other miRNAs in the cluster at the genomic location
Hsa-miR-1304	11q21	MIPF0001064; mir-1304	-
Hsa-miR-136	14q32.2	MIPF0000099; mir-136	hsa-mir-665 hsa-mir-431 hsa-mir-433 hsa-mir-127 hsa-mir-432 hsa-mir-136
Hsa-miR-4746	19p13.3	-	-
Hsa-miR-7975	19q13.42	-	-

Extended Data Table 1: Genomic location, family to which the miRNA belongs and clustered miRNAs in the region of the top 4 miRNAs (*hsa-mir-136*, *hsa-mir-1304*, *hsa-mir-7975* and *hsa-mir-4746*) identified in at least 3 of the 4 CRISPR GeCKO screens.

2. Figure 3A-B are insufficiently detailed. A table, that may be included in extended data, showing all hallmarks analyzed with their NES score and associated p-value of the performed GSEA must be included.

We have added tables in Supplementary File 4 showing all hallmarks analysed with their NES, p-values and the number of genes identified in each gene set that are included in the input data ("size" column, in Supplementary File 4). Gene sets with a positive NES are downregulated in samples transfected with the *miR-1304-5p* mimic and so we focussed on these. The HALLMARK_KRAS_SIGNALING is enriched with a positive NES in both cell lines and is in the top 10 gene sets with the largest quantity of genes (of the gene sets with a positive NES). These data are referred to in the manuscript on page 5, lines 22 to 24 and page 6, lines 1-6:

"Gene set enrichment analysis (GSEA)²⁶ was then performed with MSigDB's 'hallmark gene sets' (Supplementary File 4)²⁷. The top-10 enriched gene sets with a positive normalized enrichment score (NES) index were ranked by the quantity of genes in each gene set (i.e., the number of genes identified in each gene set that are included in the input data); for both KELLY and SH-SY5Y cell lines, 6 gene sets were identified that were significantly

($p < 0.05$ in at least one cell line) altered following *miR-1304-5p* transfection with a negative correlation: xenobiotic_metabolism, epithelial_mesenchymal_transition, estrogen_response_late, interferon_gamma_response, glycolysis, kras_signaling and mtorc1_signaling (Fig. 3B and Supplementary Table 4).”

3. Figure 3C seems to be saturated and only performed in one cell line. Same analyses in a second cell line must be included. Why a pan-RAS is used instead of an NRAS antibody? Please, explain.

We have included Western Blot analyses for an additional cell line (SHSY5Y plus KELLY) including an NRAS specific antibody (Extended data figure 7). These data are referred to in the text on page 6, lines 21 to 23:

“The expression levels of NRAS, RRAS, PTPN11 and IQGAP1 were assessed by RT-qPCR and Western blot upon *miR-1304-5p* mimic transfection into SH-SY5Y and KELLY cells (Extended Data Fig. 5B, Fig. 3C, Extended Data Fig. 7).”

Extended Data Figure 7. Western blot of the indicated proteins 72 h post-transfection of SH-SY5Y or KELLY cells with a miR-1304-5p mimic. The blots have been cropped to show the relevant bands.

Apparently, after overexpressing miR-1304-5p the total levels of ERK1/2 are also downregulated. Are these proteins also a potential target of miR-1304-5p? Since one of the main messages of the paper is that the downregulation of NRAS is responsible for miR-1304-5p effects, it's important to dissect at which levels the pathway is affected.

We agree with the reviewer that this is an interesting observation that total ERK1/2 levels are also affected. However, Target Scan did not predict that ERK1/2 is a direct target as it did for NRAS, RRAS, IQGAP1 and PTPN11 as well as many other target genes (3,336 were predicted by Target Scan) that we did not focus our attention on (we instead focussed on those genes in the MAP Kinase pathway that were common to the SHSY-5Y and KELLY cell lines as detailed in the manuscript on page 6 and Figure 3). Whilst we could confirm an effect of *miR-1304-5p* on expression of these 4 genes, it is not inconceivable that regulation of ERK expression could be at a different level, e.g., transcriptional, post transcriptional, translational, post translational) which in this context remains to be determined. However, our data points towards a downstream feedback loop on ERK expression rather than a direct effect due to the miRNA. On another note, the fact that transduction of the *miR-1304-5p* mimic shows a similar effect on cell viability as does NRAS silencing or pharmacological inhibition of the RAS/MAPK pathway

via the upstream (FTI) inhibitor lonafarnib, supports the hypothesis of a downstream feedback loop on other interactors of the RAS/MAPK pathway (including ERK) through a more upstream effect on other components of this pathway (e.g., NRAS), and this is also a reason why we selected a pharmacological inhibitor that could target this pathway upstream.

We would also like to point out that, since miRNAs act via a multitude of targets, *miR-1304-5p* is expected to do the same. We focussed on NRAS as one of the main targets based on our analyses and also as it is a clinically relevant and pharmacologically actionable target.

4. The 3'UTR analysis of Figure 3D is incomplete. To truly claim that these are direct targets, 3'UTR sites must be mutated and see if the reduction on luciferase activity is abolished.

We thank the reviewer for this useful comment. We agree that performing the mutagenesis at the site of interaction in the 3' UTR would strengthen the claim of a direct interaction. However, we analysed the ORF and UTR regions of the 4 target genes and there are 16 possible seed regions; we would therefore need to mutate each one individually and perhaps in combination which is a huge amount of work and outside the scope of this publication. Instead, we have addressed this caveat in the discussion (page 15, lines 4-11) referring to the experiment detailed below: "Due to the presence of 16 potential seed regions for direct targeting by *miR-1304-5p* in these 4 genes, site-directed mutagenesis to confirm direct action of this miRNA on these targets was not possible in this study and remains to be determined. It remains to be tested whether these genes are direct targets of *miR-1304-5p* or whether changes in their expression and activity are downstream effects. However, on overexpression of the 4 target genes individually and in combination with *miR-1304-5p* (Fig 3, extended data Fig 5), effects on the RAS/MAPK pathway are counteracted, which is supportive of these genes being direct targets of *miR-1304-5p*, although these data require further validation as detailed above".

5. Figure 3E is also incomplete. The levels of the targets (i.e NRAS, RRAS,etc..) should be shown after overexpression and with +/- *miR-1304-5p* overexpression to correlate the effects on the cell viability. Furthermore, monitoring MAPK activity through phospho-ERK would help to see if NRAS and others is sufficient to counteract the effects of *miR-1304-5p* on the RAS/MAPK pathway.

We have now included in the revised manuscript the levels of the tested targets (NRAS, RRAS, IQGAP1 and PTPN11) measured via western blot, after transduction with overexpression constructs (including the 3' UTR sequences) +/- *miR-1304-5p* mimic co-transfection, to demonstrate the effects on cell viability and protein expression levels (see extended data 5 below). Furthermore, we have also measured phospho-ERK levels, which show that overexpression of NRAS and the other targets is sufficient to counteract the effects of *miR-1304-5p* on the RAS/MAPK pathway. Notably, our previous experiments were conducted with vectors expressing only the ORFs of the four genes, with which it was not possible to show the effect of the *miR-1304-5p* mimic on the specific constructs. We therefore developed new constructs carrying the entire cDNA, with ORFs and UTRs, specifically the 3' UTRs and these data are presented in Fig. 3 and Extended Data Fig. 5 D-F (below). We hope that these experiments, along with those described above strengthen our manuscript findings further. These data are referenced in the manuscript on page 7, lines 6-11: "Notably, rescue experiments by transduction of the cDNAs of the 4 genes (including ORF and 3' UTR regions) showed an increase in SH-SY5Y cell viability (Fig.3E) and overexpression of NRAS, RRAS, IQGAP1 or PTPN11 (Extended Data Fig. 5 C-E) counteracts the effects of *miR-1304-5p* on the RAS/MAPK pathway target genes and activity as indicated by NRAS, RRAS, IQGAP1, PTPN11 and pERK expression (Fig.3E and Extended Data Fig. 5 C-E).".

Figure 3. *miR-1304-5p* inhibit NRAS expression

(...) (E) Cell viability (CTG assay) of SH-SY5Y cells transduced to express the cDNA including the 3'UTR sequence of each of the 4 indicated target genes +/- co-transfection of a *miR-1304-5p* mimic. Data are normalized to the viability of cells transfected with the empty vector. Data represent the means \pm SEM of two independent experiments. Please see extended data figure 5 D-E for corresponding expression levels of the target proteins. Statistical comparison: One-way ANOVA with Tukey's post-test (D, F, G), *** $p < 0.001$, **** $p < 0.0001$. (...)."

Extended Data Figure 5. Identification and validation of *miR-1304-5p* target genes

(...) (C-E) Western blot (C) and quantification (with ImageJ, D-E) of the indicated proteins at day 6 post-transduction with the cDNA of the 4 target genes (containing also the 3'UTR regions of each gene) +/- co-transfection of a *miR-1304-5p* mimic into SHSY5Y cells. Data are representative of n=2 (...).

6.-Figure 3F is also incomplete. The correlation of NRAS with neuroblastoma outcome is potentially interesting, but the authors did only a very preliminary analyses. Is this correlation maintained in other data sets with more samples (e.g. Kocak, SEQN)? A more thorough analyses indicating correlation with other clinical variables (age,

sex, stage, MYCN-amplification, genomic aberrations) will be more informative and may help to identify patient populations that would be more susceptible to the proposed therapy.

To address this question, we have included further analyses of other data sets as suggested by the reviewer. These cohorts cannot be stratified according to ALK TKI treatment, as this is still used in only a minority of patients and material for analysis is often inaccessible; the patient populations in these datasets could therefore only be considered as a whole group. However, analysis of the Kocak's dataset showed that NRAS expression levels predict for prognosis whereby higher expression levels correlate with an inferior event-free and overall survival in keeping with data shown in Fig 3F for the TARGET dataset ($p=0.034$, 79 vs 31 months median survival; log-rank test). Unfortunately, analysis of other clinical variables in the Kocak dataset including age, disease stage and sex is largely not feasible due to the low numbers of patients that fall into these categories. For example, there is a trend towards a correlation between NRAS expression with survival in patients >18 months of age and this analysis is significant for event-free survival ($p=0.032$) but not for overall survival (Extended data Fig.5I and J). In addition, stratification for survival according to disease stage showed a significant correlation of high NRAS with stages 1, 2 and 4, with a stronger trend and significance for event-free survival at stage 4 ($p=0.018$) (Extended data Fig.5 J). Notably, there is also a significant correlation between NRAS expression and survival for male patients ($p=0.0008$ for overall survival and $p=0.0006$ for event-free survival; Extended data Fig. 5I and J) whereby male patients with higher expression levels of NRAS have a significantly inferior outcome. According to these data, male patients and/or those with higher stage disease may stand to benefit more from this therapeutic approach. The correlation between NRAS expression with lower overall and event-free survival observed in the Kocak dataset is largely confirmed in the SEQC dataset (especially for all patients, age and sex stratification) but due to the reduced number of patients these correlations are not significant (Extended data Fig.5K and L). In the SEQC dataset we also observed a stronger correlation between higher NRAS expression with lower survival in high-risk patients and this is significant for event-free survival (Extended data Fig.5L). Based on this analysis, patients with high-risk neuroblastoma may also benefit more from NRAS-targeting therapeutic approaches.

These data are referred to in the manuscript on page 7, lines 17-26 and page 8, lines 1-6: "Notably, analysis of a second patients' dataset (Kocak) showed that NRAS expression levels predict for prognosis (...). Based on this analysis patients with high-risk neuroblastoma may also benefit more from NRAS-based therapeutic approaches."

Extended Data Figure 5. Identification and validation of *miR-1304-5p* target genes

(A) (...) (G-H) Density plots of the patients' population in Kocak (G) and SEQC (H) datasets, based on NRAS expression. (I-L) Expression of NRAS mRNA measured in 470 (Kocak, I-J) and 498 (SEQC, K-L) NB patients. From left to right the panels show overall survival (I) or event-free survival (J) in all patients, stratified by age (>18 months or <18 months), sex or stage in the Kocak dataset. From left to right the panels show overall survival (I) or event-free survival (J) in all patients, stratified by age, sex, risk-group or stage in the SEQC dataset. To note, there were no patients of 18 months of age). Log-rank test, with p values specified in the figure panels above (significant $p < 0.05$).

7. Data characterizing *miR-1304-5p* is insufficient. Caspase-3/7 DVD-cleavage assay is only partially informative and more analyses should be confirmed to understand the effect. In addition of using a method that directly quantifies number/percentage of apoptotic cells (e.g. DAPI staining, annexin-V staining), a western-blot showing the processing of caspase substrate such as PARP or fodrin will also confirm the apoptotic execution program. Furthermore, whether induction of apoptosis is the reason of decreased cell number will also have to be confirmed in presence of apoptosis inhibitors. Cell Cycle analysis is also partial. FACS plots must be shown and at different time points to confirm if any change in cell cycle precludes the induction of apoptosis.

We have added further data to confirm apoptosis including a method that directly quantifies the percentage of apoptotic (Annexin V staining) cells (Extended Data Fig.3 G-H) and a western blot showing PARP/cleaved PARP (Extended Data Fig.3F) as suggested by the reviewer. Additionally, we have conducted cell cycle analyses at different time points (48h-96h), and we have included representative FACS plots (Extended Data Fig.4). These data are referred to in the manuscript on page 5, lines 13-17: "Induction of apoptosis following transfection of

cells with a *miR-1304-5p* mimic was confirmed via WB showing increased levels of PARP cleavage (Extended Data Fig.3F) and flow cytometry by increased numbers of Annexin V-positive cells (Extended Data Fig.3G-H)".

Extended Data Figure 3. miR-1304-5p inhibits NB cell viability inducing apoptosis but not affecting cell cycle progression

(...) (E) Caspase 3/7 activity was measured by luminescence 72 h post-transfection and normalized for the Scr control transfected cells in the indicated cell lines. (F) Western blot of the indicated proteins 72 h post-transfection of SH-SY5Y or KELLY cells with a *miR-1304-5p* mimic. (G, H) Annexin V (APC) staining shows apoptotic cell fractions (represented in quadrants Q2 + Q3 as Annexin V positive, PI positive or negative) analysed 72 h post-transfection in SHSY5Y and KELLY cells with representative flow cytometry plots shown on the left and quantifications from two biological replicates on the right. Statistical comparison was conducted using a one-way Anova with Tukey`s post-test in C and D and a Student`s t-test in A, B, E, G, H. ****p<0.0001, *p<0.05.

SHSY5Y

48h

KELLY

SHSY5Y

96h

KELLY

Extended Data Figure 4. Ectopic overexpression of miR-1304-5p does not affect the cell cycle 48h and 96h post-transfection

Cell cycle profiles (according to PI incorporation into DNA) were determined 48 hours (A-F) or 96 hours (G-L) post-transfection with a *miR-1304-5p* mimic. (A-F) Percentage means and SEM of two biological replicates are shown for SHSY5Y (A) and KELLY (B) cells with representative cell cycle flow cytometry profiles (SHSY5Y in C and KELLY in D) and cell populations (SHSY5Y in E and KELLY in F) shown 48h post-transfection. (G-L) Percentage means and SEM of two biological replicates are shown for SHSY5Y (G) and KELLY (H) cells with representative cell cycle flow cytometry profiles (SHSY5Y in I and KELLY in J) and cell populations (SHSY5Y in K and KELLY in L), shown 96h post-transfection.

8. Figure 4 is incomplete. Caspase 3/7 analysis have to be complemented with other apoptotic assays, particularly offering information about the fraction of cells affected. It is also important to see if additive/synergic effect are also observed at the molecular level (ej. levels of cleaved caspase and caspase substrates).

We have conducted extensive work in order to show the synergistic effects of using a combination of a miR-1304-5p mimic and an ALK TKI on apoptosis. However, we had to reduce the ALK TKI concentration (from 1000nM to 100uM), in order to obtain enough cells for analyses by both Western blot and flow cytometry, which might decrease the synergistic effect observed compared to using a higher concentration (although data are still significant). Results are shown in Extended Data Fig. 8. which confirmed evidence previously shown by the Caspase assay (Fig.4 E, F). These data are referred to in the manuscript on page 8, lines 23-24 and page 9, lines 1-2: “Notably, we validated this effect on apoptosis via both flow cytometry to assess Annexin V/PI staining (Extended Data Fig. 8 A-D) and WB for PARP cleavage (...) (Extended Data Fig.8 E-F), using a combination of a *miR-1304-5p* mimic and an ALK TKI (ceritinib) in KELLY and SHSY5Y cells lines”.

Extended Data Figure 8. The *miR-1304-5p* mimic stimulates and enhances apoptosis when used in combination with an ALK TKI.

(A-D) Annexin V (APC) staining shows apoptotic cell fractions analysed 48h after transfection with a *miR-1304-5p* mimic with an additional 72 hours of treatment with an ALK TKI (ceritinib, 100nM concentration) in SHSY5Y (A-B) and KELLY cells (C-D) with cell populations shown in A and C and quantifications in B and D from two biological replicates. Statistical comparisons were conducted using a Student's t-test in B and D; * $p < 0.05$. (E-F) Western blot of the indicated proteins analysed 48h following transfection with a *miR-1304-5p* mimic with an additional 72 hours of treatment with an ALK TKI (ceritinib, 100nM concentration) in SHSY5Y (E) and KELLY cells (F). Early apoptosis = Annexin V positive, PI negative (quadrant Q3); late apoptosis = Annexin V and PI positive (quadrant Q2).

9.-In figure 5 and 6, caspase 3/7 assays should also be complemented as indicated above.

We have now added extensive work (as detailed above) showing increased apoptosis when cells are incubated with combinations of ALK TKIs and the FTI Imafarnib (Extended Data Fig. 12). These data are cited in the manuscript on page 10, lines 12-15: "Notably, these synergistic effects on apoptosis were confirmed via Annexin V staining assessed by flow cytometry (Extended Data Fig. 12 A-D) and PARP-cleavage via WB (Extended Data Fig. 12 E-F)".

A**SHSY5Y****C****KELLY****E****F**
Extended data Figure 12. A combination of ALK TKIs and the FTI lonafarnib increases apoptosis measured via Annexin V positivity and PARP cleavage in NB cells.

(A-D) Annexin V (APC) staining shows apoptotic cell fractions analysed after 72h of co-treatment with an ALK TKI (brigatinib or ceritinib, both at 100uM) with an FTI (lonafarnib, 1000uM) in SHSY5Y (A-B) and KELLY cells (C-D) with cell populations shown in A and C and quantifications (on the Annexin V positive population, either PI positive or negative, in quadrants Q2 + Q3) in B and D from two biological replicates. Statistical comparison was conducted using a Student's t-test in B and D; * $p < 0.05$. (E-F) Western blots of the indicated proteins analysed after 72h of co-treatment with an ALK TKI (brigatinib or ceritinib, both at 100uM) with an FTI (lonafarnib, 1000uM) in SHSY5Y and KELLY cells.

10. In figure 7C, mouse weight variation should be represented during the whole course of treatment. It has no sense to do it at a single time point.

We have included the analyses of mouse weight during the whole course of treatment in Extended Data Fig. 16 as well as in Extended Data Fig.15, together with a whole new animal study, confirming this evidence in a second PDX. These data are referenced in the manuscript at page 11, lines 12-14 and page 12, lines 3-6: "All compounds were well-tolerated, with no significant decrease in body weight nor lethal toxicity observed (Fig. 7C and Extended Data Fig. 16A)" (...) "A second *in vivo* study was conducted with subcutaneous injection of COG-N-415x PDX cells (Extended Data Figure 15 A-D), which confirmed data produced with the COG-N-426x (Felix) PDX. Notably, the COG-N-415x PDX line is MYCN amplified compared to the MYCN non-amplified COG-N-426x line, confirming that our findings could be relevant to a broader cohort of patients".

Extended Data Figure 16. Combining an ALK inhibitor (ceritinib) with the farnesyltransferase inhibitor (lonafarnib) significantly reduces tumour growth *in vivo* and lonafarnib effectively inhibits ALK and pERK expression

(A) Mouse body weight recorded at every day of treatment relative to baseline weights for each treatment group. Data points (n=6 mice) represent means \pm SEM (...).

Extended data Figure 15. A combination of an ALK inhibitor (ceritinib) with an FTI (lonafarnib) significantly reduces MYCN amplified PDX tumour growth *in vivo*

(A) Tumour volume over time of NSG mice injected sub-cutaneously with COG-N-415x primary NB cells which reached 75mm³ before daily administration of either vehicle (20% hydroxypropyl beta cyclodextrin), ceritinib (30 mg/kg), lonafarnib (40 mg/kg), or ceritinib and lonafarnib (combo, same concentrations as used for the single agent arms). The study endpoint was reached when tumours became 15 mm in diameter or following 30 days of treatment, whichever came first. Data points (n=8) represent means \pm SEM, shown until the experimental endpoint (as defined above) of the first animal within each treatment group. (B) Kaplan–Meier event-free survival analysis. *** p <0.001 (Log-rank test). (C) Mouse body weight at the experimental endpoint relative to baseline weights for each treatment group. Data points (n=8) represent means \pm SEM. One-way ANOVA with Tukey’s post-test determined significance at each experimental endpoint in A and C. **** p <0.0001. (D) Mouse body weight recorded on every day of treatment relative to baseline weights for each treatment group. Data points (n=8) represent means \pm SEM.

On-target effect should be better demonstrated. Analyzing Phospho-ERK in a single tumor section from a single mice is not sufficient. Better include western-blot analyses of samples and/or quantify % of P-ERK positive cells in multiple sections from 3-4 mice per group. Same applies to other immunohistochemistry analyses of the paper such as those shown in Extended Figure 9 and 10.

We have now included quantification of phospho-ERK staining intensity from 3 mice per treatment group, each in replicate sections in Extended Data Fig. 16E.

Extended Data Figure 16. Combining an ALK inhibitor (ceritinib) with the farnesyltransferase inhibitor (lonafarnib) significantly reduces tumour growth *in vivo* and lonafarnib effectively inhibits ALK and pERK expression

(...) (E) Quantification of phospho-ERK staining intensity from 3 mice per treatment group, each presented with data from replicate sections (n=2).

11. Figure 7B is misleading. The chosen endpoint may confuse the results of the study. Whether 15mm diameter of tumor may be adequate and ethically recommendable, 30 days of treatment it's insufficient to estimate the efficacy of the treatment. A 20-30% time extension from the last mouse sacrifice of the single-treatments arms would help to determine if the combination is more effective than the single treatments alone. So, in this case, a minimum of 35-40 days of follow-up would be the right choice.

We thank the reviewer for the comment, but we are unsure what is being requested. We do not aim to prolong the treatment in the combo compared to the single agent but rather the experiment was designed to detect a synergistic effect for the combination compared to the single agent. We selected the 30 days treatment based on our previous evidence and that gleaned from the literature using these drugs alone in murine/PDX models that showed effects without severe signs of toxicities. In our studies, the median EFS of the single agent groups are 17 and 20 days in the COG-N-426x PDX study and 17 and 17 in the COG-N-415x study, representing a 43.3% (17 days) and 33.3% (20 days) reduction in treatment time. Additionally, in the study with COG-N-415x, there is a 30%-time extension even from the last mouse sacrifice (day 21) of the single-treatment arms compared to the total days of treatment (30 days). For the COG-N-426x PDX model, the extension time is 20% considering the last animal sacrifice for the ceritinib group and less than 20% in only one mouse for the lonafarnib group, but more than 30% for all the other mice, so the reviewer's mentioned requirements are overall largely satisfied. However, please note that we have to abide by the UK Home Office law and accounting for the side-effects of the continual, daily oral gavage procedure, 30 consecutive days was the maximum time we could treat the animals.

Reviewer #2 - Neuroblastoma therapy, CRISPR (Remarks to the Author):
Authors use a high-throughput, genome-wide CRISPR-Cas9 knockout screens to identify miR-1304-5p loss as a desensitizer to ALK TKIs in both ALK wt and ALK mutant NB. They go onto that the FTI lonafarnib in addition to ALK TKIs cooperate in NB, inducing apoptosis in vitro. Combined treatment of an ALK mt NB patient derived xenograft with an FTI and an ALK TKI complete regression of tumour growth was observed although tumours rapidly regrew on cessation of therapy.

1. Introduction and discussion reiterates abstract, perhaps redundant.

We thank the reviewer for the useful comments, and we hope this edited version is acceptable.

2. Can Figs 2E and 2F use some sort of marker or color coding to distinguish ALK wt from ALK mt lines?

We have now highlighted the ALK WT cells included in these graphs and added a description in the legend of Figure 2E and F: "(...) ALK WT cells are highlighted in blue= CHP-134, GIMEN, LA-N-6, NBL-S, NGP. The other cell lines have ALK mutations as specified in Extended Data Table 2 (...)". For details on the specific ALK mutations in the other cells please refer to Extended Data Table 2.

In 2E, why do ALK wt lines show robust responses to the miRNA inhibitor?

Some ALK WT lines express ALK mRNA and protein and have p-ALK expression (PMID: 21625996) indicating that ALK is active which may account for the observed results.

Only NGP is flat, and many ALK mutant lines show responses less robust than those seen in ALK wt lines? Can the authors include an assay for inhibitors that show these inhibitors inhibit their appropriate targets as anticipated?

Please see response to reviewer 1, point 1

3. In Fig 3F, was the group split in half (the most accurate way to do this), or was the group split to maximize the p-value? If the latter, please show data for the 143 patients split in half.

For the TARGET RNAseq dataset (n= 143) the cut-off between NRAS high and low was selected to have a cohort of NRAS high patients of 10% (z-score=1.2). To obtain data with >10% of the population as NRAS high, the z-score would be below 1 and to obtain data for 50% of patients as NRAS high, the z-score would be below 0.1 which is not representative of a population. These data are in keeping with our hypothesis of NRAS as one of the bypass signalling tracks of ALK TKI resistance, and so upregulated in a higher number of ALK TKI treated/ relapsed patients whereby survival analysis using 50% of patients with NRAS upregulation could be obtained and is therefore representative. Additionally, analysis with 143 patients decreases the possibility of significantly splitting a cohort in half. Despite this, we have performed additional analyses using two additional datasets (Kocak and SEQC – see response to reviewer 1, point 6) where slightly more patients were analysed. With these data, we plotted the distribution of NRAS expression among the samples (see the density plots in Extended Data Fig.5G and H), and separated the two populations observed, so the bell-shaped curve (NRAS low) from the additional small peaks at higher expression levels (NRAS high), for both data sets. This might lead to a smaller number of patients in the NRAS high group, but it is more representative of the two sub-populations based on NRAS expression, avoiding biased analysis.

4. Can experiments in Fig 4 be extended to include ALK wt lines?

We have extended the experiments shown in Fig.4 to include an ALK WT cell line (LA-N-6) (Extended Data Fig. 10). Our results confirm the combined effects on cell viability and apoptosis on treatment with a combination of a *miR-1304-5p* mimic and an ALK TKI (brigatinib or ceritinib) (Extended Data Fig. 10). These data are referred to on page 8, lines 14-21: "(...) which significantly decreased the ED_{50s} for both brigatinib and ceritinib in *MYCN* non-amplified SH-SY5Y (Fig. 4A and B), *MYCN* amplified KELLY cells (Fig. 4C and D) and *MYCN* non-amplified ALK wild type LA-N-6 cells (Extended Data Fig. 10A and B). These data suggest that miRNA-based therapies could be promising strategies to sensitize cells to ALK inhibition. Indeed, transfection of NB cells with *miR-1304-5p* mimics prior to treatment with ALK TKIs significantly increased apoptosis as determined by caspase 3/7 activity

compared to ceritinib alone for SH-SY5Y, KELLY (Fig. 4E and F) and LA-N-6 cells (Extended Data Fig. 10C) as well as for brigatinib alone for SH-SY5Y (Fig. 4E and F) and LA-N-6 cells (Extended Data Fig. 10C)".

Extended data figure 10. The *miR-1304-5p* mimic is a therapeutic target when used in combination with ALK TKIs in the ALK WT neuroblastoma cell line LA-N-6

(A-B) LA-N-6 cell viability (measured via CTG assay) and ED₅₀ upon treatment with a combination of a *miR-1304-5p* mimic and brigatinib (A) or ceritinib (B) for 72 h at the indicated doses. ED₅₀ values shown in the graphs on the right were calculated from the non-linear fit curves on the left. Data shown are representative of two independent experiments. Statistical comparisons were conducted with a two-way ANOVA with Sidak's post-test and Student's t-test on means ± SEM. **p*<0.05. (C) Apoptosis determined by caspase 3/7 activity in LA-N-6 cells treated with a combination of a *miR-1304-5p* mimic and brigatinib (1000nM) or ceritinib (1000nM). Results shown are representative of two independent experiments. Significance was determined using a one-way ANOVA with Tukey's post-test of the means ± SEM. **p*<0.05, ****p*<0.001. (D) Western blot of the indicated proteins 72 h post-transfection of LA-N-6 with a *miR-1304-5p* mimic.

In Figs 4-6, does combination therapy cooperate to block signaling downstream of NRAS?

We have added extensive evidence on the effect of the combination treatment on signalling downstream of NRAS (specifically phospho-ERK levels), confirming inhibition of the MAP Kinase pathway upon *miR-1304-5p* transfection alone (Extended Data Fig. 3F – see response to reviewer 1, point 7), in combination with ALK TKI (Extended Data Fig. 8E-F) and also upon treatment with ALK TKI (brigatinib, ceritinib) or FTI alone (lonafarnib) or ALK TKI + FTI in combination (Extended Data Fig. 12E-F); as well as in ALK WT cells as shown above (Extended Data Fig. 10D). These data are referenced in the manuscript on page 9, line1 and page 10, lines 13-14: "(...)as well as showing inhibition of downstream NRAS signaling (specifically phospho-ERK levels) (...)".

Extended Data Figure 8. The *miR-1304-5p* mimic stimulates and enhances apoptosis when used in combination with an ALK TKI.

(A-D) Annexin V (APC) staining shows apoptotic cell fractions analysed 48h after transfection with a *miR-1304-5p* mimic with an additional 72 hours of treatment with an ALK TKI (ceritinib, 100nM concentration) in SHSY5Y (A-B) and KELLY cells (C-D) with cell populations shown in A and C and quantifications in B and D from two biological replicates. Statistical comparisons were conducted using a Student's t-test in B and D; * $p < 0.05$. (E-F) Western blot of the indicated proteins analysed 48h following transfection with a *miR-1304-5p* mimic with an additional 72 hours of treatment with an ALK TKI (ceritinib, 100nM concentration) in SHSY5Y (E) and KELLY cells (F). Early apoptosis = Annexin V positive, PI negative (quadrant Q3); late apoptosis = Annexin V and PI positive (quadrant Q2).

A**SHSY5Y****C****KELLY****E****F**
Extended data Figure 12. A combination of ALK TKIs and the FTI lonafarnib increases apoptosis measured via Annexin V positivity and PARP cleavage in NB cells.

(A-D) Annexin V (APC) staining shows apoptotic cell fractions analysed after 72h of co-treatment with an ALK TKI (brigatinib or ceritinib, both at 100uM) with an FTI (lonafarnib, 1000uM) in SHSY5Y (A-B) and KELLY cells (C-D) with cell populations shown in A and C and quantifications (on the Annexin V positive population, either PI positive or negative, in quadrants Q2 + Q3) in B and D from two biological replicates. Statistical comparison was conducted using a Student's t-test in B and D; * $p < 0.05$. (E-F) Western blots of the indicated proteins analysed after 72h of co-treatment with an ALK TKI (brigatinib or ceritinib, both at 100uM) with an FTI (lonafarnib, 1000uM) in SHSY5Y and KELLY cells.

5. Generalizability of results requires extending *in vivo* experiments beyond a single pdx line. Authors claim that the combination therapy works *in vivo* to induce apoptosis, did I miss these data? If I did not, can these data be added?

Whilst we have not specifically shown apoptosis *in vivo*, *in vitro* we do see apoptosis in the PDX-derived cells models cultured *ex vivo* (please see Fig.6B, D and F).

Additionally, we have now included a second PDX line for extensive *in vivo* studies which confirm the evidence observed in the first PDX line. Notably, the second PDX line is MYCN amplified compared to the first PDX which is MYCN non-amplified, confirming that our findings could be relevant to a broader cohort of patients (Extended Data Fig. 15). These data are referred to in the manuscript at page 12, lines 2-6: "A second *in vivo* study was conducted with subcutaneous injection of COG-N-415x PDX cells (Extended Data Figure 15 A-D), which confirmed data produced with the COG-N-426x PDX. Notably, the COG-N-415x PDX line is MYCN amplified compared to the MYCN non-amplified COG-N-426x line, confirming that our findings could be relevant to a broader cohort of patients".

Extended data Figure 15. A combination of an ALK inhibitor (ceritinib) with a FTI (lonafarnib) significantly reduces MYCN amplified PDX tumour growth *in vivo*.

(A) Tumour volume over time of NSG mice injected sub-cutaneously with COG-N-415x primary NB cells which reached 75mm³ before daily administration of either vehicle (20% hydroxypropyl beta cyclodextrin), ceritinib (30 mg/kg), lonafarnib (40 mg/kg), or ceritinib and lonafarnib (combo, same doses). Study endpoint is tumours reaching 15 mm diameter or following 30 days of treatment, whichever came first. Data points (n=8) represent means \pm SEM, shown until the experimental endpoint (as defined above) of the first animal within each treatment group. (B) Kaplan–Meier event-free survival analysis. *** $p < 0.001$ (Log-rank test). (C) Mouse body weight at the experimental endpoint relative to baseline weights for each treatment group. Data points (n=8) represent means \pm SEM. One-way ANOVA with Tukey's post-test determined significance at each experimental

endpoint in A and C. **** $p < 0.0001$. (D) Mouse body weight recorder every day of treatment relative to baseline weights for each treatment group. Data points (n=8) represent means \pm SEM.

Reviewer #3 - Neuroblastoma, ALK (Remarks to the Author):

The authors have conducted a CRISPR screen to identify genetic factors that influence sensitivity to ALK inhibition in neuroblastoma (NB). To this end, the NB SY5Y cell line was selected and surviving clones under exposure to two ALK inhibitors were identified to allow selection of genes which render resistance to ALK inhibition. This revealed both protein and non-coding genes. In a further selection step, one single miRNA across all four experiments and four miRNAs for three out of four screens were identified. The authors test these four miRNAs further and identify miR-1304-5p for subsequent functional studies. For this miRNA a more global TSG function is suggested in NB and NRAS is identified as regulated target. Next, NRAS is identified as target regulated by miR-1304-5p. Finally, in a series of in vitro and in vivo experiments, the combination of ALK inhibition with NRAS inhibition using the farnesyl transferase inhibitor Lonafarnib is shown to yield additive or synergistic effects, although upon interruption of treatment in mouse xenografts tumors do recur.

Major comments:

The major concern for this paper is that previous work has been reported also using a CRISPR screen under control and ALK inhibition conditions already pointing in a more direct and more convincing way the critical importance of NRAS activation (Berlak et al., 2022) and thus also strongly reducing the novelty of this paper.

We believe that the publication of Berlak et al does not diminish the impact nor novelty of our study. On the contrary, the data presented by both manuscripts present different mechanisms investigated using alternative approaches.

Specifically, in response to feedback from the reviewer:

1. “NRAS mut and MAPK not novel”

The Berlak et al, paper points towards the activation of NRAS as a mechanism of ALK TKI resistance, but focuses on genetic mutations rather than increased expression and furthermore, they do not add any novel mechanistic insights, rather they confirm the role of the MAPK pathway (including NF1 and NRAS mutant forms) and the response to ALK TKIs in NB. Indeed, the role of this pathway in this context has been known for some time, as the authors cite in the introduction to their paper (“Activating mutations most often occur in the kinase domain, leading to increased ALK downstream signalling via the PI3K/AKT, RAS/MAPK and JAK/STAT pathways, promoting neuroblastoma cell survival and proliferation [14–19, 22–24]”). In contrast, our research adds further mechanistic insight through the identification of *miR-1304-5p*, a previously unexplored mechanism of NRAS regulation in NB. Furthermore, we show the potential of this as a novel biomarker adding to the impact of our research.

We would argue that *miR-1304-5p* is a novel finding from our work that is NOT shown in the Berlak et al paper. Unlike NF1, *miR-1304-5p* has never been characterized in NB before and neither in the context of an ALK TKI response. Moreover, from a clinical perspective, the implication of a miRNA gene loss of function in treatment resistance, has therapeutic potential for the clinic in the future (via stimulation with mimics or oligonucleotides when this becomes clinically possible as technology develops and delivery mechanisms improve), that might not be achievable for oncosuppressor protein coding genes. Also, as biomarkers, miRNAs could also perform better than protein-coding genes (including NF1), considering miRNAs have already been detected in cancer tissues and liquid biopsies from cancer patients, either serum-free or transported in exosomes (1–4). This may change clinical decisions regarding treatment choices and monitoring of treatment response. Hence, our data suggest that *miR-1304-5p* might be further investigated as a prognostic factor for treatment response to ALK TKIs in the future. Therefore, identification of *miR-1304-5p* from our CRISPR screens not only represents a completely novel

mechanism of ALK TKI resistance in NB but also a notable source of novel emerging therapeutic targets and biomarkers for future clinical trials.

For all these reasons we strongly believe that our manuscript is novel and brings alternative insights into high-risk neuroblastoma therapy that should be considered in the design of future clinical trials and therefore merits publication in a journal with a broad readership.

This work, although referred together with other papers, is largely neglected in putting the presented data into context. While CRISPR screen like any screen will pick up and miss targets, it is troublesome that in the present screen NF1 was not identified as target, in particular given that the same cell line SY5Y was selected.

2. “it is troublesome that in the present screen NF1 was not identified as target”

In our CRISPR screen the ‘top hits’ were selected after a stringent process analysing data from 4 independent CRISPR screens using 2 different ALK TKIs. From these combined screens, we selected our hits using a > 1.8-fold change of negatively enriched sgRNAs, in treated versus DMSO conditions identified in at least 3 out of 4 screens. NF1 did appear in our list of hits as being ‘slightly enriched’ but it did not reach our stringent threshold in any of the 4 screens (fold changes of: 1.57 in ceritinib 300nM, 1.67 ceritinib 750nM, 1.24 in brigatinib 300nM, 1.18 in brigatinib 750 nM). We are happy to add this information to the discussion if the reviewer insists but, these were independent experiments from two different projects and manuscripts which are ultimately not directly comparable.

Further comments:

- the rationale for the choice of the ALK inhibitors in this study should be better explained in particular for brigatinib which inhibits both ALK and EGFR.

3. “...the rationale for the choice of the ALK inhibitors in this study should be better explained in particular for brigatinib which inhibits both ALK and EGFR”

Whilst lorlatinib is under clinical trial for ALK-positive NB, other ALK TKIs are under consideration as the subjects of clinical trials and are being used on compassionate grounds for many patients. Use of multiple ALK TKIs clinically for this patient subset is to be expected given the known need to clinically ‘rotate’ through ALK inhibitors for adults with ALK-positive lung cancer as well as children with ALK-positive anaplastic large cell lymphoma (ALCL). Hence, we chose to investigate other ALK TKIs including brigatinib. Brigatinib has been successfully used in phase I, II and III clinical trials in ALK-positive NSCLC and ongoing trials are focussing on brigatinib alone or in combination with other treatments in ALK-positive childhood ALCL which will establish their safety in this population and allow their use for other ALK-positive paediatric cancers (5–11). While brigatinib, as rightly indicated by the reviewer, inhibits both ALK and EGFR, all ALK TKIs have some ‘off-target’ activity. Hence, by studying a combination of these, we will be better placed to design therapeutic approaches for ALK TKI relapse disease when driven by a bypass mechanism. For example, ceritinib also used in Berlak et al, is also an IGF-1R inhibitor, and lorlatinib also inhibits ROS1.

Notably, the approach of our work is to ultimately test a combination treatment with low toxicities and high synergism to use upfront in children with high-risk NB and prevent resistance from occurring in the first place. This approach is also based on recent clinical evidence focussing on lorlatinib where, despite initial benefits, resistance often occurs. Lorlatinib also has distinct side-effects compared to other ALK TKIs which can limit its use including neurotoxicity and considerable weight gain. It is therefore highly likely that other ALK TKIs will be used to treat NB and should therefore be investigated in this manner including brigatinib.

- the use of lonafarnib is interesting but it should be noted that the exact mechanism of action of this class of agents is currently unknown and cytotoxic actions may also be due to the modulation of other targets, including RhoB, the centromere-binding proteins and other proteins that have not yet been identified. In this stage this is

underdeveloped in the current paper leaving doubt on the exact mechanism of the observed combinatorial effects between ALK and RAS inhibition.

4. “...the use of lonafarnib is interesting but it should be noted that the exact mechanism of action of this class of agents is currently unknown... leaving doubt on the exact mechanism of the observed combinatorial effects between ALK and RAS inhibition”

Many drugs in clinical use have incompletely understood mechanisms of action – this has never precluded their use in the clinic. Our analyses identified NRAS as a target of *miR-1304-5p*, the latter of which was identified by our screens. As there are no known specific inhibitors of NRAS, this led us to investigate farnesyltransferase inhibitors for which there are drugs. Indeed, these have been approved for use in children with progeria syndrome or laminopathies (12, 13). We also considered downstream signalling pathway proteins as therapeutic targets but as there are several pathways downstream of NRAS, we settled on an upstream component, i.e., farnesyltransferases. Combined with the data presented by Berlak et al, this decision was reinforced: Berlak et al showed that MEK (trametinib) and Raf inhibitors (LY3009120), in a preliminary *in vitro* experiment, only had partial effects on cell death. Furthermore, LY3009120 has only been investigated in one unsuccessful phase I clinical trial so far (14), while trametinib has been shown to have limited efficacy in ALK-expressing NB due to activation of pro-survival feedback signaling through Akt/mTORc (15). In contrast, we show both *in vitro*, *ex vivo* and *in vivo* that ALK TKI and the FTI lonafarnib, act with considerable synergism in all these preclinical models without any obvious toxicity, paving the way for the use of ALK TKIs in combination with FTIs in children with NB.

- references to risk staging in NB are not appropriate (ref 9, 10 line 40), excellent recent reviews for NB also covering the more recent staging system and risk factors are available, also the paper by Akkerman et al. is adding the newest views on risk factors in NB.

We thank the reviewer for the useful suggestion and we added the suggested literature in the text on page 2, lines 17-20: “ALK amplification and/or ALK kinase-domain point mutations (8-10% overall and 15% of high-risk cases) ^{11,12,17-19}. Recent evidence has also highlighted that telomere maintenance mechanisms and mutations in RAS and p53 and alterations in their pathways may be a useful source for risk determination at diagnosis and treatment stratification ²⁰”. If the reviewer had other specific suggestions on the material to cite, we would be happy to include this too after revision.

- line 41-45: this section is somewhat misleading to readers, the authors refer to high risk NBs with genomic aberrations and refer to ALK mutations occurring in 8-10% of NB cases, it seems as this figure refers to high risk only but this is not correct, ALK mutations occur across all NB risk entities

We thank the reviewer for this comment and have edited our manuscript accordingly: “...ALK amplification and/or ALK kinase-domain point mutations (8-10% overall and 15% of high-risk cases of cases)”.

- overall, the paper hinges on different findings and then moves further to NRAS as target, first protein coding genes are identified but hardly explored although the discussion starts on several protein coding genes while this is not expanded in the results section. The screen of a large panel of cell lines (disregarding ALK status or drug sensitivity) attributes a more global tumor suppressor activity for the *miR-1304* but no further experimental data support the validity or mechanistic basis of these data, e.g. the NRAS or NF1 status in these cell lines could also be taken into account at least.

We have strengthened several results, including further evidence towards an association between *miR-1304-5p* and the tested MAPK target genes (including NRAS) and a clear increase in apoptosis in the context of ALK TKIs. Please see response to reviewer 1, point 7 and reviewer 2, point 4. We have also analysed NF1 expression in the cell lines of this manuscript (SHSY5Y and KELLY) before and after *miR-1304-5p* overexpression, confirming that NF1 is not a functional target of *miR-1304-5p* (please see the image below prepared for the reviewer but not included in the manuscript).

Extended Data Figure. miR-1304-5p inhibits NB cell viability inducing apoptosis but not affecting cell cycle progression in CHLA20 cells

(A) Confirmation of *miR-1304-5p* upregulation by RT-qPCR upon transfection of the mimic (hsa-miR-1304-5p) relative to the mimic-scrambled control (Scr control) with image inserts showing their effects on cell confluency for CHLA20. (B) Caspase 3/7 activity was measured by luminescence normalized for the Scr control transfected cells in the indicated cell lines 72 h post-transfection. (C) Cell cycle analysis 72 h post-transfection of the indicated cell lines, shown as the percentage of cells in each phase of the cell cycle. Data shown represent the means \pm SEM of three independent experiments. Statistical comparison was conducted using a one-way Anova with Tukey's post-test in E and F and a Student's t-test in A-D. *p<0.05. Not shown in the manuscript.

References

1. C. L. Au Yeung, N. N. Co, T. Tsuruga, T. L. Yeung, S. Y. Kwan, C. S. Leung, Y. Li, E. S. Lu, K. Kwan, K. K. Wong, R. Schmandt, K. H. Lu, S. C. Mok, Exosomal transfer of stroma-derived miR21 confers paclitaxel resistance in ovarian cancer cells through targeting APAF1, *Nat. Commun.* **7**, 1–14 (2016).
2. H. L. Zhang, L. F. Yang, Y. Zhu, X. D. Yao, S. L. Zhang, B. Dai, Y. P. Zhu, Y. J. Shen, G. H. Shi, D. W. Ye, Serum miRNA-21: Elevated levels in patients with metastatic hormone-refractory prostate cancer and potential predictive factor for the efficacy of docetaxel-based chemotherapy, *Prostate* **71**, 326–331 (2011).
3. H. Sasaki, M. Yoshiike, S. Nozawa, W. Usuba, Y. Katsuoka, K. Aida, K. Kitajima, H. Kudo, M. Hoshikawa, Y. Yoshioka, N. Kosaka, T. Ochiya, T. Chikaraishi, Expression Level of Urinary MicroRNA-146a-5p Is Increased in Patients With Bladder Cancer and Decreased in Those After Transurethral Resection, *Clin. Genitourin. Cancer* **14**, e493–e499 (2016).
4. L. Sadovska, P. Zayakin, K. Eglītis, E. Endzeliņš, I. Radoviča-Spalviņa, E. Avotiņa, J. Auders, L. Keiša, I. Liepniece-Karele, M. Leja, J. Eglītis, A. Linē, Comprehensive characterization of RNA cargo of extracellular vesicles in breast cancer patients undergoing neoadjuvant chemotherapy, *Front. Oncol.* **12**, 1–16 (2022).
5. S. Sugawara, M. Kondo, T. Yokoyama, T. Kumagai, M. Nishio, K. Goto, K. Nakagawa, T. Seto, N. Yamamoto, K. Kudou, T. Asato, P. Zhang, Y. Ohe, Brigatinib in Japanese patients with tyrosine kinase inhibitor-naive ALK-positive non-small cell lung cancer: first results from the phase 2 J-ALTA study, *Int. J. Clin. Oncol.* **27**, 1828–1838 (2022).
6. D. R. Camidge, H. R. Kim, M.-J. Ahn, J. C.-H. Yang, J.-Y. Han, J.-S. Lee, M. J. Hochmair, J. Y.-C. Li, G.-C. Chang, K. H. Lee, C. Gridelli, A. Delmonte, R. Garcia Campelo, D.-W. Kim, A. Bearz, F. Griesinger, A. Morabito, E. Felip, R. Califano, S. Ghosh, A. Spira, S. N. Gettinger, M. Tiseo, N. Gupta, J. Haney, D. Kerstein, S. Popat, Brigatinib versus Crizotinib in ALK-Positive Non-Small-Cell Lung Cancer, *N. Engl. J. Med.* **379**, 2027–2039 (2018).
7. D. R. Camidge, H. R. Kim, M. J. Ahn, J. C. H. Yang, J. Y. Han, M. J. Hochmair, K. H. Lee, A. Delmonte, M. R. Garcia Campelo, D. W. Kim, F. Griesinger, E. Felip, R. Califano, A. Spira, S. N. Gettinger, M. Tiseo, H. M. Lin, N. Gupta, M. J. Hanley, Q. Ni, P. Zhang, S. Popat, Brigatinib Versus Crizotinib in Advanced ALK Inhibitor-Naive ALK-Positive Non-Small Cell Lung Cancer: Second Interim Analysis of the Phase III ALTA-1L Trial, *J. Clin. Oncol.* **38**, 3592–3603 (2020).
8. D. R. Camidge, H. R. Kim, M. J. Ahn, J. C. H. Yang, J. Y. Han, M. J. Hochmair, K. H. Lee, A. Delmonte, M. R. Garcia Campelo, D. W. Kim, F. Griesinger, E. Felip, R. Califano, A. I. Spira, S. N. Gettinger, M. Tiseo, H. M. Lin, Y. Liu, F. Vranceanu, H. Niu, P. Zhang, S. Popat, Brigatinib Versus Crizotinib in ALK Inhibitor-Naive Advanced ALK-Positive NSCLC: Final Results of Phase 3 ALTA-1L Trial, *J. Thorac. Oncol.* **16**, 2091–2108 (2021).
9. R. Heregger, F. Huemer, G. Hutarew, S. Hecht, L. Cheveresan, D. Kotzot, E. Schamschula, G. Rinnerthaler, T. Melchardt, L. Weiss, R. Greil, Sustained response to brigatinib in a patient with refractory metastatic pheochromocytoma harboring R1192P anaplastic lymphoma kinase mutation: a case report from the Austrian Group Medical Tumor Therapy next-generation sequencing registry and discussion, *ESMO Open* **6**, 100233 (2021).
10. J. Chen, F. Facchinetti, F. Braye, A. A. Yurchenko, L. Bigot, S. Ponce, D. Planchard, A. Gazzah, S. Nikolaev, S. Michiels, D. Vasseur, L. Lacroix, L. Tselikas, C. Nobre, K. A. Olaussen, F. Andre, J. Y. Scoazec, F. Barlesi, J. C. Soria, Y. Loriot, B. Besse, L. Friboulet, Single-cell DNA-seq depicts clonal evolution of multiple driver alterations in osimertinib-resistant patients, *Ann. Oncol.* **33**, 434–444 (2022).
11. M. J. Ahn, H. R. Kim, J. C. H. Yang, J. Y. Han, J. Y. C. Li, M. J. Hochmair, G. C. Chang, A. Delmonte, K. H. Lee, R. G. Campelo, C. Gridelli, A. I. Spira, R. Califano, F. Griesinger, S. Ghosh, E. Felip, D. W. Kim, Y. Liu, P. Zhang, S. Popat, D. R. Camidge, Efficacy and Safety of Brigatinib Compared With Crizotinib in Asian vs. Non-Asian Patients With Locally Advanced or Metastatic ALK-Inhibitor-Naive ALK+ Non-Small Cell Lung Cancer: Final Results From the Phase III ALTA-1L Study, *Clin. Lung Cancer* **23**, 720–730 (2022).
12. FDA, FDA Approves First Treatment for Hutchinson-Gilford Progeria Syndrome and Some Progeroid Laminopathies (2020) (available at <https://www.fda.gov/news-events/press-announcements/fda-approves-first-treatment-hutchinson-gilford-progeria-syndrome-and-some-progeroid-laminopathies>).
13. S. Dhillon, Lonafarnib: First Approval, *Drugs* **81**, 283–289 (2021).

14. R. J. Sullivan, A. Hollebecque, K. T. Flaherty, G. I. Shapiro, J. R. Ahnert, M. J. Millward, W. Zhang, L. Gao, A. Sykes, M. D. Willard, D. Yu, A. E. Schade, K. A. Crowe, D. L. Flynn, M. D. Kaufman, J. R. Henry, S. Bin Peng, K. A. Benhadji, I. Conti, M. S. Gordon, R. V. Tiu, D. S. Hong, A phase I study of LY3009120, a pan-RAF inhibitor, in patients with advanced or metastatic cancer, *Mol. Cancer Ther.* **19**, 460–467 (2020).
15. G. Umapathy, J. Guan, D. E. Gustafsson, N. Javanmardi, D. Cervantes-Madrid, A. Djos, T. Martinsson, R. H. Palmer, B. Hallberg, MEK inhibitor trametinib does not prevent the growth of anaplastic lymphoma kinase (ALK)-addicted neuroblastomas, *Sci. Signal.* **10** (2017), doi:10.1126/scisignal.aam7550.

Reviewers' Comments:

Reviewer #1:

Remarks to the Author:

I would like to thank the authors for their efforts in addressing my concerns and performing the suggested experiments.

Reviewer #2:

Remarks to the Author:

Authors did not address my request to analyze apoptosis in vivo, despite including a second apoptosis expt and re-illustrating for me that they did show this in vitro.

Reviewer #4:

Remarks to the Author:

The revised manuscript and additional work to address reviewers' comments is much appreciated but does not improve upon the lack of novelty for this manuscript based on approach and agents used. As mentioned, Berlak et al performed ALK TKI therapy with CRISPR screen following treatment with a more clinically applicable agent in NB along with ceritinib (lorlatinib) to identify ALK TKI resistance mechanisms and already showed dependence on the RAS pathway contributing to ALK TKI resistance. While I agree with the authors that the focus of this manuscript diverged differently based on their CRISPR results, to focus on miRNA influences to ALKi resistance, the additional data does not completely support that miR-1304-5p directly suppresses NRAS, nor that the rather effective in vivo combination of an FTI and ALK TKI are truly working through inhibition of NRAS gene expression specifically to directly improve in vivo ALK TKI responses. More work needs to be done to demonstrate their claim that Lornfarnib works in this setting via suppression of NRAS given the xenograft finding of decreased Phospho ERK expression is rather nonspecific, as pERK is downstream of many pathways.

In their response to review #3, the authors state: "Also, as biomarkers, miRNAs could also perform better than protein-coding genes (including NF1), considering miRNAs have already been detected in cancer tissues and liquid biopsies from cancer patients".

In contrast to this comment, Berko et al. (Nat Comm 2023) showed very clearly that NF1 mutations (and other RAS pathway and ALK compound mutations) arise as a mechanism of ALK inhibitor (Lorlatinib) resistance clinically in ALK-driven relapsed/refractory neuroblastoma. This was detected in the circulating tumor DNA in real time from patient samples collected on the Phase I clinical trial. Therefore, it is unclear how detection of miRNA's can outperform ctDNA evaluations for ras/alk resistance mutations as a biomarker of ALK TKI resistance (maybe I misunderstood, but protein coding genes are not being evaluated as integral biomarkers, only ctDNA for genomic alterations in said protein coding genes like NF1).

The authors defend their choice for using brigatinib and ceritinib as ALK inhibitors for their NBL CRISPR screen by saying: "Use of multiple ALK TKIs clinically for this patient subset is to be expected given the known need to clinically 'rotate' through ALK inhibitors for adults with ALK-positive lung cancer as well as children with ALK-positive anaplastic large cell lymphoma (ALCL)." Stating that ALK fusion+ cancers and ALK mutated/amplified NBL are the same with respect to how ALK inhibitors are used, prioritized, or chosen is inappropriate. Tumors that harbor ALK FUSIONS (ALCL, SCLC) are exquisitely sensitive to all generations of ALK inhibitors, leading patients to start with a first generation ALK TKI (crizotinib), then once resistant, rotate to second generation ALK TKI's (ensartinib, brigatinib), and once resistant go on to 3rd generation TKI like lorlatinib (that was designed to target resistant ALK mutations that arise in ALK fusion + cancers treated with ALK TKI's). This "rotation through many ALK TKI's" is not the current clinical practice in neuroblastoma given early generation inhibitors, even in combination with chemotherapy, show very limited responses clinically and preclinically in NB. This is because NBs harbor de novo ALK hotspot activating mutations (or amplification) that are intrinsically resistant to first and second generation ALK TKI's, which has been shown and published, thus limiting choice of ALKi to use clinically. As mentioned, so far, only lorlatinib seems to show clinical responses across aggressive

recurrent and newly diagnosed HR NBL tumors, with responses seen in all hotspot ALK mutations. Therefore, the lack of lorlatinib use on this study specifically to evaluate for clinically relevant NBL ALK TKI resistant mechanisms significantly dampens the possibility that these results and miR-1304-5p is clinically impactful as a biomarker.

I agree with the authors statement, however, that ALK TKI 's in combination with other targeted therapies is greatly warranted to overcome tumor heterogeneity and, to me, the de novo ALK TKI resistance in ALK aberrant NBLs. Unfortunately, while the in vivo results are impressive for the combination of Lornfarnib and ceritinib causing what appears to be complete tumor regression, the very tiny tumor volume of 75mm³ at treatment initiation is not the gold standard for showing antitumor effects of agents on an established tumor burden (usually starting therapy when tumors reach 150-250 mm³). In addition, the additional detail provided in this rebuttal/revised manuscript still does not support that Lonafarnib is working to enhance ceritinib responses via inhibiting NRAS. I agree with other reviewers, that additional in vivo experiments to show how wild type ALK NBL PDXs respond to this combination as well as using an ALK-mutated tumor that has NRAS knocked out would help uncover true mechanisms for how Lonafarnib synergizes with ceritinib, especially in NBL models that have mutations like F1174L that are truly clinically resistant to single agent ceritinib. Stating that ALK protein expression is high in wild type NBL cells and thus driving those WT tumors to support results seen to the combinations of FTI and ALK TKI in WT ALK NBL in vitro is an assumption and was not experimentally proven.

Lastly, I could not reproduce the survival outcome data based on NRAS gene expression presented in extended figures. When making such large comparisons using the Kocak or TARGET datasets that contain large sample size, it is important to present the significance by showing the Bonferroni correction of the p value, which is not done. In addition, my calculations utilizing those exact same datasets shows that NRAS high gene expression is actually associated with improved survival in both the KOCAK and TARGET data (looking at all NBL patients). Given that the effect of NRAS expression on prognosis was a key driver for the authors to follow this lead as a target of their miRNA is also of concern.

Response to Reviewers

Reviewer #1 – Original

I would like to thank the authors for their efforts in addressing my concerns and performing the suggested experiments.

We thank the Reviewer for their positive feedback.

Reviewer #2 - Original

Authors did not address my request to analyze apoptosis *in vivo*, despite including a second apoptosis expt and re-illustrating for me that they did show this *in vitro*.

We thank the reviewer for their positive feedback, and we apologize for not fully addressing the request before. We understood the previous comment as a request of clarification, which we included in our answers.

Because we showed that the combination treatment induced apoptosis in the PDX cells *in vitro*, we would expect a similar effect *in vivo*, which is indirectly shown by the tumours' growth remission. Indeed, as the tumours in the combination treatment group were undetectable, there was no tumour left to analyse for apoptosis. However, we conducted additional experiments on the tumours from our *in vivo* studies of the remaining 3 treatment groups for which residual tumour was available (vehicle, certinib single agent, lonafarnib single agent), by staining the tumours with a cleaved caspase3 (CC3; CST, Cat#9661) antibody as a marker of apoptosis, and show a mean % of CC3+ cells at or below 15 for all 3 groups consistent with their continued growth. These data are shown below for the reviewer but we have not included them in the manuscript as we feel this does not add anything meaningful to the paper.

Reviewer #4 – Replacement for Reviewer #3

The revised manuscript and additional work to address reviewers' comments is much appreciated but does not improve upon the lack of novelty for this manuscript based on approach and agents used.

As mentioned, Berlak et al performed ALK TKI therapy with CRISPR screen following treatment with a more clinically applicable agent in NB along with ceritinib (lorlatinib) to identify ALK TKI resistance mechanisms and already showed dependence on the RAS pathway contributing to ALK TKI resistance.

It is unfortunate the reviewer does not see the novelty of our research. To reiterate, the Berlak et al. paper shows that NRAS mutations arise in NB following ALK inhibitor treatment resulting in constitutive RAS activation. NRAS activation was confirmed by western blotting for phosphorylated ERK1/2. By contrast, our data provide additional and further evidence of the role of NRAS (wild-type, i.e., not mutated) expression in resistance to ALK inhibitors due to loss of inhibition caused by *miR-1304-5p* loss and furthermore that inhibition of NRAS activity by lonafarnib activity *in vitro* and *in vivo* in ALK aberrant NB restores sensitivity to ALK TKI. Indeed, our data also demonstrate that ALK TKIs can be effective in ALK WT neuroblastoma, paving the way for their use in the clinic, beyond ALK-mutant NB. In further evidence of the latter, we have now also included additional data on the effect of combining the third generation ALK TKI lorlatinib with the FTI lonafarnib, which shows high synergy, thereby increasing the clinical applicability of our findings (Extended Data Figure 13).

Extended Data Figure 13. A combination of the ALK inhibitor lorlatinib and the FTI lonafarnib act synergistically in ALK mutant PDX and ALK WT cell lines via induction of apoptosis.

(A) Dose-response matrix of lonafarnib (0.1-3000nM) and lorlatinib (0.1-3000nM) alone or in combination, following 72 h incubation with COG-N-415 (ALK mutant, MYCN amplified) PDX cells. Loewe synergy scores (Synergy Finder) and cell viability (CTG) are the results from two biological replicates. Colour gradients: % cell viability normalised to DMSO (from green: 100%, to red: 0%). Scores >10 represent synergism. (B) Apoptosis (caspase 3/7 activity per cell population normalized to DMSO) of COG-N-415 cells treated with a combination of lonafarnib (1 μ M) and lorlatinib (1 μ M), or single agents (same doses), for 72 h. Results shown are representative of two biological replicates. (C-D) Loewe synergy scores calculated with the Synergy Finder tool⁸⁵ for COG-N-415 PDX cells treated with a combination of lonafarnib and lorlatinib shown as both 2D (C) and 3D (D) models and are represented as colour gradients from antagonistic (darker green, loewe score <-10) to synergistic (darker red, loewe score >10). Synergy scores >- 10 and <10 are considered additive. (E) Dose-response matrix of lonafarnib (1-3000nM) and lorlatinib (1-3000nM) alone or in combination, following 72 h incubation with NGP (ALK WT, MYCN amplified) cells. Loewe synergy scores (Synergy Finder) and cell viability (CTG) result from two biological replicates. Colour gradients: % cell viability normalised to DMSO (from green: 100%, to red: 0%). Scores >10 represent synergism. (F) Apoptosis (caspase 3/7 activity per cell population normalized to DMSO) of NGP cells treated with a combination of lonafarnib (1 μ M) and lorlatinib (1 μ M), or single agents (same doses), for 72 h. Results shown are representative of two biological replicates. (G-H) Loewe synergy scores calculated with the Synergy Finder tool⁸⁵ for NGP cells treated with a combination of lonafarnib and lorlatinib shown as both 2D (G) and 3D (H) models and are represented as color gradients from antagonistic (darker green, loewe score <-10) to synergistic (darker red, loewe score >10). Synergy scores >- 10 and <10 are considered additive.

While I agree with the authors that the focus of this manuscript diverged differently based on their CRISPR results, to focus on miRNA influences to ALKi resistance, the additional data does not completely support that miR-1304-5p directly suppresses NRAS,

In the revised manuscript, we addressed a request to perform mutagenesis at the site of interaction in the 3`UTR to strengthen the claim of a direct interaction, which was previously raised by reviewer 1: We analysed the ORF and UTR regions of the 4 target genes (including NRAS) and there are 16 possible seed regions; we would therefore need to mutate each one individually and perhaps in combination, which is a huge amount of work and outside the scope of this publication. Instead, we have addressed this caveat in the discussion (page 15, lines 4-11) referring to the experiment detailed below: “Due to the presence of 16 potential seed regions for direct targeting by *miR-1304-5p* in these 4 genes, site-directed mutagenesis to confirm direct action of this miRNA on these targets was not possible in this study and remains to be determined. It remains to be tested whether these genes are direct targets of *miR-1304-5p* or whether changes in their expression and activity are downstream effects. However, on overexpression of the 4 target genes individually and in combination with *miR-1304-5p* (Fig 3, extended data Fig 5), effects on the RAS/MAPK pathway are counteracted, which is supportive of these genes being direct targets of *miR-1304-5p*, although these data require further validation as detailed above”. Reviewer 1 was happy with this response.

nor that the rather effective in vivo combination of an FTI and ALK TKI are truly working through inhibition of NRAS gene expression specifically to directly improve in vivo ALK TKI responses. More work needs to be done to demonstrate their claim that Lornfarnib works in this setting via suppression of NRAS given the xenograft finding of decreased Phospho ERK expression is rather nonspecific, as pERK is downstream of many pathways.

Since lonafarnib is a farnesyltransferase (FT) inhibitor, this agent is not known to inhibit NRAS **expression**, but rather inhibits NRAS **activity** (due to the lack of the farnesyl group) and its downstream pathways (e.g., pERK). Hence, we are not saying that the combination works through inhibition of NRAS expression, rather through indirect downstream effects and we show this *in vivo* as can be seen in Fig. 7 and Extended data Fig 16D-E. This confirmation of suppression of NRAS activity was the same as that used in the study by Berlak cited by the reviewer (see above). However, we have conducted immunohistochemistry for NRAS expression (Origene, Cat#TA505835) in the PDX tumours treated with lonafarnib or ALK TKI and these data show that there is not a significant change in NRAS expression amongst the different treatment groups, as shown in the figure below.

In their response to review #3, the authors state: “Also, as biomarkers, miRNAs could also perform better than protein-coding genes (including NF1), considering miRNAs have already been detected in cancer tissues and liquid biopsies from cancer patients”.

In contrast to this comment, Berko et al. (Nat Comm 2023) showed very clearly that NF1 mutations (and other RAS pathway and ALK compound mutations) arise as a mechanism of ALK inhibitor (Lorlatinib) resistance clinically in ALK-driven relapsed/refractory neuroblastoma. This was detected in the circulating tumor DNA in real time from patient samples collected on the Phase I clinical trial. Therefore, it is unclear how detection of miRNA’s can outperform ctDNA evaluations for ras/alk resistance mutations as a biomarker of ALK TKI resistance (maybe I misunderstood, but protein coding genes are not being evaluated as integral biomarkers, only ctDNA for genomic alterations in said protein coding genes like NF1).

We thank the reviewer for the comments but in our answer to Reviewer 3’s comment we did not mean that *miR-1304* detection would be a better biomarker than NF1 mutations but rather an additional route/pathway that could be explored in the future. In several studies, combinations of biomarkers such as different miRNAs but also mutated genes and mRNAs or proteins too, have a diagnostic/prognostic benefit in patients when combined as shown from ROC/AUC analyses, either in serum, plasma or tumour tissues (1–3). Studies of liquid biopsies of *mir-1304-5p* have not been performed and are outside of the scope of this paper – we are merely saying that our findings perhaps present another opportunity for biomarker development in the future. However, we thank the reviewer for pointing out the interesting findings from the Berko et al. paper, which we had cited in our manuscript in the context of our research.

The authors defend their choice for using brigatinib and ceritinib as ALK inhibitors for their NBL CRISPR screen by saying: “Use of multiple ALK TKIs clinically for this patient subset is to be expected given the known need to clinically ‘rotate’ through ALK inhibitors for adults with ALK-positive lung cancer as well as children with ALK-positive anaplastic large cell lymphoma (ALCL).”

Stating that ALK fusion+ cancers and ALK mutated/amplified NBL are the same with respect to how ALK inhibitors are used, prioritized, or chosen is inappropriate.

Tumors that harbor ALK FUSIONS (ALCL, SCLC) are exquisitely sensitive to all generations of ALK inhibitors, leading patients to start with a first generation ALK TKI (crizotinib), then once resistant, rotate to second generation ALK TKI’s (ensartinib, brigatinib), and once resistant go on to 3rd generation TKI like lorlatinib (that was designed to target resistant ALK mutations that arise in ALK fusion + cancers treated with ALK TKI’s). This “rotation through many ALK TKI’s” is not the current clinical practice in neuroblastoma given early generation inhibitors, even in combination with chemotherapy, show very limited responses clinically and preclinically in NB.

This is because NBs harbor de novo ALK hotspot activating mutations (or amplification) that are intrinsically resistant to first and second generation ALK TKI’s, which has been shown and published, thus limiting choice of ALKi to use clinically. As mentioned, so far, only lorlatinib seems to show clinical responses across aggressive recurrent and newly diagnosed HR NBL tumors, with responses seen in all hotspot ALK mutations. Therefore, the lack of lorlatinib use on this study specifically to evaluate for clinically relevant NBL ALK TKI resistant mechanisms significantly dampens the possibility that these results and *miR-1304-5p* is clinically impactful as a biomarker.

We are aware of the differences in ALK status across cancers. Whilst lorlatinib has to date been the focus for clinical trials in NB, this does not exclude the use of other ALK TKIs in the future. While crizotinib alone is not a good choice for ALK mutant NB due to the inherently resistant nature of the e.g., ALK F1174L variant, other ALK TKIs are efficacious. Of note, the BrigaPED trial (EudraCT Number:2021-002713-34) is open to patients with ALK-aberrant relapsed/refractory solid tumours

including neuroblastoma and other ALK TKIs have shown promising effects in clinical trials of NB patients, including ceritinib (4, 5) and are currently used alone or in combination with other drugs (6). In addition, as mentioned above, we also show that combination therapies (i.e. lonafarnib with ALK TKIs including lorlatinib or ceritinib) can be effective in ALK WT neuroblastoma, paving the way for their use in the clinic, for NB patients without ALK mutations. In evidence of this, we show the effects of a combination of lonafarnib (FTI) and lorlatinib (ALK TKI) on cell viability (and apoptosis) in both the ALK WT NGP cell line and the PDX line COG-N-415 (Extended Data Fig. 13, shown above). In both cell lines this combination showed synergistic effects, suggesting that combining an FTI with lorlatinib could allow a reduction in the doses of single agents used in the treatment of patients whilst also increasing the effectiveness of the treatment.

I agree with the authors statement, however, that ALK TKI 's in combination with other targeted therapies is greatly warranted to overcome tumor heterogeneity and, to me, the de novo ALK TKI resistance in ALK aberrant NBLs. Unfortunately, while the in vivo results are impressive for the combination of Lonafarnib and ceritinib causing what appears to be complete tumor regression, the very tiny tumor volume of 75mm³ at treatment initiation is not the gold standard for showing antitumor effects of agents on an established tumor burden (usually starting therapy when tumors reach 150-250 mm³).

We have selected 75mm³ as a starting point based on previous studies in the literature, whereby treatment is initiated when tumours reach 75-100mm³ or 5-6mm diameter, for NB *in vivo* studies (please see (7-9)). In addition, some tumours in our study were larger (up to 171mm³) at treatment initiation and a strong effect was still observed. We would also like to add that under UK Home Office regulations which must be abided by in our studies, and according to Workman et al., mice have to be culled when tumours reach 1.5 cm in diameter for therapeutic studies for welfare reasons (10). Of note, in the guidelines, it is stated that: "Tumour burden should always be limited to the minimum required for a valid scientific outcome. For example, efficacy studies should be terminated once durable, statistically significant therapeutic effects can be shown." Given the aggressive growth rate of these tumours *in vivo*, whereby they almost double in size in as little as a couple of days, we must be careful to balance scientific rigour with animal welfare.

In addition, the additional detail provided in this rebuttal/revised manuscript still does not support that Lonafarnib is working to enhance ceritinib responses via inhibiting NRAS.

We thank the reviewer for this comment, but we would like to mention that NRAS is not the only target of lonafarnib, with lonafarnib being an FTI and furthermore, does not inhibit NRAS expression, rather its downstream activity, hence as mentioned above, we have shown downstream effects on pERK.

I agree with other reviewers, that additional in vivo experiments to show how wild type ALK NBL PDXs respond to this combination as well as using an ALK-mutated tumor that has NRAS knocked out would help uncover true mechanisms for how Lonafarnib synergizes with ceritinib, especially in NBL models that have mutations like F1174L that are truly clinically resistant to single agent ceritinib. Stating that ALK protein expression is high in wild type NBL cells and thus driving those WT tumors to support results seen to the combinations of FTI and ALK TKI in WT ALK NBL in vitro is an assumption and was not experimentally proven.

To our knowledge, the F1174L ALK mutation does not necessarily confer resistance to ceritinib (11). COG-N-415, which is the PDX model with the F1174L ALK mutation used in this study is indeed sensitive to ceritinib (300-1000nM) (Fig. 6), although was less sensitive to the concentration of ceritinib used *in vivo* (30mg/kg), which was administered at a sub-optimal level (Extended data Fig

15A), since the focus of this study was to examine the effect of the combination of ceritinib plus the FTI. Hence, a lower dose was used to highlight the synergistic effects of the combination over the single agent. We have also shown *in vitro* that upregulation of the miRNA inhibits growth of NB cells which have no ALK kinase domain mutations (LAN6 cell line) as well as those with kinase-domain mutant, hyperactive ALK (Fig 2f). In addition, 3 PDXs derived from ALK aberrant NB (COG-N-557, COG-N-415 and FELIX) respond to a combination of lofaranib and certitinib *in vitro* (fig 6). As the reviewer notes, we have not shown this for ALK WT cells, rather our data here is observational based on the results in fig 2E, F, which demonstrate the effects of *miR1304-5p* on cell viability. However, we show in the LAN-6 cell line, that the cells are responsive to the ALK TKI brigatinib and ceritinib as well as a combination of ALK TKI and the *miR-1304-5p* mimic (extended data fig 10). Additionally, as mentioned above, we now also show in the manuscript that a combination of the FTI lonafarnib and the ALK TKI lorlatinib is synergistic in both ALK mutant PDX lines and ALK WT NB cells (Extended Data Fig. 13). However, if the reviewer feels it necessary, we would be willing to tone down any claims regarding the activity of lonafarnib and ALK TKI for the treatment of ALK WT NB in the manuscript, by perhaps adding 'ALK-mutant NB' to the title of the manuscript.

Lastly, I could not reproduce the survival outcome data based on NRAS gene expression presented in extended figures. When making such large comparisons using the Kocak or TARGET datasets that contain large sample size, it is important to present the significance by showing the Bonferroni correction of the p value, which is not done.

The Bonferroni correction is a method to reduce the type I error rate across multiple comparisons. By default, the R2 platform "scans" a range of cutoff values, applying a log-rank test to each, and selects the cut-off point which yields the minimum P value. Using this methodology, type I error is inflated due to hundreds of comparisons, necessitating multiple-testing correction. In our analyses, we instead manually chose an NRAS expression value cut-off, based on the expression distribution of the cohort, as represented in Extended Data Figure 5 G-H. This was done because the population distribution appears bimodal. Therefore, we would find it helpful if the reviewer could please be more specific as we are not clear as to which data we should apply the Bonferroni correction.

In addition, my calculations utilizing those exact same datasets shows that NRAS high gene expression is actually associated with improved survival in both the KOCAK and TARGET data (looking at all NBL patients). Given that the effect of NRAS expression on prognosis was a key driver for the authors to follow this lead as a target of their miRNA is also of concern.

Could the reviewer be more specific on how they divided the population into high NRAS and low NRAS in their analyses? Because of the bimodal NRAS expression distribution mentioned above, we manually separated the smaller population (peak) with higher NRAS expression from the larger population with lower NRAS expression (Extended Data Figure 5 G-H).

Due to the reviewer's observation that the location of the dichotomisation can affect the direction of the NRAS expression effect, we additionally present analysis of the same datasets using cox proportional-hazards regression (R 4.2.2, `survival::coxph`), which avoids dichotomising the continuous expression data, and allows estimation of the effect of a quantity while adjusting for other known factors (such as age and sex). We performed cox regression for OS and EFS across all three datasets, detailed in the table below. The regressions were fit (in Wilkinson notation) as $\text{survival} \sim \text{NRAS_expression} + \text{sex} + \text{stage} + \text{age} + \text{mycn_status}$.

Hazard ratios and P values from Cox regression

dataset	survival type	HR for NRAS expr.	P value
Kocak	EFS	1.0006567	0.005765

dataset	survival type	HR for NRAS expr.	P value
	OS	1.0006118	0.020748
SEQC	EFS	1.006576	0.08913
	OS	1.012641	0.00513
TARGET	EFS	1.09791	0.000127
	OS	1.05432	0.0347

The hazard ratios are uniformly greater than one, indicating increased hazard (worse survival) with increasing NRAS expression.

Note: in the Kocak dataset, using the scan cut-off method in R2, many arbitrary genes such as GAPDH, CATSPER2 (found in sperm), and ADIPOQ (expressed in adipose tissue) yield significant (Bonferroni-corrected) relationships with overall survival.

References

1. Y. Wada, M. Shimada, Y. Morine, T. Ikemoto, Y. Saito, Z. Zhu, X. Wang, A. Etxart, Y. Park, L. Bujanda, I. J. Park, A. Goel, Circulating miRNA Signature Predicts Response to Preoperative Chemoradiotherapy in Locally Advanced Rectal Cancer, *JCO Precis. Oncol.*, 1788–1801 (2021).
2. Z. Yang, M. J. LaRiviere, J. Ko, J. E. Till, T. Christensen, S. S. Yee, T. A. Black, K. Tien, A. Lin, H. Shen, N. Bhagwat, D. Herman, A. Adallah, M. H. O'Hara, C. M. Vollmer, B. W. Katona, B. Z. Stanger, D. Issadore, E. L. Carpenter, A multianalyte panel consisting of extracellular vesicle miRNAs and mRNAs, cfDNA, and CA19-9 shows utility for diagnosis and staging of pancreatic ductal adenocarcinoma, *Clin. Cancer Res.* **26**, 3248–3258 (2020).
3. S. Du, Y. Zhao, C. Lv, M. Wei, Z. Gao, X. Meng, Applying Serum Proteins and MicroRNA as Novel Biomarkers for Early-Stage Cervical Cancer Detection, *Sci. Rep.* **10**, 20–22 (2020).
4. M. Fischer, L. Moreno, D. S. Ziegler, L. V. Marshall, C. M. Zwaan, M. S. Irwin, M. Casanova, C. Sabado, B. Wulff, M. Stegert, L. Wang, F. K. Hurtado, F. Branle, B. Georger, J. H. Schulte, Ceritinib in paediatric patients with anaplastic lymphoma kinase-positive malignancies: an open-label, multicentre, phase 1, dose-escalation and dose-expansion study, *Lancet Oncol.* **22**, 1764–1776 (2021).
5. J. Guan, S. Fransson, J. T. Siaw, D. Treis, J. Van Den Eynden, D. Chand, G. Umopathy, K. Ruuth, P. Svenberg, S. Wessman, A. Shamikh, H. Jacobsson, L. Gordon, J. Stenman, P. J. Svensson, M. Hansson, E. Larsson, T. Martinsson, R. H. Palmer, P. Kogner, B. Hallberg, Clinical response of the novel activating ALK-I1171T mutation in neuroblastoma to the ALK inhibitor ceritinib, *Cold Spring Harb. Mol. Case Stud.* **4**, 1–23 (2018).
6. K. Schellekens, R. A. Schoot, N. K. A. van Eijkelenburg, A. Huitema, M. Casanova, D. Reinhardt, L. V. Marshall, I. Aerts, G. Barone, L. Chesler, J. Tall, D. Hughes, R. Dandis, L. Moreno, P. L. H. Winkler-Seinstra, K. D. Wilner, C. M. Zwaan, J. van der Lugt, A phase 1b study of crizotinib in combination with temsirolimus in pediatric ALK- or MET-aberrated relapsed or refractory neuroblastoma (ITCC-053): Results of the phase 1 part., *J. Clin. Oncol.* **41**, 10036–10036 (2023).
7. E. R. Tucker, I. Jiménez, L. Chen, A. Bellini, C. Gorrini, E. Calton, Q. Gao, H. Che, E. Poon, Y. Jamin, B. M. Da Costa, K. Barker, S. Shrestha, J. C. Hutchinson, S. Dhariwal, A. Goodman, E. Del Nery, P. Gestraud, J. Bhalshankar, Y. Iddir, E. Saber-Ansari, A. Saint-Charles, B. Georger, M. E. M. Da Costa, C. Pierre-Eugène, I. Janoueix-Lerosey, D. Decaudin, F. Nemati, A. M. Carcaboso, D. Surdez, O. Delattre, S. L. George, L. Chesler, D. A. Tweddle, G. Schleiermacher, Combination Therapies Targeting ALK-aberrant Neuroblastoma in Preclinical Models, *Clin. Cancer Res.* **29**, 1317–1331 (2023).
8. K. M. Ferguson, S. L. Gillen, L. Chaytor, E. Poon, D. Marcos, R. L. Gomez, L. M. Woods, L. Mykhaylechko, L. Elfari, B. Martins da Costa, Y. Jamin, J. S. Carroll, L. Chesler, F. R. Ali, A. Philpott, Palbociclib releases the latent differentiation capacity of neuroblastoma cells., *Dev. Cell* **58**, 1967-1982.e8 (2023).
9. R. M. Trigg, L. C. Lee, N. Prokoph, L. Jahangiri, C. P. Reynolds, G. A. Amos Burke, N. A. Probst, M. Han, J. D. Matthews, H. Kai Lim, E. Manners, S. Martinez, J. Pastor, C. Blanco-Aparicio, O. Merkel, I. G. de los Fayos Alonso, P. Kodajova, S. Tangermann, S. Högl, J. Luo, L. Kenner, S. D. Turner, The targetable kinase PIM1 drives ALK inhibitor resistance in high-risk neuroblastoma independent of MYCN status, *Nat. Commun.* **10**, 5428 (2019).
10. P. Workman, E. O. Aboagye, F. Balkwill, A. Balmain, G. Bruder, D. J. Chaplin, J. A. Double, J. Everitt, D. A. H. Farningham, M. J. Glennie, L. R. Kelland, V. Robinson, I. J. Stratford, G. M. Tozer, S. Watson, S. R. Wedge, S. A. Eccles, V. Navaratnam, S. Ryder, Guidelines for the welfare and use of animals in cancer research, *Br. J. Cancer* **102**, 1555–1577 (2010).
11. D. N. Debruyne, N. Bhatnagar, B. Sharma, W. Luther, N. F. Moore, N. K. Cheung, N. S. Gray, R. E. George, ALK inhibitor resistance in ALK^{F1174L}-driven neuroblastoma is associated with AXL activation and induction of EMT, *Oncogene* **35**, 3681–3691 (2016).

Reviewers' Comments:

Reviewer #4:

Remarks to the Author:

The abstract still states FTI's synergize with ALK TKI's by "inducing apoptosis both in vitro and in vivo". The in vivo in that statement should be removed from abstract since the authors did not show apoptosis in vivo.

I agree with the authors that they should change the title to say "ALK mutant neuroblastoma" as the full mechanism of this combination in ALK wild type NBs has not been fully explored and in vivo studies of FTI + ALK TKI combinations were limited to ALK mutated NBs

While I appreciate the effort made to test lorlatinib with Lornafarnib, the new data (Extended Data Figure 13) shows questionable efficacy results of the lorlatinib monotherapy lane that affects any interpretation of the combination therapy. In prior publications that used clinical grade lorlatinib (PF-06463922, Pfizer), the same COG-415X cell line used in this paper was exquisitely sensitive to lorlatinib with IC50 of ~ 15 nanomolar (nM) in vitro (in addition to similar low nM IC50 of many other F1174 ALK mutated NBs), using the same cell survival assay read out (CTG) these authors used for their synergy assay (Infinato, Cancer Discovery 2016). In contrast, extended Figure 13A shows no cytotoxicity (all green=100% viable) of COG415X in vitro up to 3000 nM of lorlatinib (bottom line of Ext Fig. 13A, where Lornafarnib is 0nM, thus showing single agent activity of lorlatinib). I wonder if the lack of efficacy may be due to the compound used to represent lorlatinib or the age of the lorlatinib if it was indeed clinical grade? Based on this very large discrepancy, Extended figure 13 should be removed along with reference to it in the results section, as the "synergy" data is uninterpretable.

I would also consider toning down the last statement of the combination of lornafarnib and ALK TKI leading relapsed NB to become a chronic disease rather than lethal one, as the in vivo treatments performed were all very short lived (15-20 days) and you do not present data showing that sustained treatment of this combination in vivo (over 60 days or more) does not lead to progressive/resistant disease or more toxicity. As such, I think your in vivo data reflects that this combination is worthy of further investigation for response in ALK wild type NB and to ensure tolerability of a more lengthy treatment in ALK mutant NB models, especially since, clinically, there are a lot of toxicities of lornafarnib that led to clinical discontinuation in up to 20% of patients on multiple clinical trials and those particular toxicities (GI, LIVER) are also found in brigatinib and ceritinib clinically as well, that may not be captured in brief mouse tox studies.

Minor:

Please ensure the axis labels and numbering for synergy experiments are consistent and that they make sense - in some of the extended figs, the read out for the caspase 3/7 experiments goes up to 1000, yet the labeling states this is in percent (%) CASP3/7 divided by alive cells. In other figures of Casp 3/7 activation, it is just a number 1-8 and not a percent, yet with same Y axis label.

Titles of sections and figure legends need to accurately reflect the data shown:

Line 846. Figure 2. "Inhibition of miR-1304 decreases the sensitivity of a range of NB cell lines to brigatinib and ceritinib" - in reality, Figure 2 (A-D) only shows ONE NB CELL LINE (SH-SY5Y) treated with the ALK inhibitors following miR-1304 inhibition. Figure 2 E-F only show a panel of NB cell lines with mir1304 inhibited/expressed alone (no brigatinib or ceritinib are used) so the title does not accurately depict what is in the figure. Therefore, Please change title of Line 91 to read Inhibition of miRNA miR-1304-5p decreases sensitivity to ALK TKIs in an ALK mutant NB. And please change the title of the Figure 2 legend, Line 846 to something like, "miR-1304 inhibition affects viability of a range of NB cell lines and increases sensitivity of ALK mutant NB cells to brigatinib and ceritinib"

Line 258 should say "in ALK-mutated NB patient derived xenografts" as no wild type NB PDX's were tested in vivo.

Line 340, 384 and line 394, etc: Most NBs express the ALK gene and ALK protein, including ALK

wild type tumors, so I would change the wording from "ALK-expressing" to either "ALK-mutated" or "ALK-aberrant" and italicize ALK if referring to the gene.

Responses to reviewer #4

The abstract still states FTI's synergize with ALK TKI's by "inducing apoptosis both in vitro and in vivo". The in vivo in that statement should be removed from abstract since the authors did not show apoptosis in vivo.

The reference to '*in vivo*' has been removed.

I agree with the authors that they should change the title to say "ALK mutant neuroblastoma" as the full mechanism of this combination in ALK wild type NBs has not been fully explored and in vivo studies of FTI + ALK TKI combinations were limited to ALK mutated NBs

We have changed the title.

While I appreciate the effort made to test lorlatinib with Lornafarnib, the new data (Extended Data Figure 13) shows questionable efficacy results of the lorlatinib monotherapy lane that affects any interpretation of the combination therapy. In prior publications that used clinical grade lorlatinib (PF-06463922, Pfizer), the same COG-415X cell line used in this paper was exquisitely sensitive to lorlatinib with IC50 of ~ 15 nanomolar (nM) in vitro (in addition to similar low nM IC50 of many other F1174 ALK mutated NBs), using the same cell survival assay read out (CTG) these authors used for their synergy assay (Infarinato, Cancer Discovery 2016). In contrast, extended Figure 13A shows no cytotoxicity (all green=100% viable) of COG415X in vitro up to 3000 nM of lorlatinib (bottom line of Ext Fig. 13A, where Lornafarnib is 0nM, thus showing single agent activity of lorlatinib). I wonder if the lack of efficacy may be due to the compound used to represent lorlatinib or the age of the lorlatinib if it was indeed clinical grade? Based on this very large discrepancy, Extended figure 13 should be removed along with reference to it in the results section, as the "synergy" data is uninterpretable.

We thank the reviewer for this feedback, but we are reporting the data as we see them in our hands with the cells and compound that we have. There are perhaps some subtle differences that might account for the differences in our findings to those of Infarinato et al, which may be in part due to the source of the compound (ours was purchased from MedChem Express, was dissolved in DMSO and diluted in media before cells were exposed for 72 hours and outcome assessed by CTG). In Infarinato et al., they used the compound directly supplied by Pfizer and a 120-hour drug exposure with viability assessed by CTG. MedChem Express states that the purity of their compound is greater than 99% and we prepare frozen aliquots in DMSO which are stored at -20 before defrosting, diluting in media before applying to cells. This should preserve the stability and activity of the compound which was purchased no more than 6 months before use and shipped to us. We have also compared the data to other published reports and there is considerable variation, even with well-established cell lines as shown in the table below. In general, drug direct from Pfizer seems to have better activity but there is still variability between studies and cell lines with this source. Equally, a longer exposure time of 120 hours compared to 72 hours also shows a stronger response, perhaps also accounting for some of the differences. Importantly, we have used consistent methods and drug preparation throughout our studies enabling comparisons to be made across our data.

Cell line	Lorlatinib IC50/GI50 (nM)	Reference	Source of lorlatinib	Notes on experimental conditions
COG-N-415	~3000 (3µM)	Our Manuscript	MedChem Express	CTG used to assess viability following 72 hours exposure
	1000-2200 (1-2.2 µM)	PMID: 38032104 (Valencia-Sama et al)	Selleckchem	AlamarBlue used to assess viability after 72 hours exposure
	15	Infarinato et al	Pfizer	Viability assessed using CTG 120 hours after treatment
NGP	>3000 (3µM)	Our Manuscript	MedChem Express	CTG used to assess viability following 72 hours exposure
	28,000 (28µM)	PMID: 38032104	Selleckchem	AlamarBlue used to assess

		(Valencia-Sama et al)		viability after 72 hours exposure
	609	PMID: 36602782 (Tucker et al., suppl tab.1)	Selleckchem	XTT assay used to assess viability following 72 hours exposure
SHSY5Y	2000	Our unpublished data	MedChem Express	CTB used to assess viability following 72 hours exposure
	800±500	PMID: 38032104 (Valencia-Sama et al)	Selleckchem	AlamarBlue used to assess viability after 72 hours exposure
	3,674.8 ± 990.3	PMID: 36602782 (Tucker et al., suppl tab.1)	Selleckchem	XTT assay used to assess viability following 72 hours exposure
	300	PMID: 30322862 (Radaelli et al.,)	Pfizer	CTG used to assess viability following 72 hours exposure
	25,000 (25µM)	PMID: 30459283 (Emdal et al.,)	Selleckchem	ATP lite assay used to assess viability after 48 hours exposure
	16.5	Infarinato et al	Pfizer	viability assessed using CTG 120 hours after treatment
KELLY	600	Our unpublished data	MedChem Express	CTB used to assess viability following 72 hours exposure
	1200±1000	PMID: 38032104 (Valencia-Sama et al)	Selleckchem	AlamarBlue used to assess viability after 72 hours exposure
	35±6	PMID: 37147298 (Berko et al.,)	Selleckchem	Viability assessed using CTG 120 hours after treatment
	26.6	Infarinato et al	Pfizer	viability assessed using CTG 120 hours after treatment

I would also consider toning down the last statement of the combination of lornafarnib and ALK TKI leading relapsed NB to become a chronic disease rather than lethal one, as the in vivo treatments performed were all very short lived (15-20 days) and you do not present data showing that sustained treatment of this combination in vivo (over 60 days or more) does not lead to progressive/resistant disease or more toxicity. As such, I think your in vivo data reflects that this combination is worthy of further investigation for response in ALK wild type NB and to ensure tolerability of a more lengthy treatment in ALK mutant NB models, especially since, clinically, there are a lot of toxicities of lornafarnib that led to clinical discontinuation in up to 20% of patients on multiple clinical trials and those particular toxicities (GI, LIVER) are also found in brigatinib and ceritinib clinically as well, that may not be captured in brief mouse tox studies.

We appreciate these comments and the clinical toxicity associated with lornafarnib. Indeed, longer mouse studies are required which focus on the long-term outcomes and toxicities associated with these combinations of drugs. As such, we have toned down our descriptions and interpretation of these data, and have removed the last sentence of the abstract, and the final 3 sentences of the discussion have been changed.

Minor:

Please ensure the axis labels and numbering for synergy experiments are consistent and that they make sense - in some of the extended figs, the read out for the caspase 3/7 experiments goes up to 1000, yet the labeling states this is in percent (%) CASP3/7 divided by alive cells. In other figures of Casp 3/7 activation, it is just a number 1-8 and not a percent, yet with same Y axis label.

Many thanks for pointing out this oversight – this has been corrected.

Titles of sections and figure legends need to accurately reflect the data shown:

Line 846. Figure 2. "Inhibition of miR-1304 decreases the sensitivity of a range of NB cell lines to brigatinib and ceritinib" - in reality, Figure 2 (A-D) only shows ONE NB CELL LINE (SH-SY5Y) treated with the ALK inhibitors following miR-1304 inhibition. Figure 2 E-F only show a panel of NB cell lines with mir1304 inhibited/expressed alone (no brigatinib or ceritinib are used) so the title does not accurately depict what is in the figure. Therefore, Please change title of Line 91 to read Inhibition of miRNA miR-1304-5p decreases sensitivity to ALK TKIs in an ALK mutant NB. And please change the

title of the Figure 2 legend, Line 846 to something like, "miR-1304 inhibition affects viability of a range of NB cell lines and increases sensitivity of ALK mutant NB cells to brigatinib and ceritinib"
Line 258 should say "in ALK-mutated NB patient derived xenografts" as no wild type NB PDX's were tested in vivo.

Line 340, 384 and line 394, etc: Most NBs express the ALK gene and ALK protein, including ALK wild type tumors, so I would change the wording from "ALK-expressing" to either "ALK-mutated" or "ALK-aberrant" and italicize ALK if referring to the gene.

We have changed the requested titles and legends as requested.